# Improved Regret Bounds for Non-Convex Online-Within-Online Meta Learning

**Jiechao Guan[1], Hui Xiong[1, 2, ∗]**
[1]AI Thrust, The Hong Kong University of Science and Technology (Guangzhou), China
[2]Guangzhou HKUST Fok Ying Tung Research Institute, China
{jiechaoguan, xionghui}@hkust-gz.edu.cn

## Abstract

Online-Within-Online (OWO) meta learning stands for the online multi-task learning paradigm in which both tasks and data within each task become available in a sequential order. In this work, we study the OWO meta learning of the initialization and step size of within-task online algorithms in the non-convex setting, and provide improved regret bounds under mild assumptions of loss functions. Previous work analyzing this scenario has obtained for bounded and piecewise Lipschitz functions an averaged regret bound $O((\frac{\sqrt{m}}{T^{1/4}} + \frac{(\log m)\log T}{\sqrt{T}} + V)\sqrt{m})$ across $T$ tasks, with $m$ iterations per task and $V$ the task similarity. Our first contribution is to modify the existing non-convex OWO meta learning algorithm and improve the regret bound to $O((\frac{1}{T^{1/2-\alpha}} + \frac{(\log T)^{9/2}}{T} + V)\sqrt{m})$, for any $\alpha \in (0, 1/2)$. The derived bound has a faster convergence rate with respect to $T$, and guarantees a vanishing task-averaged regret with respect to $m$ (for any fixed $T$). Then, we propose a new algorithm of regret $O((\frac{\log T}{T} + V)\sqrt{m})$ for non-convex OWO meta learning. This regret bound exhibits a better asymptotic performance than previous ones, and holds for any bounded (not necessarily Lipschitz) loss functions. Besides the improved regret bounds, our contributions include investigating how to attain generalization bounds for statistical meta learning via regret analysis. Specifically, by online-to-batch arguments, we achieve a transfer risk bound for batch meta learning that assumes all tasks are drawn from a distribution. Moreover, by connecting multi-task generalization error with task-averaged regret, we develop for statistical multi-task learning a novel PAC-Bayes generalization error bound that involves our regret bound for OWO meta learning.

## 1 Introduction

Meta learning, also referred to as *learning to learn*, is a multi-task learning paradigm that transfers knowledge from past tasks to the new task for fast adaptation (Thrun & Pratt, 1998). Due to the advantage of reducing annotation cost and training time for novel task, meta learning has received increasing attention over the last decade, both from practical (Finn et al., 2017; Snell et al., 2017; Ye et al., 2020) and theoretical perspectives (Baxter, 2000; Balcan et al., 2019; Chen et al., 2020).

Traditional meta learning theory mainly investigates generalization bounds for statistical/batch meta learning (Baxter, 2000; Pentina & Lampert, 2014; Maurer et al., 2016; Guan et al., 2022). In this setting, samples within each task are assumed to be drawn from the same data distribution and processed in a batch, and different tasks are drawn from the same task distribution. Recently, there emerges an interest in studying online meta learning, where tasks are observed sequentially and require real-time processing (Alquier et al., 2017; Finn et al., 2019). According to (Denevi et al., 2019), there are two main frameworks in online meta learning setting: (1) Online-Within-Batch (OWB) framework, where tasks are available in a sequential order but the data within each task are available in one batch, see (Finn et al., 2019; Zhuang et al., 2020; Acar et al., 2021). (2) Online-Within-Online (OWO) framework, where both tasks and data within each task are available and processed sequentially, see (Alquier et al., 2017; Denevi et al., 2019; Khodak et al., 2019; Balcan et al.,

---

∗Corresponding Author.

2019; 2021). The goal of this paper is to investigate the OWO meta learning of *initialization-based* online optimization algorithms in the non-convex setting, and provide strong theoretical guarantees.

Initialization-based online algorithms (e.g. online gradient descent (Zinkevich, 2003)) typically have two hyper-parameters to learn: the initialization of model's parameter, and step size to update the parameter. OWO meta learning aims to utilize knowledge from previous tasks to set a good initialization and step size of online algorithm on novel task for fast adaptation. Recall that the metric used to measure the performance of online single-task learner is the so-called regret, which is defined as the gap between cumulative loss of online learner and the cumulative loss of the best fixed strategy in hindsight. Analogously, the metric used to measure the performance of OWO meta-learner is the averaged regret across $T$ training tasks, denoted as *task-averaged regret* (Balcan et al., 2019). Previous work (Balcan et al., 2021) analyzing the non-convex OWO meta learning of initialization-based online algorithms has attained for bounded and piecewise Lipschitz functions a task-averaged regret bound $O((\frac{\sqrt{m}}{T^{1/4}} + \frac{(\log m)\log T}{\sqrt{T}} + V)\sqrt{m})$, with $m$ iterations per task and $V$ the task similarity. However, this bound has two limitations: (1) For any fixed $T$, it cannot guarantee a vanishing regret with respect to (w.r.t.) $m$; (2) The convergence rate w.r.t. $T$ is slow. In this work, we attempt to address the two issues and provide improved regret bounds with fine-grained analysis.

Our first contribution is based on the modification of existing non-convex OWO meta learning algorithm from Balcan et al. (2021). We improve the regret bounds of online algorithms for learning initialization and step size respectively, and combine them to get the sharper task-averaged regret bound of $O((\frac{1}{T^{1/2-\alpha}} + \frac{(\log T)^{9/2}}{T} + V)\sqrt{m})$, for any $\alpha \in (0, 1/2)$. The derived bound has a faster convergence rate w.r.t. $T$, and guarantees a vanishing task-averaged regret w.r.t. $m$ (for any fixed $T$). Then, we propose a new and more efficient algorithm of regret $O((\frac{\log T}{T} + V)\sqrt{m})$ for non-convex OWO meta learning. This regret bound exhibits a better asymptotic performance than previous ones, and holds for any bounded (not necessarily Lipschitz) loss functions. Furthermore, we show how to attain generalization bounds for statistical meta learning via regret analysis. Specifically, by online-to-batch arguments, we achieve a new transfer risk bound for non-convex batch meta learning in which different tasks are drawn from a task distribution. Moreover, by revealing the connection between multi-task generalization error and task-averaged regret, we develop for statistical multi-task learning a novel PAC-Bayes generalization error bound that involves our improved regret bound for non-convex OWO meta learning setting. To the best of our knowledge, this is the first PAC-Bayes bound for batch/online multi-task setting that imposes distribution assumption over the data per task.

To summarize, our contributions are four-fold: **(1)** For non-convex OWO meta learning, we improve regret bound from $O((\frac{\sqrt{m}}{T^{1/4}} + \frac{(\log m)\log T}{\sqrt{T}} + V)\sqrt{m})$ to $O((\frac{1}{T^{1/2-\alpha}} + \frac{(\log T)^{9/2}}{T} + V)\sqrt{m})$ ($\alpha \in (0, 1/2)$) for bounded and piecewise Lipschitz functions. **(2)** We design a new and efficient OWO meta learning algorithm of sharper regret bound $O((\frac{\log T}{T} + V)\sqrt{m})$ for bounded (not necessarily Lipschitz) functions. **(3)** We obtain a new transfer risk bound for statistical meta learning via regret analysis. **(4)** We derive a PAC-Bayes bound for multi-task learning, shedding light on proving PAC-Bayes multi-task generalization bound with the regret bound from non-convex OWO meta learning.

## 2 RELATED WORK

**Regret Bounds for Online-Within-Batch Meta Learning**. In OWB meta learning setting, the meta parameter is transferred and updated sequentially across $T$ tasks. Concretely, when encountering a training task, OWB meta learning algorithm takes the meta parameter and the training data from the task as input, and outputs a model suitable for that task. The incurred loss of the task-specific model over the evaluation data is denoted as the online loss on the task. The gap between the cumulative loss of updated meta parameters across $T$ tasks and the cumulative loss of the best fixed meta parameter in hindsight is defined as the regret. Thus, the regret bound for OWB meta learning is of $O(\log T)$ in the strongly-convex setting (see (Finn et al., 2019, Cor 2)), and of $O(\sqrt{T})$ in the convex/non-convex setting (see (Acar et al., 2021, Thm 1) and (Zhuang et al., 2020, Thm 1)). These regret bounds are irrelevant to the sample size $m$ per task, and are achieved under very strong assumptions of loss functions (e.g. including Lipschitzness, smoothness, and Hessian smoothness).

**Regret Bounds for Online-Within-Online Meta Learning**. OWO meta learning is more challenging than OWB meta learning. Existing algorithms in OWO meta learning can be categorized into two groups: (1) The first group learns a common meta parameter (e.g. a feature map

(Alquier et al., 2017) or a meta regularization function (Denevi et al., 2019)) shared across $T$ tasks and uses the meta parameter to learn specific model for each task. The averaged regret bound across $m$ iterations and $T$ tasks in (Alquier et al., 2017, Thm 3.1) is of $O(1/\sqrt{m} + 1/\sqrt{T})$. The regret bound in (Denevi et al., 2019, Thm 1) is of $O(\sqrt{\log m/m} + 1/\sqrt{T})$. However, both works focus on convex OWO meta learning, and it is hard to extend their analysis (e.g. especially the primal-dual technique in Denevi et al. (2019)) to the non-convex setting. **(2)** The second group learns the initialization and step size of online algorithms for each task, and combines the regrets for learning these two hyper-parameters to obtain ultimate averaged regret across $T$ tasks. Concretely, the task-averaged regret bound in (Khodak et al., 2019, Thm 5.1) is of $O((\log T/T + \log T/\sqrt{T} + V)\sqrt{m})$ for convex and Lipschitz functions. The bound in (Balcan et al., 2021, Thm 3.3) is of $O((\sqrt{m}/T^{1/4} + (\log m)\log T/\sqrt{T} + V)\sqrt{m})$ for non-convex and piecewise Lipschitz functions. Our work focuses on non-convex setting, improves regret bound to $O((1/T^{1/2-\alpha} + (\log T)^{9/2}/T + V)\sqrt{m})$ in Theorem 2 for piecewise Lipschitz functions, and further obtains a sharper regret bound of $O((\log T/T + V)\sqrt{m})$ in Theorem 3 for non-Lipschitz functions.

**Generalization Bounds for Statistical Meta Learning**. The basic assumption in statistical meta learning theory is that all tasks are sampled from the same task distribution (Baxter, 2000). Under this assumption, Guan et al. (2022) summarize existing generalization bounds for meta learning into three groups: (1) Generalization bounds based on hypothesis space complexity (e.g. covering number based ones (Baxter, 2000) and Gaussian complexity based ones (Maurer et al., 2016)). (2) Generalization bounds based on PAC-Bayes theory (Pentina & Lampert, 2014; 2015; Guan & Lu, 2022). (3) Generalization bounds based on algorithmic stability (Maurer, 2005; Chen et al., 2020). However, with the tool of stability analysis, Guan et al. (2022) point out that, under the statistical task distribution assumption, the optimal generalization bound for meta learning is of $O(1/\sqrt{T})$, where $T$ is the number of training tasks. The optimal bound is irrelevant to the sample size $m$ per task and hence is slow, indicating the limitation of task distribution assumption. Our transfer risk bound in Theorem 4 via regret analysis is different from the above generalization bounds for meta learning. Our bound of $O(V^2/\sqrt{m})$ decreases with higher similarity among $T$ training tasks, shedding more light on the generalization ability of meta learning models. More discussions about generalization bounds for statistical multi-task learning can be found in Section A of the Appendix.

## 3 PRELIMINARY

In online learning, an action space $\Theta$ is a compact subset of $\mathbb{R}^d$. A loss function $\ell : \Theta \mapsto \mathbb{R}_{\geq 0}$ is called $\alpha$-strongly convex with respect to (w.r.t.) certain norm $\|\cdot\|$ in $\mathbb{R}^d$, if for any $x, y \in \Theta$, we have $\ell(x) - \ell(y) \geq \langle g, x - y \rangle + \frac{\alpha}{2}\|x - y\|^2$, where $g \in \partial\ell(y)$ and $\partial\ell(y)$ denotes the set of subgradients of $\ell$ at $y$. If $\ell$ is differentiable, $\partial\ell$ denotes the set of (unique) gradient of $\ell$, i.e. $\partial\ell = \{\nabla\ell\}$. When $\alpha = 0$, $\ell$ is called convex. $\ell$ is called uniformly $L$-Lipschitz over $\Theta$ w.r.t. the norm $\|\cdot\|$, if for any $x, y \in \Theta$, $|\ell(x) - \ell(y)| \leq L\|x - y\|$. Let $\mathcal{P}(\Theta)$ be the set of all probability distributions over action space $\Theta$. For any $\rho \in \mathcal{P}(\Theta)$ and any loss function $\ell$, we use $\langle \rho, \ell \rangle = \mathbb{E}_{\theta \sim \rho}\ell(\theta)$ for brevity when the context is clear. We use boldface letter $\boldsymbol{x}$ to denote the real vector in high-dimensional space, and $\hat{\boldsymbol{x}}$ as its normalized version $\boldsymbol{x}/\|\boldsymbol{x}\|$. We also use abbreviation $[m] = \{1, 2, ..., m\}$ for any integer $m$.

**Online Learning**. Vanilla online learning (also called online single-task learning setting) is always cast as a $m$-round optimization problem . At each round $i \in [m]$, the online learner selects an action $\theta_i \in \Theta$. Then a loss function $\ell_i : \Theta \mapsto \mathbb{R}_{\geq 0}$ is revealed from the nature and the online learner suffers loss $\ell_i(\theta_i)$ of the chosen action $\theta_i$. The quantity used to measure the performance of online learner is the so-called regret, which is the difference between the cumulative loss of chosen actions during $m$ rounds and the cumulative loss of the best fixed action in hindsight. Detailed introduction of regret bounds for convex/non-convex online learning can be found in Section A of the Appendix.

**Online-Within-Online Meta Learning**. In non-convex OWO meta learning setting, the online meta-learner will encounter $T$ tasks and each task $t \in [T]$ is composed of a sequence of $m$ loss functions $\{\ell_{ti} : \Theta \mapsto \mathbb{R}_{\geq 0}\}_{i \in [m]}$. Concretely, at $i$-th round of the $t$-th task, the online meta-learner selects an action $\theta_{ti} \in \Theta$, and then suffers the loss $\ell_{ti}(\theta_{ti})$, with $\ell_{ti}$ chosen adversarially from the nature. The regret for task $t$ is defined as $R_{t,m} \triangleq \sum_{i=1}^m \mathbb{E}_{\theta \sim \rho_{ti}}\ell_{ti}(\theta) - \min_{\theta \in \Theta}\sum_{i=1}^m \ell_{ti}(\theta)$. Then, the quantity to measure the performance of online meta-learner is the following task-averaged regret:

$$\bar{R}_{T,m} = \frac{1}{T}\sum_{t=1}^T R_{t,m} = \frac{1}{T}\sum_{t=1}^T\sum_{i=1}^m \mathbb{E}_{\theta \sim \rho_{ti}}\ell_{ti}(\theta) - \ell_{ti}(\theta_t^*), \quad where \quad \theta_t^* \in \arg\min_{\theta \in \Theta}\sum_{i=1}^m \ell_{ti}(\theta). \quad (1)$$

If we set the probability distribution $\rho_t^*$ as Dirac measure $\delta_{\theta_t^*}$ that only has mass 1 at the point $\theta_t^*$, we can rewrite the above task-averaged regret in a concise form: $\bar{R}_{T,m} = \frac{1}{T}\sum_{t=1}^{T}\sum_{i=1}^{m}\langle\rho_{ti} - \rho_t^*, \ell_{ti}\rangle$. The upper bound on $\bar{R}_{T,m}$ is denoted as $\bar{U}_{T,m}$. The goal of OWO meta learning is to improve the single-task regret by leveraging information from other tasks. Formally speaking, we expect that when $T$ is large enough, the task-averaged regret $\bar{R}_{T,m}$ is smaller than the single-task regret $R_{t,m}$.

# 4 REGRET BOUNDS FOR NON-CONVEX ONLINE META LEARNING

Following previous work (Balcan et al., 2021), we study the non-convex OWO meta learning of the initialization and step size of the Exponentially Weighted Aggregation (EWA) algorithm. We first provide a general framework for analyzing the regret bounds for non-convex OWO meta Learning. Such framework is motivated by the form of regret upper bound of EWA (see regret bound for piecewise Lipschitz functions in Eq. (2) and regret bound for non-Lipschitz functions in Eq. (3)): $U_t(\rho_{t1}, \lambda_t) = mb\lambda_t + \frac{V(\rho_{t1},\rho_t^*)^2}{\lambda_t} + g(m)$, where $b > 0$, $\lambda_t$ is the step size, $\rho_{t1} \in \mathcal{P}(\Theta)$ the initialization distribution over the action space $\Theta$ for task $t$, $\rho_t^* \in \mathcal{P}(\Theta)$ may have some dependence on the optimal action $\theta_t^*$, $V(\rho_{t1}, \rho_t^*)^2$ is the non-negative function of $\rho_{t1}$ and $\rho_t^*$ (e.g. the divergence between $\rho_{t1}$ and $\rho_t^*$), $g(m)$ is the term that cannot be optimized by meta learning. We then set $V^2 = \min_{\rho \in \mathcal{P}(\Theta)} \frac{1}{T}\sum_{t=1}^{T} V(\rho, \rho_t^*)^2$ as the task similarity among $T$ tasks. We will also use $f_t(\rho) = V(\rho, \rho_t^*)^2$ for brevity when the context is clear. Using the scaling technique, the regret bound $U_t(\rho, \lambda)$ of the EWA algorithm run with initialization $\rho \in \mathcal{P}(\Theta)$ and step size $\lambda = v/\sqrt{mb}$ for $v > 0$ can be rewritten as an equivalent form: $U_t(\rho, v) = (v + \frac{f_t(\rho)}{v})\sqrt{mb} + g(m)$. Therefore the OWO meta learning algorithm always consists of two online sub-algorithms: one is to play action over functions $\{f_s(\rho) = V(\rho, \rho_s^*)^2\}_{s \in [t-1]}$ to determine the initialization distribution $\rho_{t1}$ for task $t$, another is to play action over functions $\{h_s(v) = v + f_s(\rho_{s1})/v\}_{s\in[t-1]}$ to determine the step size $v_t$. The process of utilizing $\{\rho_s^*\}_{s\in[t-1]}$ and $\{v_s\}_{s\in[t-1]}$ to determine the initialization distribution $\rho_{t1}$ and the step size $v_t$ for EWA algorithm on task $t$ can be considered as transferring knowledge from past tasks to the novel task. Combining the regret bounds for this two online sub-algorithms attains the following task-averaged regret bound for non-convex OWO meta learning algorithms with general functions (i.e. without boundedness, convexity or Lipschitzness assumptions of $\{\ell_{ti}\}_{t,i\geq 1}$).

**Theorem 1** *Assume that the upper regret bound for each task $t \in [T]$ has the form of $U_t(\rho, v) = (v + \frac{f_t(\rho)}{v})\sqrt{mb} + g(m)$. Assume we have a sub-algorithm that achieves $F_T(\rho)$ regret w.r.t. any $\rho \in \mathcal{P}(\Theta)$ by setting distributions $\rho_{t1}$ on $f_t(\rho) = V(\rho, \rho_t^*)^2$, and another sub-algorithm that achieves non-increasing $H_T(v)$ regret w.r.t. any $v > 0$ by playing actions $v_t > 0$ on $h_t(v) = v + \frac{f_t(\rho_{t1})}{v}$ for all $t \in [T]$. Then, running the OWO meta learning algorithm (consisting of these two sub-algorithms) with the step size $v_t/\sqrt{mb}$ and initialization $\rho_{t1}$ at each task $t$, for $\rho^* = \arg\min_{\rho\in\mathcal{P}(\Theta)}\sum_{t=1}^{T} f_t(\rho)$ the optimal initialization and $V$ the task-similarity, we get the task-averaged regret upper bound:*

$$\frac{1}{T}\sum_{t=1}^{T}\sum_{i=1}^{m}\langle\rho_{ti} - \rho_t^*, \ell_{ti}\rangle \leq \Big(\frac{H_T(V)}{T} + \min\{\frac{F_T(\rho^*)}{VT}, 2\sqrt{\frac{F_T(\rho^*)}{T}}\} + 2V\Big)\sqrt{mb} + g(m).$$

We need to mention that the task-averaged regret upper bound framework in Theorem 1 is not only suitable to analyze EWA, but also applicable to any online algorithm (e.g. Follow-The-Perturbed-Leader in (Suggala & Netrapalli, 2020, Thm 1)) with the regret bound of form $U_t(\rho, v) = (v + \frac{f_t(\rho)}{v})\sqrt{mb} + g(m)$. In Sections 4.1-4.2, we leverage different strategies to learn initialization $\rho_{t1}$ for Lipschitz and non-Lipschitz functions. For learning the step size $v_t$, we consistently use Follow-The-Leader (FTL) algorithm, as running FTL can achieve the following logarithmic regret bound.

**Proposition 1** *Assume that FTL algorithm runs on the sequence of functions $\{h_t(v) = v + \frac{f_t(\rho_{t1})}{v}\}_{t\in[T]}$ over the domain $[0, D]$, where $D^2 \geq \max_{t\in[T]} f_t(\rho_{t1})$, then we have the regret bound:*

$$\sum_{t=1}^{T} h_t(v_t) - \min_{v\in[0,D]}\sum_{t=1}^{T} h_t(v) \leq \frac{D^3}{4}\sum_{t=1}^{T}\frac{\left|1 - \sum_{s=1}^{t-1} f_t(\rho_{t1})/\sum_{s=1}^{t-1} f_s(\rho_{s1})\right|^2}{\sum_{s=1}^{t} f_s(\rho_{s1})}.$$

*Furthermore, if for all $t \in [T]$, $f_t(\rho_{t1}) \in [B^2, D^2]$ with $D \geq B > 0$, then we have the logarithmic regret upper bound:* $\sum_{t=1}^{T} h_t(v_t) - \min_{v\in[0,D]}\sum_{t=1}^{T} h_t(v) \leq \frac{D^7}{4B^6}(\log T + 1)$.

Table 1: Different task-averaged regret bounds for OWO meta learning algorithms under different assumptions of loss functions $\{\ell_{ti}\}_{t,i=1}^{T,m}$. $T$ is the number of tasks, and $m$ is the number of iterations per task. Concretely, the **task-averaged regret upper bound** $= \big(\textbf{Bound I} + \textbf{Bound II} + V\big)\sqrt{m}$, where **Bound I** is the regret upper bound for learning the initialization, **Bound II** is the regret upper bound for learning the step size, $V$ represents the task similarity among different tasks, $\alpha \in (0, 1/2)$. The explicit form of these regret bounds and task similarities are given in Table B.1 of the Appendix.

| **Existing Works** | **Task-Averaged Regret** | **Assumptions of $\ell_{ti}$** | **Bound I** | **Bound II** |
|---|---|---|---|---|
| Khodak et al. (2019) | $\frac{1}{T}\sum_{t,i=1}^{T,m} \ell_{ti}(\theta_{ti}) - \ell_{ti}(\theta_t^*)$ | Convex & Uniformly Lipschitz | $O(\frac{\log T}{T})$ | $O(\frac{\log T}{\sqrt{T}})$ |
| Balcan et al. (2021) | $\frac{1}{T}\sum_{t,i=1}^{T,m} \langle \rho_{ti}, \ell_{ti} \rangle - \ell_{ti}(\theta_t^*)$ | Bounded & Piecewise Lipschitz | $O(\frac{m^{d/2}}{T^{1/4}})$ | $O(\frac{(\log m)\log T}{\sqrt{T}})$ |
| Our Theorem 2 | $\frac{1}{T}\sum_{t,i=1}^{T,m} \langle \rho_{ti}, \ell_{ti} \rangle - \ell_{ti}(\theta_t^*)$ | Bounded & Piecewise Lipschitz | $O(\frac{1}{T^{1/2-\alpha}})$ | $O(\frac{(\log T)^{9/2}}{T})$ |
| Our Theorem 3 | $\frac{1}{T}\sum_{t,i=1}^{T,m} \langle \rho_{ti} - \rho_t^*, \ell_{ti} \rangle$ | Bounded | $O(\frac{\log T}{T})$ | $O(\frac{\log T}{T})$ |

## 4.1 REGRET BOUNDS FOR NON-CONVEX PIECEWISE LIPSCHITZ FUNCTIONS

In this section, we let $\rho : \Theta \mapsto \mathbb{R}_{\geq 0}$ be an unnormalized distribution over $\Theta$. For any loss function $\ell : \Theta \mapsto \mathbb{R}$, $\langle \rho, \ell \rangle = \int_{\Theta} \ell(\theta)\rho(\theta)\mathrm{d}\theta / \int_{\Theta} \rho(\theta)\mathrm{d}\theta$. The update rules of Exponentially Weighted Aggregation (EWA) algorithm (with initialized distribution $\rho_1$ and step size $\lambda$) for each round $i$ can be summarized as follows: (1) Set normalization factor $P_i = \int_{\Theta} \rho_i(\theta)\mathrm{d}\theta$; (2) Sample $\theta_i$ with probability $p_i(\theta_i) = \frac{\rho_i(\theta_i)}{P_i}$; (3) Suffer $\ell_i(\theta_i)$ and observe $\ell_i(\cdot)$; (4) $\forall \theta \in \Theta$, set $\rho_{i+1}(\theta) = e^{-\lambda \ell_i(\theta)}\rho_i(\theta)$. We next give the formal definition of piecewise Lipschitz functions proposed by Balcan et al. (2018).

**Definition 1** *(Piecewise Lipschitzness) The sequence of random loss functions $\{\ell_i\}_{i=1}^{m}$ is piecewise $L$-Lipschitz ($L > 0$) that are $\beta$-dispersed, if $\forall m, \forall \epsilon \geq m^{-\beta}$, in expectation over the randomness of the loss functions, we have $\mathbb{E}[\max_{\|\theta - \theta'\|_2 \leq \epsilon} |\{i \in [m] \mid \ell_i(\theta) - \ell_i(\theta') > L\|\theta - \theta'\|_2\}|] \leq \tilde{O}(\epsilon m)$.*

Piecewise Lipschitzness is a weakly Lipschitz condition, indicating that the loss functions are discontinuous in a concentrated small region. The soft-$O$ suppresses dependence on logarithmic terms. Let $\{\ell_i : \Theta \mapsto [0, M]\}_{i \in [m]}$ be a sequence of piecewise $L$-Lipschitz functions that are $\beta$-dispersed. Let $V(\rho, \theta^*)^2 = -\log\big(\int_{\mathcal{B}(\theta^*, m^{-\beta})} \rho(\theta)\mathrm{d}\theta / \int_{\Theta} \rho(\theta)\mathrm{d}\theta\big)$, where $\mathcal{B}(\theta^*, \epsilon)$ is the $\epsilon$-radius ball around the minimizer $\theta^*$. Then, adapting the analysis from (Balcan et al., 2021, Thm 2.1) to the $M$-bounded functions obtains the regret bound for EWA algorithm with the initialization $\rho_1$ and the step size $\lambda$:

$$\sum_{i=1}^{m} \mathbb{E}_{\theta \sim \rho_i} \ell_i(\theta) - \sum_{i=1}^{n} \ell_i(\theta^*) \leq \lambda M^2 m + \frac{V(\rho_1, \theta^*)^2}{\lambda} + \tilde{O}((L+1)m^{1-\beta}). \qquad (2)$$

Detailed proofs of Eq. (2) can be found in Lemma D.1. Therefore, the task-similarity is defined as $V^2 = \min_{\rho:\Theta \mapsto \mathbb{R}_{\geq 0}, \int_{\Theta} \rho(\theta)\mathrm{d}\theta = 1} -\frac{1}{T}\sum_{t=1}^{T} \log \int_{\mathcal{B}(\theta_t^*, m^{-\beta})} \rho(\theta)\mathrm{d}\theta$. At task $t$, denote $\Theta_t = \mathcal{B}(\theta_t^*, m^{-\beta})$, we need to design an algorithm to minimize the function $f_t(\rho) = -\log(\int_{\Theta_t} \rho(\theta)\mathrm{d}\theta / \int_{\Theta} \rho(\theta)\mathrm{d}\theta)$. We borrow the idea from Balcan et al. (2021) to discretize $\Theta$ and translate the minimization problem $\min_{\rho:\Theta \mapsto \mathbb{R}_{\geq 0}} f_t(\rho)$ over the set of distributions into an online convex optimization problem. Concretely, at task $t$, define the discretization $\mathcal{D}_t = \{A = \cap_{s \leq t} \Theta_s^{(\boldsymbol{\theta}_{[s]})} : \boldsymbol{\theta} \in \{0,1\}^t, \mathrm{vol}(A) > 0\}$ of $\Theta$, where $\Theta_s^{(0)} = \Theta_s, \Theta_s^{(1)} = \Theta \backslash \Theta_s$. Then we use elements of these discretization to construct non-negative vectors in $\mathbb{R}_{\geq 0}^{|\mathcal{D}_t|}$: for any unnormalized distribution $\rho : \Theta \mapsto \mathbb{R}_{\geq 0}$, let $\boldsymbol{\rho}(t) \in \mathbb{R}_{\geq 0}^{|\mathcal{D}_t|}$ denote the vector with entries $\boldsymbol{\rho}(t)_{[A]} = \int_A \rho(\theta)\mathrm{d}\theta$ for $A \in \mathcal{D}_t$. We will use $\nu$ as the uniform measure, i.e. $\boldsymbol{\nu}(t)_{[A]} = \mathrm{vol}(A)$, and use $\boldsymbol{\nu} = \boldsymbol{\nu}(T), \boldsymbol{\rho} = \boldsymbol{\rho}(T)$ for brevity. With simple calculations, it is not difficult to see that $f_t(\rho) = -\log(\int_{\Theta_t} \rho(\theta)\mathrm{d}\theta / \int_{\Theta} \rho(\theta)\mathrm{d}\theta) = -\log \langle \boldsymbol{\rho}_t^*, \hat{\boldsymbol{\rho}} \rangle$, where $\boldsymbol{\rho}_{t \, [A]}^* = \mathbf{1}_{A \subset \Theta_t}$. Notice that $\min_{\|\boldsymbol{\rho}\|_1 = 1} f_t(\boldsymbol{\rho}) = -\log \langle \boldsymbol{\rho}_t^*, \boldsymbol{\rho} \rangle$ is a convex optimization problem over the $|\mathcal{D}_T|$-dimensional simplex. Thus, we can use the optimal Follow-The-Regularized-Leader (FTRL) algorithm with the KL-divergence regularization to solve this problem. The regret bound is exhibited in Proposition 2.

---

**Algorithm 1** Non-convex OWO meta learning algorithm for bounded piecewise Lipschitz functions.

1: **Input:** step size $\eta$ for FTRL, mixture parameter $\gamma \in (0,1]$, domain upper bound $D$; initialized distribution $\rho_{11} : \Theta \mapsto \mathbb{R}_{\geq 0}$ and initialized step size $\lambda_1 = \sqrt{(D^2 - \log \gamma)/(mM^2)}$ for EWA.

2: **for** task $t \in [T]$ **do**

3:     **for** round $i \in [m]$ **do**

4:         Set $P_{ti} = \int_\Theta \rho_{ti}(\theta)\mathrm{d}\theta$ and sample $\theta_{ti}$ with probability $p_{ti}(\theta_{ti}) = \rho_{ti}(\theta_{ti})/P_{ti}$

5:         Suffer loss $\ell_{ti}(\theta_{ti})$, observe $\ell_{ti}(\cdot)$, and update $\rho_{t,i+1}(\theta) = e^{-\lambda_t \ell_{ti}(\theta)} \rho_{ti}(\theta)$     // EWA step

6:     Sample $\theta_t^*$ with probability $p_{tm}$ and obtain task-$t$ optimum $\theta_t^* \in \Theta$

7:     Set $\mathbf{1}_{\mathcal{B}(\theta_t^*, m^{-\beta})}$ to be the function that is $\mathbf{1}$ in the ball $\mathcal{B}(\theta_t^*, m^{-\beta})$ and otherwise 0

8:     Update $\rho_{t+1,1}$ to $\boldsymbol{\rho}_{t+1}(t) = \arg\min_{\|\boldsymbol{\rho}\|_1 = 1, \boldsymbol{\rho} \geq \gamma\hat{\boldsymbol{\nu}}(t)} \mathrm{KL}(\boldsymbol{\rho}\|\hat{\boldsymbol{\nu}}(t)) - \eta \sum_{s \leq t} \log \langle \boldsymbol{\rho}_s^*(t), \boldsymbol{\rho}\rangle$,

$\lambda_{t+1} = \sqrt{\dfrac{-\sum_{s \leq t} \log \langle \boldsymbol{\rho}_s^*(s), \boldsymbol{\rho}_s(s)\rangle}{tmM^2}}$     // meta-update step

---

**Proposition 2** *Assume that FTRL algorithm with initialization $\hat{\boldsymbol{\nu}}$ and regularization $\mathrm{KL}(\boldsymbol{\rho}\|\hat{\boldsymbol{\nu}})$ runs on functions $\{f_t(\boldsymbol{\rho}_t)\}_{t \in [T]}$ over the set $\Delta = \{\boldsymbol{\rho} : \|\boldsymbol{\rho}\|_1 = 1, \boldsymbol{\rho} \geq \gamma\hat{\boldsymbol{\nu}}\}$, then $\sum_{t=1}^T f_t(\boldsymbol{\rho}_t) - f_t(\boldsymbol{\rho}^*) \leq 2GL\sqrt{T}/\gamma$, where $\boldsymbol{\rho}^* \in \arg\min_{\boldsymbol{\rho} \in \Delta} \sum_{t=1}^T f_t(\boldsymbol{\rho})$, $G^2 \geq \mathrm{KL}(\boldsymbol{\rho}^*\|\hat{\boldsymbol{\nu}})$, $L^2 = \frac{1}{T} \sum_{t=1}^T \frac{\mathrm{vol}(\Theta)^2}{\mathrm{vol}(\Theta_t)^2}$.*

Notice that $\mathrm{vol}(\Theta_t) = V_d m^{-\beta d}$, where $V_d = \frac{\pi^{d/2}}{\Gamma(d/2+1)}$ is the volume of the unit ball in the $d$-dimensional Euclidean space, $\Gamma(x+1) = x\Gamma(x)$ is the Gamma function. Then, apply Proposition 1 to learn the step size $v_t$ in the piecewise Lipschitz setting and choose a proper $\gamma$, we attain the regret.

**Proposition 3** *Assume that FTL algorithm runs on the sequence of functions $\{h_t(v) = v + \frac{f_t(\rho_{t1})}{v}\}_{t \in [T]}$ with $f_t(\rho_{t1}) = -\log \langle \boldsymbol{\rho}_t^*, \boldsymbol{\rho}_{t1}\rangle$ over the domain $[0, D]$, where $D^2 = \max_{t \in [T]} \log \frac{\mathrm{vol}(\Theta)}{\gamma \mathrm{vol}(\Theta_t)} \geq f_t(\rho_{t1})$, for any $t \in [T]$. Denote $B^2 = \min_{t \in [T]} \log \frac{\mathrm{vol}(\Theta)}{\gamma \mathrm{vol}(\Theta_t) + (1-\gamma)\mathrm{vol}(\Theta)}$. Then if we set $\gamma = m^{d\beta}/T^\alpha \in (0, 1]$ with $\alpha \in (0, 1/2)$, we have the following regret upper bound:*

$$\sum_{t=1}^T h_t(v_t) - \min_{v \in [0,D]} \sum_{t=1}^T h_t(v) \leq \frac{(\alpha \log T + \log(\mathrm{vol}(\Theta)/V_d))^{7/2}(\log T + 1)}{4B^6}.$$

For large enough $T$, if we set $\alpha \in [\ln m^{d\beta}/\ln T, \ln(\frac{em^{d\beta}}{e-1} - \frac{V_d}{(e-1)\mathrm{vol}(\Theta)})/\ln T] \subset (0, \frac{1}{2})$, we have $B^2 \geq 1$, and the regret bound in Proposition 3 has the order of $O((\log T)^{9/2})$ (see more explanations in Remark D.1 of the Appendix). We list the whole pseudo code of the OWO meta learning algorithm for piecewise Lipschitz functions $\{\ell_{ti}\}_{t, i \geq 1}$ in Algorithm 1, and achieve our first improved regret bound for this OWO meta learning algorithm by combining regret bounds in Propositions 2-3.

**Theorem 2** *Under the conditions of Theorem 1, for any task $t \in [T]$, let $\{\ell_{ti} : \Theta \mapsto [0, M]\}_{i \in [m]}$ be a sequence of piecewise $L$-Lipschitz functions that are $\beta$-dispersed. Set $\alpha \in (0, \frac{1}{2})$, let $g(m) = \tilde{O}((L+1)m^{1-\beta})$, then using FTRL algorithm in Proposition 2 and FTL algorithm in Proposition 3 respectively to learn the initialization and step size in Algorithm 1 obtains the regret upper bound:*

$$\bar{R}_{T,m} \leq g(m) + \Big\{ \frac{(\alpha \log T + \log(\mathrm{vol}(\Theta)/V_d))^{7/2}(\log T + 1)}{4TB^6}$$

$$+ \min \Big\{ \frac{\mathrm{vol}(\Theta)(\mathrm{KL}(\boldsymbol{\rho}^*\|\hat{\boldsymbol{\nu}}) + 1)}{V_d V T^{\frac{1}{2} - \alpha}}, 2\sqrt{\frac{\mathrm{vol}(\Theta)(\mathrm{KL}(\boldsymbol{\rho}^*\|\hat{\boldsymbol{\nu}}) + 1)}{V_d T^{\frac{1}{2} - \alpha}}} \Big\} + 2V \Big\} \sqrt{m}M.$$

**Remark 1** *Our task-averaged regret bound $\bar{U}_{T,m}$ in Theorem 2 has 3 improvements over the regret bound in (Balcan et al., 2021, Thm 3.3) that is obtained under the same assumptions (see Table 1 for details): (1) Our bound guarantees a vanishing task-averaged regret. For any fixed $T$, our $\bar{U}_{T,m}$ is a vanishing bound w.r.t. $m$ (i.e. $\bar{U}_{T,m} = o(m)$). However, in (Balcan et al., 2021, Thm 3.3) the bound $\bar{U}_{T,m} = O((m^{d/2}/T^{1/4} + (\log m)\log T/\sqrt{T} + V)\sqrt{m})$ is not a vanishing bound w.r.t. $m$ since the action space dimension $d \geq 1$. (2) Our regret bound of $O(1/T^{1/2-\alpha})$ ($\alpha \in (0, 1/2)$) for learning the initialization distribution in each task is sharper than that of $O(m^{d/2}/T^{1/4})$ in Balcan et al. (2021). (3) Our regret bound of $O((\log T)^{9/2}/T)$ for learning the step size has a better convergence rate than that of $O((\log m)\log T/\sqrt{T})$ in Balcan et al. (2021). More comparisons between our regret bound and that in Balcan et al. (2021) can be found in Table B.1 of the Appendix.*

---

**Algorithm 2** Non-convex OWO meta learning algorithm for bounded non-Lipschitz functions.

1: **Input:** initialized distribution $\rho_{11} \in \mathcal{P}(\Theta)$ and learning rate $\lambda_1 > 0$.
2: **for** task $t \in [T]$ **do**
3:     **for** round $i \in [m]$ **do**
4:         $\rho_{ti} = \arg\min_{\rho \in \mathcal{P}(\Theta)} \mathrm{KL}(\rho || \rho_{t1}) + \lambda_t \sum_{j=1}^{i-1} \langle \ell_{tj}, \rho \rangle$              // EWA step
5:         Suffer loss $\langle \rho_{ti}, \ell_{ti} \rangle$ and observe $\ell_{ti}(\cdot)$
6:     Update $\rho_{t+1,1} = \frac{1}{t} \sum_{s=1}^{t} \rho_{sm}, \lambda_{t+1} = \sqrt{\frac{\sum_{s=1}^{t} \mathrm{KL}(\rho_{sm} || \rho_{s1})}{tmM^2}}$          // meta-update step

---

## 4.2 REGRET BOUNDS FOR NON-CONVEX NON-LIPSCHITZ FUNCTIONS

In this section, we will describe the problem with the language of measure theory. For probability distributions $\rho, \pi \in \mathcal{P}(\Theta)$, define the Radon-Nikodym (RN) derivative of $\rho$ w.r.t. $\pi$ as $\frac{d\rho}{d\pi}$, when $\rho$ is absolutely continuous w.r.t. $\pi$ (i.e. $\rho \ll \pi$). If $\rho \not\ll \pi$, we simply set $\frac{d\rho}{d\pi} = +\infty$. We refer readers to Page 247 in Yeh (2014) for more properties of RN derivative. For distribution $\rho, \pi$, the KL-divergence between them is defined as $\mathrm{KL}(\rho || \pi) = \mathbb{E}_{\theta \sim \rho} \log \frac{d\rho}{d\pi}$, and the $\chi^2$-divergence is $\chi^2(\rho || \pi) = \mathbb{E}_{\theta \sim \pi} [(\frac{d\rho}{d\pi})^2 - 1]$. Then the update rule at $i$-th round of exponentially weighted aggregation (EWA) algorithm (with initialization distribution $\rho_1$ and step size $\lambda$) can be summarized as $\rho_i = \arg\min_{\rho \in \mathcal{P}(\Theta)} \sum_{j=1}^{i-1} \langle \rho, \ell_j \rangle + \mathrm{KL}(\rho || \rho_1)/\lambda$, and the posterior $\rho_i$ has an analytic form: $\rho_i(d\theta) = \exp\{-\lambda \sum_{j=1}^{i-1} \ell_j(\theta)\} \rho_1(d\theta) / \int \exp\{-\lambda \sum_{j=1}^{i-1} \ell_j(\theta)\} \rho_1(d\theta)$. According to (Alquier, 2021, Thm 2.1), for any loss function $\ell_i : \Theta \mapsto [0, M]$, EWA has the following regret upper bound:

$$\sum_{i=1}^{m} \langle \rho_i - \rho, \ell_i \rangle \leq \lambda M^2 m + \frac{\mathrm{KL}(\rho || \rho_1)}{\lambda}, \quad \forall \rho \in \mathcal{P}(\Theta). \tag{3}$$

Therefore, the task similarity $V^2 = \min_{\rho \in \mathcal{P}(\Theta)} \frac{1}{T} \sum_{t=1}^{T} \mathrm{KL}(\rho_t^* || \rho)$. Then, we need to choose online algorithms to learn respectively the initialization $\rho_{t1}$ and step size $v_t$ in Eq. (3) of EWA on task $t$. For learning $\rho_{t1}$, we choose FTL algorithm over functions $\{\mathrm{KL}(\rho_t^* || \rho_{t1})\}_{t \geq 1}$ to yield the regret bound.

**Proposition 4** *Given a sequence of distributions $\{\rho_t^*\}_{t \in [T]}$, assume that FTL algorithm runs on the sequence of $\{\mathrm{KL}(\rho_t^* || \rho)\}_{t \in [T]}$ to determine $\rho$, i.e. $\rho_{t1} = \arg\min_{\rho \in \mathcal{P}(\Theta)} \sum_{s=1}^{t-1} \mathrm{KL}(\rho_s^* || \rho)$, and further assume $G^2 \geq \max_{t \in [T]} \chi^2(\rho_t^* || \rho_{t1})$, then we can obtain the following regret upper bound:*

$$\sum_{t=1}^{T} \mathrm{KL}(\rho_t^* || \rho_{t1}) - \min_{\rho \in \mathcal{P}(\Theta)} \sum_{t=1}^{T} \mathrm{KL}(\rho_t^* || \rho) \leq \sum_{t=1}^{T} \frac{\chi^2(\rho_t^* || \rho_{t1})}{t} \leq G^2 (\log T + 1).$$

The above logarithmic regret bound is non-trivial, since $\mathrm{KL}(\rho_t^* || \rho_{t1})$ is the *functional* of distribution $\rho_{t1}$, and it is hard to find a norm in the space of distributions $\rho_{t1}$ to verify the strong-convexity or Lipschitzness of functional $\mathrm{KL}(\rho_t^* || \rho_{t1})$. Thus, we are unable to utilize regret analysis for the functions over Euclidean space (Shalev-Shwartz, 2012). Instead, we use the analytic form of the optimal distribution of FTL and the properties of RN derivative to get this logarithmic regret (see Section E). Then, applying Proposition 1 to learn the step size achieves the following regret bound.

**Proposition 5** *Given a sequence of functions $\{f_t(\rho_{t1}) = \mathrm{KL}(\rho_t^* || \rho_{t1})\}_{t \in [T]}$, assume that there exist $D^2 \geq \max_{t \in [T]} \mathrm{KL}(\rho_t^* || \rho_{t1}), B^2 \leq \min_{t \in [T]} \mathrm{KL}(\rho_t^* || \rho_{t1})$ with $B > 0$. Assume that FTL algorithm runs on the sequence of functions $\{h_t(v) = v + \frac{f_t(\rho_{t1})}{v}\}_{t \in [T]}$ over the domain $[0, D]$. Then we have*

$$\sum_{t=1}^{T} h_t(v_t) - \min_{v \in [0, D]} \sum_{t=1}^{T} h_t(v) \leq \frac{D^7}{4B^6} (\log T + 1).$$

If we use EWA algorithm to learn distribution $\rho_t^*$ (i.e. set $\rho_t^* = \rho_{tm}$) for task $t$, then the above condition $B > 0$ truly holds (see proof at the end of Section E). The pseudo code of OWO meta learning algorithm with $\rho_t^* = \rho_{tm}$ is listed in Algorithm 2 for non-Lipschitz functions. The corresponding task-averaged regret upper bound is given as follow by combining results in Propositions 4-5, achieving the fastest convergence rate $O((\log T/T + V)\sqrt{m})$ with respect to $T$ in the current work.

**Theorem 3** *Under the conditions of Theorem 1, for any task $t \in [T]$, let $\{\ell_{ti} : \Theta \mapsto [0, M]\}_{i \in [m]}$ be a sequence of $M$-bounded functions. Let $G^2 \geq \max_{t \in [T]} \chi^2(\rho_t^* || \rho_{t1}), D^2 \geq \max_{t \in [T]} \mathrm{KL}(\rho_t^* || \rho_{t1})$*

*and* $\min_{t\in[T]}\mathrm{KL}(\rho_t^*||\rho_{t1}) \geq B^2 > 0$, *then using FTL algorithm to respectively learn the initialization and step size of EWA algorithm in Algorithm 2 attains the task-averaged regret upper bound:*

$$\frac{1}{T}\sum_{t=1}^{T}\sum_{i=1}^{m}\langle\rho_{ti}-\rho_t^*,\ell_{ti}\rangle \leq \Big(\frac{D^7(\log T+1)}{4TB^6}+\min\Big\{\frac{G^2(\log T+1)}{VT},2G\sqrt{\frac{\log T+1}{T}}\Big\}+2V\Big)\sqrt{m}M.$$

**Remark 2** *Let's compare the regret bound in Theorem 3 (for bounded functions) with the bound in Theorem 2 (for bounded piecewise Lipschitz functions) and the bound in (Khodak et al., 2019, Thm 3.2) (for convex and Lipschitz functions) in Table 1. **(1)** In Theorem 3, the regret bounds for learning the initialization and the step size are $O(\log T/T)$, both of which are sharper than the corresponding regret bounds of $O(1/T^{1/2-\alpha})$ and $O((\log T)^{9/2}/T)$ in Theorem 2. Besides, Theorem 3 obtains improved regret bounds without piecewise Lipschitz assumption of loss functions. **(2)** When compared with Khodak et al. (2019), both our regret bound and theirs for learning the initialization have the same order of $O(\log T/T)$; but our bound for learning the step size is of $O(\log T/T)$, sharper than theirs of $O(\log T/\sqrt{T})$. The improvement is obtained by using primal-dual technique from Shalev-Shwartz & Kakade (2008) to analyze the strong-convexity of functions $\{h_t(v) = v + f_t(\rho_{t1})/v\}_{t\in[T]}$. Besides, we achieve the improved result for bounded functions, without convexity or Lipschitzness assumptions of loss functions $\{\ell_{ti}\}_{t\in[T],i\in[m]}$. We further discuss the limitations of our Theorem 3 and our Algorithm 2 in Remark B.1 and Remark H.1, respectively.*

## 5 GENERALIZATION BOUNDS FOR META LEARNING VIA REGRET ANALYSIS

In this section, we show how to derive generalization bound for statistical meta learning via regret analysis. Concretely, in Section 5.1, we provide a novel transfer risk bound for non-convex batch meta learning under the task distribution assumption. In Section 5.2, we yield a PAC-Bayes generalization bound for statistical multi-task learning that supposes independence between tasks.

### 5.1 IMPROVED TRANSFER RISK BOUNDS FOR STATISTICAL META LEARNING

In statistical meta learning, let $\tau$ be a probability measure over the set of all data distributions $\mu$ on bounded loss functions $\ell : \Theta \mapsto [0, M]$. A sequence of loss function $\{\ell_{ti}\}_{t\in[T],i\in[m]}$ is generated by drawing $m$ loss functions i.i.d. from each in a sequence of distributions $\{\mu_t\}_{t\in[T]}$, where each $\mu_t$ is regarded as a random variable and is drawn i.i.d. from $\tau$. We use the cumulative information $\{\rho_{t1},\lambda_t\}_{t\in[T]}$ from previous $T$ training tasks in OWO meta learning to run an online learning algorithm EWA on the novel task with loss functions $\{\ell_i\}_{i\in[m]}$ generated i.i.d. from distribution $\mu$ that is drawn i.i.d. from $\tau$, to output a sequence of probability distributions $\{\rho_i\}_{i\in[m]}$. Using online-to-batch arguments, we achieve the transfer risk bound for non-convex batch meta learning.

**Theorem 4** *Assume that for each task $t \in [T]$, there exist $G^2 \geq \max_{t\in[T]}\chi^2(\rho_t^*||\rho_{t1})$, and $\min_{t\in[T]}\mathrm{KL}(\rho_t^*||\rho_{t1}) \geq B^2 > 0$. Assume that the novel task consists of loss functions $\{\ell_i\}_{i\in[m]} \overset{i.i.d.}{\sim} \mu$, $\mu \overset{i.i.d.}{\sim} \tau$, and for any optimal distribution $\rho^*$ over task $\mu$, there exists $H > 0$ such that $\mathrm{KL}(\rho^*||\frac{1}{T}\sum_{t=1}^{T}\rho_{t1}) \leq H$. Then we use $(\frac{1}{T}\sum_{t=1}^{T}\rho_{t1}, \sqrt{\sum_{t=1}^{T}\mathrm{KL}(\rho_t^*||\rho_{t1})/(TmM^2)})$ to run EWA algorithm for novel task $\mu \sim \tau$ with loss functions $\{\ell_i\}_{i\in[m]}$ to output probability distributions $\{\rho_i\}_{i\in[m]}$. Then let $\bar{\rho} = \frac{1}{m}\sum_{i=1}^{m}\rho_i$, for the optimal distribution $\rho^*$ over task $\mu$ that does not dependent on $\{\ell_i\}_{i\in[m]}$, with probability $1 - \delta$ over the draw of probability distributions $\{\mu_t\}_{t=1}^{T}$:*

$$\mathbb{E}_{\mu\sim\tau}\mathbb{E}_{\{\ell_i\}_{i=1}^m\sim\mu^m}\mathbb{E}_{\ell\sim\mu}\mathbb{E}_{\theta\sim\bar{\rho}}\ell(\theta) \leq \mathbb{E}_{\mu\sim\tau}\mathbb{E}_{\ell\sim\mu}\mathbb{E}_{\theta\sim\rho^*}\ell(\theta) + M\Big(\sqrt{\frac{4G^2\log T}{Tm}}+\frac{3V^2}{\sqrt{m}B}+\frac{H}{B}\sqrt{\frac{\log 1/\delta}{2Tm}}\Big).$$

**Remark 3** *(Comparisons among different transfer risk bounds) **(1)** The first group of bounds holds for meta learning algorithms that learn a shared representation across tasks: The bound in (Alquier et al., 2017, Thm 6.1) is $O(1/\sqrt{m} + 1/\sqrt{T})$. The bound in (Denevi et al., 2019, Cor 42) is $O(\sqrt{\log m/m} + 1/\sqrt{T})$. **(2)** The second group holds for algorithms that learn the initialization and step size for each task: The bound in (Khodak et al., 2019, Thm 5.1) is $O(\log T/(m\sqrt{T}) + 1/\sqrt{m} + 1/\sqrt{Tm})$. The bound in Theorem 4 $O(\sqrt{\log T/(Tm)} + 1/\sqrt{m} + 1/\sqrt{Tm})$ is slightly larger, because our work focuses on non-convex setting and $\sum_t f_t(\rho)/v$ is not convex w.r.t. $(\rho, v)$. However, (Khodak et al., 2019) focuses on convex loss and $\sum_t \|\theta - \theta_t^*\|_2^2/v$ is convex w.r.t. $(\theta, v)$, hence able to use Jensen inequality to get a slightly better transfer risk bound (see their Thm E.1).*

## 5.2 PAC-Bayes Generalization Bounds for Statistical Multi-Task Learning

In statistical multi-task learning, each task $t$ has the training dataset $S_t = \{z_{ti}\}_{i=1}^m$, where $z_{ti}$ is drawn i.i.d. from the sample space $\mathcal{Z}$ according to the data distribution $\mu_t$. We also so suppose the independence between datasets $S_i$ and $S_j$ $(i \neq j)$ drawn from different tasks. We use $\mathcal{F}_{ti} = \sigma(\{z_{sj}\}_{1 \leq s \leq t, 1 \leq j \leq i})$ to denote the $\sigma$-algebra induced by the sequence of random variables up until the end of $i$-round at $t$-th task. Formally, an online learning algorithm $\Pi_{tm} = \{\rho_{ti}\}_{i=1}^m$ is the set of $m$ probability distributions over the action space $\Theta$. For convenience, we abbreviate $\rho_{ti} = \rho_{ti}(\{z_{sj}\}_{1 \leq s \leq t-1, 1 \leq j \leq m} \cup \{z_{tj}\}_{1 \leq j \leq i-1})$. A random variable associated to the online learning algorithm $\Pi_{tm}$ is $M_{\Pi_{tm}} = \frac{1}{m} \sum_{i=1}^m \langle \rho_{ti}, c_{ti} \rangle$, where $c_{ti}(\theta) = \mathbb{E}_{z \sim \mu_t}[\ell(\theta, z)|\theta] - 1/m \sum_{i=1}^m \ell(\theta, z_{ti})$. Note that $M_{\Pi_{tm}}$ is a sum of normalized martingale differences since $\mathbb{E}[\langle \rho_{ti}, c_{ti} \rangle | \mathcal{F}_{t,i-1}] = 0$. The generalization error of the action $\theta \in \Theta$ is defined as: $\text{gen}(\theta, S_t) = \mathbb{E}_{z \sim \mu_t}[\ell(\theta, z)|\theta] - 1/m \sum_{i=1}^m \ell(\theta, z_{ti})$. In PAC-Bayes learning, for each task $t$, the algorithm $\mathcal{A}_t$ takes the training sample $S_t$ and a prior as input and outputs a posterior $\mathcal{A}_t(S_t) \in \mathcal{P}(\Theta)$. The generalization error of posterior $\mathcal{A}_t(S_t)$ is defined as $\overline{\text{gen}}(\mathcal{A}_t, S_t) = \mathbb{E}_{\theta \sim \mathcal{A}_t(S_t)} \text{gen}(\theta, S_t)$. Then we obtain the following proposition that connects the multi-task generalization error of posteriors $\{\mathcal{A}_t(S_t)\}_{t \geq 1}$ in statistical multi-task learning and the task-averaged regret in OWO meta learning.

**Proposition 6** *Let $\bar{R}_{T,m} = \sum_{t,i=1}^{T,m} \langle \rho_{ti} - \mathcal{A}_t(S_t), \frac{\ell_{ti}}{T} \rangle$, then $\sum_{t=1}^T \overline{\text{gen}}(\mathcal{A}_t, S_t) = \frac{\bar{R}_{T,m}}{m} - \frac{1}{T} \sum_{t=1}^T M_{\Pi_{tm}}$.*

Using $\bar{U}_{T,m}$ in our Theorem 3 to upper bound the task-averaged regret $\bar{R}_{T,m}$, and applying simple concentration inequality to upper bound the sum of martingale differences $\sum_t M_{\Pi_{tm}}$, we obtain for statistical multi-task learning a novel PAC-Bayes bound that does not appear in existing literature.

**Theorem 5** *Let $\mathcal{A}_t$ be the statistical learning algorithm for task $t \in [T]$. Assume that for all actions $\theta \in \Theta$, samples $z \in \mathcal{Z}$, $\ell(\theta, z) \in [0, M]$. Then, with probability at least $1 - \delta$ over the draw of $\{S_t\}_{t \in [T]}$, the multi-task generalization error of statistical learning algorithms $\{\mathcal{A}_t\}_{t \in [T]}$ satisfies:*

$$\frac{1}{T} \sum_{t=1}^T \mathbb{E}_{\theta \sim \mathcal{A}_t(S_t)} \Big[ \mathbb{E}_{z \sim \mu_t} \ell(\theta, z) - \frac{1}{m} \sum_{i=1}^T \ell(\theta, z_{ti}) \Big] \leq \frac{\bar{U}_{T,m}}{m} + M \sqrt{\frac{2 \log \frac{1}{\delta}}{Tm}}.$$

**Remark 4** *We give two insights into the PAC-Bayes bound in our Theorem 5. **(1)** The most related work to our bound is the PAC-Bayes bound for multi-task generalization error in PAC-Bayes meta learning theory (in the batch setting), which assumes that the priors for each task are the same and are sampled from the hyper-posterior $\mathcal{Q}$. The tightest PAC-Bayes bound in this field is from (Guan & Lu, 2022, Prop 3) of $O(\{\mathbb{E}_{P \sim \mathcal{Q}} \sum_{t=1}^T \text{KL}(\mathcal{A}_t(S_t)||P) + \ln(Tm)\}/(Tm)) \approx O(1/m + \ln(Tm)/(Tm))$. Such PAC-Bayes bound for batch multi-task learning is sharper than our bound of $O(V/\sqrt{m} + 1/\sqrt{Tm})$ in our Theorem 5 for statistical multi-task learning, indicating the difficulty of online meta learning w.r.t. batch meta learning. **(2)** Our Proposition 6 for the first time reveals the connection between multi-task generalization error in PAC-Bayes theory and the task-averaged regret in non-convex OWO meta learning. This gives a promising direction of proving PAC-Bayes multi-task generalization bounds by applying regret analysis and simple concentration inequality.*

## 6 Conclusion

We study the non-convex online-within-online (OWO) meta learning of the initialization and step size of exponentially weighted aggregation (EWA) algorithm. We extend the averaged regret upper bound analysis to the non-convex setting, and typically propose to learn the step size with Follow-The-Leader (FTL) algorithm to guarantee the logarithmic regret. For learning the initialization, we develop two algorithms based on the type of loss functions. For piecewise Lipschitz functions, we choose Follow-The-Regularized-Leader algorithm to learn the discrete initialization distribution and achieve a sublinear regret. For non-Lipschitz functions, we utilize FTL algorithm to learn the continuous initialization distribution and derive a logarithmic regret. Both strategies lead to improved regret bound for non-convex OWO meta learning. Furthermore, by online-to-batch arguments, we yield a new transfer risk bound for batch meta learning. By online-to-PAC techniques, we achieve a novel PAC-Bayes generalization bound for statistical multi-task learning, revealing a promising framework of proving PAC-Bayes bounds for multi-task setting via regret analysis. Our ongoing research includes exploring the OWO meta learning of other online algorithms (e.g. Follow-The-Perturbed-Leader), and investigating whether we can obtain its optimal task-averaged regret bound.

## 7 ACKNOWLEDGEMENT

We would like to thank all reviewers for their constructive suggestions to improve the quality of this paper. This work was partially supported by National Natural Science Foundation of China (Grant No.92370204), Guangzhou-HKUST(GZ) Joint Funding Program (Grant No.2023A03J0008), Science and Technology Planning Project of Guangdong Province(Grant No.2023A0505050111), Education Bureau of Guangzhou Municipality, and Guangdong Science and Technology Department.

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

# APPENDIX

## A  ADDITIONAL RELATED WORK

**Online Convex Optimization**. In online convex optimization (OCO) setting where the loss functions $\{\ell_i\}_{i \in [m]}$ are convex, the regret is defined as $R_m \triangleq \sum_{i=1}^m \ell_i(\theta_i) - \ell_i(\theta^*)$, where $\theta^* = \arg\min_{\theta \in \Theta} \sum_{i=1}^m \ell_i(\theta)$. The upper bound $U_m \geq R_m$ on the regret is always called regret bound. The goal of online learner is to choose a sequence of actions $\{\theta_i\}_{i \in [m]}$ that will guarantee the vanishing regret, i.e. $R_m = o(m)$ or $U_m = o(m)$. In OCO problem, two famous algorithms are Follow-The-Leader (FTL) and Follow-The-Regularized-Leader (FTRL). At $i$-th round, FTL chooses the action $\theta_i = \arg\min_{\theta \in \Theta} \sum_{j=1}^{i-1} \ell_j(\theta)$; and FTRL selects the action $\theta_i = \arg\min_{\theta \in \Theta} \sum_{j=1}^{i-1} \ell_j(\theta) + R(\theta)$, where $R : \Theta \mapsto \mathbb{R}_{\geq 0}$ is always a strongly-convex regularization function like $\ell_2$-norm. When $\{\ell_i\}_{i \in [m]}$ are convex and Lipschitz, FTRL achieves optimal regret of $O(\sqrt{m})$ (see (Shalev-Shwartz, 2012, Thm 2.11) for upper bound and (Hazan, 2023, Thm 3.2) for lower bound). When $\{\ell_i\}_{i \in [m]}$ are strongly-convex and Lipschitz, FTL achieves optimal regret of $O(\log m)$ (see (Shalev-Shwartz & Kakade, 2008, Cor 1) for upper bound and (Takimoto & Warmuth, 2000, Thm 4) for lower bound). We introduce FTRL and FTL in detail because these two algorithms are utilized as the main components of our OWO meta learning algorithm in Section 4.

**Online Non-Convex Optimization**. However, in online non-convex optimization setting where the loss functions $\{\ell_i\}_{i \in [m]}$ are non-convex, minimizing regret is more difficult than that in convex case. It has been demonstrated that all deterministic online algorithms (e.g. the aforementioned FTL and FTRL for OCO) cannot obtain vanishing regret in the non-convex setting (see (Suggala & Netrapalli, 2020, Prop 3) and (Cesa-Bianchi & Lugosi, 2006, Sect 4.1) for details). Therefore, at $i$-th round we need to add randomness into the algorithm, and choose the action drawn from certain probability distribution $\rho_i \in \mathcal{P}(\Theta)$. The corresponding regret is defined as $R_m \triangleq \sum_{i=1}^m \mathbb{E}_{\theta \sim \rho_i} \ell_i(\theta) - \ell_i(\theta^*)$, where $\theta^* \in \arg\min_{\theta \in \Theta} \sum_{i=1}^m \ell_i(\theta)$. If $\{\ell_i\}_{i \in [m]}$ are Lipschitz, using Follow-The-Perturbed-Leader algorithm allows to achieve the optimal regret $O(\sqrt{m})$ (see (Suggala & Netrapalli, 2020, Thm 1)).

**Generalization Bounds for Statistical Multi-Task Learning**. Statistical multi-task learning refers to the multi-task setting where data within each task are assumed to be independently sampled from the same distribution. The investigation of generalization bounds for statistical multi-task learning has a long history and dates back to (Baxter, 1995). In the last decades, different generalization bounds are proposed for multi-task learning, mostly based on model-capacity theory: for example, the generalization bound based on covering number complexity (Ando & Zhang, 2005), the bounds based on the VC-dimension of hypothesis space (Baxter, 2000; Ben-David & Borbely, 2008), the bound based on Gaussian complexity (Maurer et al., 2016). Apart from them, there also exist generalization bounds for multi-task learning that are derived based on algorithmic stability analysis, like (Maurer, 2005; Liu et al., 2017). Recent works also use localized Rademacher complexity analysis to obtain improved generalization bounds (Yousefi et al., 2018). However, little literature use PAC-Bayes theory to prove bounds for multi-task learning, except for the PAC-Bayes generalization bounds in statistical meta leaning that involves the multi-task generalization bounds (Pentina & Lampert, 2014; Amit & Meir, 2018; Guan & Lu, 2022). But the PAC-Bayes multi-task generalization bounds in meta learning theory assume the priors for different training tasks are the same and are sampled from the same hyper-prior distribution. Therefore, to the best of our knowledge, there is still no explicit PAC-Bayes generalization bound for standard statistical multi-task learning setting.

## B  EXPLICIT FORM OF REGRET BOUNDS FOR OWO META LEARNING

**Remark B.1** (*Limitations of the regret bound in Theorem 3*) *For any task $t \in [T]$, if we set the distribution $\rho_t^*$ as Dirac measure $\delta_{\theta_t^*}$ that only has mass $1$ at the minimizer $\theta_t^*$, the regret in Theorem 3 degenerates to the regret defined in Eq. (1) (i.e. the regret in our Theorem 2). To let $\mathrm{KL}(\rho_t^* \| \rho_{t1})$ (where $\rho_{t1} = 1/(t-1) \sum_{s=1}^{t-1} \rho_s^*$) make sense, there should exist at least one $s \in [t-1]$, such that $\rho_s^*(\{\theta_t^*\}) > 0$ (otherwise the RN derivative $\mathrm{d}\rho_t^*/\mathrm{d}(\sum_{s=1}^{t-1} \rho_s^*) = \infty$). However, the optimal actions for the past tasks may not be the optimal one for future tasks. Hence the regret bound in our Theorem 3 may be vacuous under the regret definition in Eq. (1). This also indicates to some extent the limitation of the regret (defined as the gap between the cumulative loss w.r.t. $\{\rho_{ti}\}_{i=1}^m$ and the loss w.r.t. the optimal distribution $\rho_t^*$) and the $f$-divergence based regret upper bounds (which, including the $f$-divergence between $\rho_t^*$ and the initial distribution $\rho_{t1}$, will become vacuous if we set $\rho_t^*$ as a Dirac measure. See more details in (Alquier, 2021, Thm 2.1)) for EWA-type algorithm.*

Table B.1: Different task-averaged regret bounds for OWO meta learning algorithms under difference assumptions of loss functions $\{\ell_{ti}\}_{t\in[T],i\in[m]}$. In these bounds, $T$ is the number of tasks, and $m$ is the number of iterations per task. Concretely, the **task-averaged regret upper bound** = (**Bound I** + **Bound II** + $V)\sqrt{m} + g(m)$, where **Bound I** is the regret upper bound for learning the step size, $V$ represents the task similarity among $T$ different tasks. For all $t \in [T]$, $\mathrm{vol}(\Theta_t) = V_d m^{-d\beta}$. In (Khodak et al., 2019, Thm 3.3) and our Theorem 3, $g(m) = 0$. In (Balcan et al., 2021, Thm 3.3) and our Theorem 2, $g(m) = O(m^{1-\beta})$. We set $O(\min\{\frac{F_T(\rho^*)}{VT}, 2\sqrt{\frac{F_T(\rho^*)}{T}}\}) = O(\frac{F_T(\rho^*)}{VT}) = O(\frac{F_T(\rho^*)}{T})$ for all **Bound I** in Table 1 of the main text for brevity. In the bound of Khodak et al. (2019), $d_\phi(\theta_t^*\|\theta_{t1})$ represents the Bregman divergence between vectors $\theta_t^*$ and $\theta_{t1}$ with respect to function $\phi$.

| Existing Works | Task-Averaged Regret | Assumptions of $\{\ell_{ti}\}_{t\in[T],i\in[m]}$ | Bound I | Bound II | Task Similarity $V^2$ |
|---|---|---|---|---|---|
| Khodak et al. (2019) | $\frac{1}{T}\sum_{t,i=1}^{T,m} \ell_{ti}(\theta_{ti}) - \ell_{ti}(\theta_t^*)$ | Convex & Lipschitz | $O\left(\min\left\{\max_{t\in[T]}\|\theta_t^*\|^2\frac{\log T}{VT}, \max_{t\in[T]}\|\theta_t^*\|\sqrt{\frac{\log T}{T}}\right\}\right)$ | $O\left(\min\left\{\frac{1}{V\sqrt{T}}, \frac{1}{T^{\frac14}}\right\} + \frac{\log T}{\sqrt{T}}\left(d_\phi(\theta_t^*\|\theta_{t1})\right)^{\frac32}\right)$ | $\min_\theta \frac{1}{T}\sum_{t=1}^T d_\phi(\theta_t^*\|\theta)$ |
| Balcan et al. (2021) | $\frac{1}{T}\sum_{t,i=1}^{T,m} \langle \rho_{ti}, \ell_{ti}\rangle - \ell_{ti}(\theta_t^*)$ | Bounded & Piecewise Lipschitz | $O\left(\min\left\{\frac{\mathrm{KL}(\rho^*\|\hat\nu)^{\frac14}\mathrm{vol}(\Theta)^{\frac12}}{VT^{\frac14}\mathrm{vol}(\Theta_t)^{\frac12}}, \frac{\mathrm{KL}(\rho^*\|\hat\nu)^{\frac18}\mathrm{vol}(\Theta)^{\frac14}}{T^{\frac18}\mathrm{vol}(\Theta_t)^{\frac14}}\right\}\right)$ | $O\left(\min\left\{\frac{1}{V\sqrt{T}}, \frac{1}{T^{\frac14}}\right\} + \frac{\log T}{\sqrt{T}}\left(\log\frac{\mathrm{vol}(\Theta)}{\mathrm{vol}(\Theta_t)^{\frac34}T^{\frac14}}\mathrm{KL}(\rho^*\|\hat\nu)^{\frac12}\right)^{\frac32}\right)$ | $\min_\rho \frac{-1}{T}\sum_{t=1}^T \log\int_{\mathcal{B}(\theta_t^*,\epsilon)}\rho(\mathrm{d}\theta)$ |
| Our Theorem 2 | $\frac{1}{T}\sum_{t,i=1}^{T,m} \langle \rho_{ti}, \ell_{ti}\rangle - \ell_{ti}(\theta_t^*)$ | Bounded & Piecewise Lipschitz | $O\left(\min\left\{\frac{\mathrm{KL}(\rho^*\|\hat\nu)\mathrm{vol}(\Theta)}{VT^{\frac12-\alpha}}, \frac{\mathrm{KL}(\rho^*\|\hat\nu)^{\frac12}\mathrm{vol}(\Theta)^{\frac12}}{T^{\frac14-\frac\alpha2}}\right\}\right)$ | $O\left(\left(\frac{\log T + \log\mathrm{vol}(\Theta)}{T}\right)^{\frac27}\log T\right)$ | $\min_\rho \frac{-1}{T}\sum_{t=1}^T \log\int_{\mathcal{B}(\theta_t^*,\epsilon)}\rho(\mathrm{d}\theta)$ |
| Our Theorem 3 | $\frac{1}{T}\sum_{t,i=1}^{T,m} \langle \rho_{ti} - \rho_t^*, \ell_{ti}\rangle$ | Bounded | $O\left(\min\left\{\frac{\max_{t\in[T]}\chi^2(\rho_t^*\|\rho_{t1})\log T}{VT}, \sqrt{\max_{t\in[T]}\chi^2(\rho_t^*\|\rho_{t1})}\sqrt{\frac{\log T}{T}}\right\}\right)$ | $O\left(\frac{\log T\left(\max_{t\in[T]}\mathrm{KL}(\rho_t^*\|\rho_{t1})^{\frac72}\right)}{T\left(\min_{t\in[T]}\mathrm{KL}(\rho_t^*\|\rho_{t1})\right)^3}\right)$ | $\min_\rho \frac{1}{T}\sum_{t=1}^T \mathrm{KL}(\rho_t^*\|\rho)$ |

**Remark B.2** *(Our Three Technical Novelties in Deriving Improved Regret Bounds for OWO meta learning) (1) The first novelty lies in deriving improved regret bound for the online algorithm that runs over the functions $\{h_t(v) = v + f_t(\rho_{t1})/v\}_{t\in[T]}$ on the domain $[B^2, D^2]$ to learn the step size $v_t$ of EWA algorithm. Throughout the whole paper, we choose the efficient Follow-The-Leader (FTL) algorithm to learn the step size $v_t = \arg\min_{v\in[B^2,D^2]} \sum_{s=1}^{t-1} h_s(v) = \left(\sum_{s=1}^{t-1} f_s(\rho_{s1})/(t-1)\right)^{1/2}$ and use the primal-dual analysis from (Shalev-Shwartz & Kakade, 2008, Cor 1) to derive a logarithmic regret bound of $O(\log T)$. The key step in obtaining the logarithmic regret $O(\log T)$ is to show that $\min_{t\in[T]} f_t(\rho_{t1})$ is strictly positive (i.e. $B > 0$) to guarantee the strong-convexity of $h_t(v)$ and the boundedness of $\partial h_t(v_t)$ (i.e. the Lipschitz property of $h_t(v)$ at the point $v_t$). The positiveness of $\min_{t\in[T]} f_t(\rho_{t1})$ is guaranteed for piecewise Lipschitz functions in Proposition 3 and for non-Lipschitz functions in Proposition 5 respectively, via a fine-grained estimation of the lower bound $B^2$ of $f_t(\rho_{t1})$. In contrast, existing works (Khodak et al., 2019, Prop B.2) and (Balcan et al., 2021, Cor 3.2) both choose $\epsilon$-FTL algorithm to optimize the functions $\{h_t(v) = v + (f_t(\rho_{t1}) + \epsilon^2)/v\}_{t\in[T]}$ on the domain $[0, D^2]$ (where $D^2 \geq \max_t f_t(\rho_{t1})$) to learn the step size $v_t$, and lead to the regret $O(T\epsilon^2 + (\log T)/\epsilon^2)$, which is of $O(\sqrt{T}\log T)$ if we set $\epsilon = 1/T^{1/4}$ and is slower than our bound. (2) The second novelty lies in deriving improved regret bound for the online algorithm that runs over the functions $\{f_t(\rho) = V(\rho, \rho_t^*)\}_{t\in[T]}$ in the piecewise Lipschitz case. We use Follow-The-Regularized-Leader (FTRL) algorithm to achieve the regret of $O(T^{1/2+\alpha})$, $\alpha \in (0, \frac{1}{2})$, and the mixture parameter $\gamma = m^{d\beta}/T^\alpha$ is irrelevant to the optimal $\rho^*$ (hence $\gamma$ can be set in advance). Existing work (Balcan et al., 2021, Thm 3.2) also uses FTRL algorithm, but Balcan et al. (2021) leverage a more complicated analysis, attaining a larger regret bound $O(m^{d/2}T^{3/4})$. Nevertheless, the choice of $\gamma$ in (Balcan et al., 2021) depends on the knowledge of optimal parameter $\rho^*$ that contains information of $T$ training tasks, which is unfeasible in the sequential online meta learning setting. (3) The third novelty lies in deriving improved regret bound for the online algorithm that runs over the functions $\{f_t(\rho) = \mathrm{KL}(\rho_t^*||\rho)\}_{t\in[T]}$ in the non-Lipschitz case. Obtaining regret bounds for FTL algorithm run over the functions $\{\mathrm{KL}(\rho_t^*||\rho)\}_{t\in[T]}$ is hard, because $\mathrm{KL}(\rho_t^*||\rho)$ is the functional of the probability distribution $\rho$, and thus we are unable to use traditional regret analysis (e.g. the gradient boundedness and strong convexity analysis of the functions) for the functions on Euclidean space Shalev-Shwartz (2012). Instead, we leverage an insightful lemma from (Frigyik et al., 2008, Thm II.1) to obtain the analytic form $\rho_{t1} = \arg\min_{\rho\in\mathcal{P}(\theta)} \sum_{s=1}^{t-1} \mathrm{KL}(\rho_s^*||\rho) = \frac{1}{t-1}\sum_{s=1}^{t-1} \rho_s^*$ of the solution of FTL algorithm. Then, we use this analytic form, as well as the properties of RN derivative to estimate the upper bound of the regret and ultimately obtain a non-trivial logarithmic regret $O(\log T)$, achieving so far the tightest regret bound for learning the initialization of EWA algorithm.*

**Remark B.3** *(Comparisons between our Task-Averaged Regret Bound and the Vanilla Averaged Regret Bound for EWA Algorithm). Recall that in Theorem 3, $G^2 \geq \max_{t\in[T]} \chi^2(\rho_t^*||\rho_{t1})$, $D^2 \geq \max_{t\in[T]} \mathrm{KL}(\rho_t^*||\rho_{t1})$, $\min_{t\in[T]} \mathrm{KL}(\rho_t^*||\rho_{t1}) \geq B^2$, then our task-averaged regret upper bound is*

$$\frac{1}{T}\sum_{t=1}^{T}\sum_{i=1}^{m}\langle\rho_{ti} - \rho_t^*, \ell_{ti}\rangle \leq \left(\frac{D^7(\log T + 1)}{4TB^6} + \min\left\{\frac{G^2(\log T + 1)}{VT}, 2G\sqrt{\frac{\log T + 1}{T}}\right\} + 2V\right)\sqrt{m}M.$$

*The vanilla averaged regret bound (Alquier, 2021, Thm 2.1) for EWA algorithm (with the step size $\lambda_t = 1/(M\sqrt{m})$ and the initialization $\rho_{t1}$ for task $t$) across $T$ tasks without knowledge transfer is*

$$\frac{1}{T}\sum_{t=1}^{T}\sum_{i=1}^{m}\langle\rho_{ti} - \rho_t^*, \ell_{ti}\rangle \leq \frac{1}{T}\sum_{t=1}^{T}(\lambda_t M^2 m + \frac{\mathrm{KL}(\rho_t^*||\rho_{t1})}{\lambda_t}) = \sqrt{m}M(1 + \frac{1}{T}\sum_{t=1}^{T}\mathrm{KL}(\rho_t^*||\rho_{t1})).$$

*We can observe that: (1) If we assume the same upper bound $D^2$ on $\mathrm{KL}(\rho_t^*||\rho_{t1})$, the vanilla averaged regret bound is of $O(\sqrt{m}MD^2)$, which is independent of $T$ and could not decrease with the increase of $T$. This demonstrates that we are unable to obtain tighter regret bound for multi-task learning than that for single-task learning, if we do not share any knowledge across different tasks. (2) When $T$ is large enough, our task-averaged regret bound is of $O(\sqrt{m}MV)$. Such regret bound is typically sharper than the vanilla averaged regret bound $O(\sqrt{m}MD^2)$ when $V \ll D^2$, i.e. when different tasks share a high degree of similarity. This indicates that, when $T$ training tasks are similar enough, leveraging knowledge from previous tasks can achieve better theoretical guarantee than single-task learning, validating the advantages of OWO meta learning. (3) In our Theorem 3, the proposed task similarity notion $V^2 = \min_\rho \frac{1}{T}\sum_{t=1}^{T}\mathrm{KL}(\rho_t^*||\rho) = \frac{1}{T}\sum_{t=1}^{T}\mathrm{KL}(\rho_t^*||\frac{1}{T}\sum_{t=1}^{T}\rho_t^*)$ is actually the so-called generalized Jensen-Shannon divergence, which is always used to measure the similarity among different distributions $\{\rho_t^*\}_{t\in[T]}$ in information theory (Lin, 1991, Section V).*

**Remark B.4** *(More Discussions between our Task-Averaged Bound for OWO Meta Learning and the Task-Averaged Regret Bound obtained via Dynamic Regret Analysis). One of the anonymous Reviewers suggest we make a comparison between our task-averaged regret in Theorem 3 and the task-averaged regret obtained via dynamic regret analysis:* **(1)** *First, we need to derive a task-averaged regret bound for OWO meta learning through the lens of dynamic regret analysis. Denote $\phi_t(\theta) = \sum_{i=1}^{m} \ell_{ti}(\theta)$, then the regret for OWO meta learning in our Eq.(1) can be rewritten roughly as $\frac{1}{T}\sum_{t=1}^{T} \mathbb{E}_{\theta \sim \bar{\rho}_t} \phi_t(\theta) - \phi_t(\theta_t^*)$ (actually, we think we cannot rigorously rewrite the task-averaged regret as the equivalent form of dynamic regret, because we cannot write $\sum_{i=1}^{m} \mathbb{E}_{\theta \sim \rho_{ti}} \ell_{ti}(\theta)$ as the expectation of $\phi_t(\theta)$ over a common distribution $\rho \in \mathcal{P}(\Theta)$), where $\bar{\rho}_t = \frac{1}{m}\sum_{i=1}^{m} \rho_{ti}$. Assume that $\sum_{t=1}^{T-1} \|\phi_t - \phi_{t+1}\| \le V_T$, then according to the latest dynamic regret bound in (Gao et al., 2018, Thm 1) (to the best of our knowledge this is the latest dynamic regret bound for non-convex online learning), the task-averaged regret is bounded by $\frac{1}{T}\sum_{t=1}^{T} \mathbb{E}_{\theta \sim \bar{\rho}_t} \phi_t(\theta) - \phi_t(\theta_t^*) \le O(\frac{\sqrt{T+V_T T}}{T}) = O(\frac{\sqrt{1+V_T}}{\sqrt{T}})$.* **(2)** *Next, we compare our regret $O((\frac{\log T}{T}+V)\sqrt{m})$ and the regret $O(\frac{\sqrt{1+V_T}}{\sqrt{T}})$ obtained via dynamic regret analysis, from 3 aspects:* **(i)** *Our regret analysis does not adopt the bounded total variation assumption (i.e. $V_T$ is a bounded constant), when compared with dynamic regret analysis.* **(ii)** *Our regret bound $O((\frac{\log T}{T} + V)\sqrt{m})$ is more informative, revealing the importance of task similarity $V$ to the generalization of OWO meta leaning algorithm.* **(iii)** *Our task-averaged regret bound $O((\frac{\log T}{T} + V)\sqrt{m})$ has a faster convergence rate w.r.t. $T$ when compared with $O(\frac{\sqrt{1+V_T}}{\sqrt{T}})$.*

**Remark B.5** *(More Discussions on the Advantages of our Theorem 2 when compared with our Theorem 3) One of the anonymous Reviewers suggest we discuss more the value/advantages of the first improved regret bound when compared with the second improved regret bound. The detailed explanations between their differences lie in the following 4 aspects:* **(1)** *The first improved regret bound in our Theorem 2 is not a special case of our Theorem 3. The main reason is that Theorem 2 uses $V^2 = \min_{\rho:\Theta \mapsto \mathbb{R}_{\ge 0}, \int_{\Theta} \rho(\theta) d\theta = 1} - \frac{1}{T}\sum_{t=1}^{T} \log \int_{\mathcal{B}(\theta_t^*, m^{-\beta})} \rho(\theta) d\theta$ as the similarity notion between different tasks, but Theorem 3 uses the task similarity notion $V^2 = \min_{\rho \in \mathcal{P}(\Theta)} \frac{1}{T}\sum_{t=1}^{T} \mathrm{KL}(\rho_t^* \| \rho)$. Besides, the task similarity in Theorem 2 is defined according to the specific property (i.e. $\epsilon$-radius) of the piecewise-Lipschitz function, and hence is particularly applicable to the piecewise-Lipschitz setting.* **(2)** *The action-space-discretization technique (described in Section 4.1 to obtain the first regret bound in Theorem 2) is of independent interest. The defined task similarity in Theorem 2 also requires a novel action-space-discretization method to translate the minimization problem $\min_{\rho \in \Theta} f_t(\rho)$ over the set of distributions into a tractable online convex optimization problem.* **(3)** *At present, the first regret bound has higher application value than the second regret bound. The Algorithm 1 (corresponding to the first regret bound in our Theorem 2) can be applied in the continuum domain, but Algorithm 2 (corresponding to the second regret bound in our Theorem 3) at the current stage is still not easy to be applied in the continuum domain (see more explanations in our Remark H.1).* **(4)** *The first improved regret is obtained under the same assumptions (i.e. piecewise Lipschitzness and bounded loss functions) as that in (Balcan et al., 2021, Thm 3.3), via a more technical analysis. We list the improved regret in our paper to make a fair comparison and show rigorous improvements over existing result (Balcan et al., 2021).*

**Remark B.6** *(The Potential Improvement Space of our Theoretical Results). According to Theorem 1, we decompose the task-averaged regret bound problem into two subproblems:* **(1)** *minimizing $\{f_t(\rho) = V(\rho, \rho_t^*)^2\}_{t \in [T]}$ to learn initialization $\rho$, and* **(2)** *minimizing $\{h_t(v) = v + f_t(\rho_{t1})/v\}_{t \in [T]}$ to learn step size $v$. Combining the above two results leads to task-averaged regret bounds in our Theorems 2-3, which are actually optimal w.r.t. $m$ (i.e. of order $O(\sqrt{m})$). Therefore, what we can improve is the convergence rate w.r.t. $T$, and our explanations are three-fold:* **(1)** *For learning the initialization $\rho$, our Proposition 4 achieves a logarithmic regret $O(\log T)$. According to the related work of OCO in Appendix A, the optimal regret for strongly-convex online optimization is $O(\log T)$. Therefore, we believe that our Proposition 4 achieves the optimal regret for learning the initialization.* **(2)** *For learning the step size $v$, our Propositions 3 and 5 actually achieve the (polynomial) logarithmic regret $O(\log T)$. Therefore, we also obtain optimal or near optimal regret for learning the step size $v$.* **(3)** *Consider* **(1)** *and* **(2)**, *if we still adopt the regret upper bound decomposition framework, we should refine the proof in Theorem 1. For example, there seems to be some improvement space in the 4-th inequality in the proof of our Theorem 1, since other inequalities in this proof hold due to the definition of regret bound; If not, we should use other task-averaged regret analysis to see whether we could achieve better convergence rate w.r.t. $T$ or a smaller multiplier constant.*

# C  PROOFS OF THE TASK-AVERAGED REGRET BOUND FRAMEWORK FOR NON-CONVEX OWO META LEARNING

***Proof of Theorem 1.***

$$
\frac{1}{T}\sum_{t=1}^{T}\sum_{i=1}^{m}\langle\rho_{ti}-\rho_t^*,\ell_{ti}\rangle
$$

$$
\leq\frac{1}{T}\sum_{t=1}^{T}\left[mb\lambda_t+\frac{V(\rho_{t1},\rho_t^*)^2}{\lambda_t}+g(m)\right]=\frac{1}{T}\sum_{t=1}^{T}U_t(\rho_{t1},\lambda_t)
$$

$$
=\frac{1}{T}\sum_{t=1}^{T}\left[\sqrt{mb}\lambda_t+\frac{V(\rho_{t1},\rho_t^*)^2}{\sqrt{mb}\lambda_t}\right]\sqrt{mb}+g(m)
$$

$$
=\frac{1}{T}\sum_{t=1}^{T}\left[v_t+\frac{f_t(\rho_{t1})}{v_t}\right]\sqrt{mb}+g(m)\qquad(v_t=\sqrt{mb}\lambda_t)
$$

$$
\leq\min_{v>0}\sqrt{mb}\left[\frac{1}{T}\sum_{t=1}^{T}(v+\frac{f_t(\rho_{t1})}{v})+\frac{H_T(v)}{T}\right]+g(m)
$$

$$
=\min_{v>0}\sqrt{mb}\left[v+\frac{\sum_{t=1}^{T}f_t(\rho_{t1})}{Tv}+\frac{H_T(v)}{T}\right]+g(m)
$$

$$
\leq\min_{v>0,\rho\in\mathcal{P}(\Theta)}\sqrt{mb}\left[v+\frac{\sum_{t=1}^{T}f_t(\rho)+F_T(\rho)}{Tv}+\frac{H_T(v)}{T}\right]+g(m)
$$

$$
\leq\min_{v>0}\sqrt{mb}\left[v+\frac{\sum_{t=1}^{T}f_t(\rho^*)+F_T(\rho^*)}{Tv}+\frac{H_T(v)}{T}\right]+g(m)
$$

$$
=\min_{v>0}\sqrt{mb}\left[v+\frac{V^2}{v}+\frac{F_T(\rho^*)}{Tv}+\frac{H_T(v)}{T}\right]+g(m)
$$

$$
\leq\left(\frac{H_T(V)}{T}+\min\{\frac{F_T(\rho^*)}{VT},2\sqrt{\frac{F_T(\rho^*)}{T}}\}+2V\right)\sqrt{mb}+g(m),
$$

where the last inequality holds by taking $v=V$ or $v=\sqrt{V^2+F_T(\rho^*)/T}$. $\qquad\square$

**Lemma C.1** *(Shalev-Shwartz & Kakade, 2008, Corollary 1) Let $\ell_1,...,\ell_T$ be a sequence of functions such that for all $t\in[T]$, $\ell_t$ is $\sigma_t$-strongly convex. Assume that the FTL algorithm runs on this sequence and for each $t\in[T]$, let $g_t$ be in $\partial\ell_t(\theta_t)$. Then*

$$
\sum_{t=1}^{T}\ell_t(\theta_t)-\min_{\theta\in\Theta}\sum_{t=1}^{T}\ell_t(\theta)\leq\frac{1}{2}\sum_{t=1}^{T}\frac{\|g_t\|^2}{\sigma_{1:t}}.
$$

*Furthermore, let $L=\max_t\|g_t\|$ and assume that for all $t\in[T]$, $\sigma_t\geq\sigma$. Then the regret is bounded by $\frac{L^2}{2\sigma}(\log T+1)$.*

***Proof of Proposition 1.*** Notice that for all $t\in[T]$, $\frac{\mathrm{d}h_t(v)}{\mathrm{d}v}=1-\frac{f_t(\rho_{t1})}{v^2}$, $\frac{\mathrm{d}^2h_t(v)}{\mathrm{d}v^2}=\frac{2f_t(\rho_{t1})}{v^3}\geq\frac{2f_t(\rho_{t1})}{D^3}$. Therefore $h_t(v)$ is a $\frac{2f_t(\rho_{t1})}{D^3}$-strongly convex function. Besides, since $v_t=\sqrt{\frac{\sum_{s=1}^{t-1}f_s(\rho_{s1})}{t-1}}$, then $\frac{\mathrm{d}h_t(v)}{\mathrm{d}v}\Big|_{v=v_t}=1-\frac{\sum_{s=1}^{t-1}f_t(\rho_{t1})}{\sum_{s=1}^{t-1}f_s(\rho_{s1})}$, and applying Lemma C.1 obtains the first result. For the second

result, using $f_t(\rho_{t1}) \in [B^2, D^2]$, we bound the above regret as follow:

$$\frac{D^3}{4} \sum_{t=1}^{T} \frac{\left|1 - \sum_{s=1}^{t-1} f_t(\rho_{t1}) / \sum_{s=1}^{t-1} f_s(\rho_{s1})\right|^2}{\sum_{s=1}^{t} f_s(\rho_{s1})}$$

$$\leq \frac{D^3}{4B^2} \sum_{t=1}^{T} \frac{\left[\max\{1, \frac{\sum_{s=1}^{t-1} f_t(\rho_{t1})}{\sum_{s=1}^{t-1} f_s(\rho_{s1})}\}\right]^2}{t}$$

$$\leq \frac{D^3}{4B^2} \sum_{t=1}^{T} \frac{\left[\max\{1, \frac{(t-1)D^2}{(t-1)B^2}\}\right]^2}{t}$$

$$= \frac{D^7}{4B^6} \sum_{t=1}^{T} \frac{1}{t} \leq \frac{D^7}{4B^6}\Big(\log T + 1\Big). \quad \square$$

## D PROOFS OF THE REGRET BOUND FOR OWO META LEARNING WITH BOUNDED PIECEWISE-LIPSCHITZ FUNCTIONS

**Lemma D.1** *Let $\ell_1, ..., \ell_m : \Theta \mapsto [0, M]$ be any sequence of piecewise $L$-Lipschitz functions that are $\beta$-dispersed. Let $\theta^* \in \arg\min_{\theta \in \Theta} \sum_{i=1}^{m} \ell_i(\theta)$, and $V(\rho, \theta^*)^2 = -\log \frac{\int_{\mathcal{B}(\theta^*, m^{-\beta})} \rho(\theta)\mathrm{d}\theta}{\int_{\Theta} \rho(\theta)\mathrm{d}\theta}$, where $\rho$ is the initial distribution over $\Theta$ in EWA Algorithm. Then EWA has the following regret bound:*

$$\sum_{i=1}^{m} \mathbb{E}_{\theta \sim \rho_i} \ell_i(\theta) - \sum_{i=1}^{n} \ell_i(\theta^*) \leq \lambda M^2 m + \frac{V(\rho, \theta^*)^2}{\lambda} + \tilde{O}((L+1)m^{1-\beta}).$$

*Proof.* The demonstration strategy is to upper bound $P_{m+1}$ and lower bound $P_{m+1}$ respectively. For any $i \in [m]$, define the utility function $u_i(\theta) = M - \ell_i(\theta)$. It is not difficult to see that running EWA algorithm on utility functions $\{u_i\}_{i \in [m]}$ obtains the same sequence of actions $\{\theta_i\}_{i \in [m]}$ as running EWA on $\{\ell_i\}_{i \in [m]}$. For upper-bounding $P_{m+1}$, applying Jensen's inequality and basic inequality $1 + x \leq e^x$, we have:

$$\frac{P_{i+1}}{P_i} = \frac{\int_{\Theta} e^{\lambda u_i(\theta)} \rho_i(\theta)\mathrm{d}\theta}{P_i}$$

$$= \int_{\Theta} e^{\lambda u_i(\theta)} p_i(\theta)\mathrm{d}\theta$$

$$= \int_{\Theta} e^{\lambda M \frac{u_i(\theta)}{M} + 0(1 - \frac{u_i(\theta)}{M})} p_i(\theta)\mathrm{d}\theta$$

$$\leq \int_{\Theta} \left\{\frac{u_i(\theta)}{M} e^{\lambda M} + [1 - \frac{u_i(\theta)}{M}]e^0\right\} p_i(\theta)\mathrm{d}\theta$$

$$= \int_{\Theta} \left\{1 + \frac{u_i(\theta)}{M}(e^{\lambda M} - 1)\right\} p_i(\theta)\mathrm{d}\theta$$

$$= 1 + \frac{e^{\lambda M} - 1}{M} \int_{\Theta} u_i(\theta) p_i(\theta)\mathrm{d}\theta$$

$$\leq \exp\left\{\frac{e^{\lambda M} - 1}{M} \int_{\Theta} u_i(\theta) p_i(\theta)\mathrm{d}\theta\right\}.$$

Then $\frac{P_{m+1}}{P_1} = \prod_{i=1}^{m} \frac{P_{i+1}}{P_i} \leq \exp\left\{\frac{e^{\lambda M} - 1}{M} \sum_{i=1}^{m} \mathbb{E}_{\theta \sim p_i} u_i(\theta)\right\}$ and we can upper-bound $P_{m+1}$. For lower-bounding $P_{m+1}$, let us first upper bound $\sum_{i=1}^{m} u_i(\theta^*) - u_i(\theta)$, $\forall \theta \in \mathcal{B}(\theta^*, \epsilon)$, where $\epsilon = m^{-\beta}$. Notice that the utility functions $\{u_i\}_{i \in [m]}$ are also dispersed $L$-Lipschitz. Suppose there exist at most $k$ discontinuities w.r.t. to $\{u_i\}_{i \in [m]}$ in the ball $\mathcal{B}(\theta^*, \epsilon)$, then for any $\theta \in \mathcal{B}(\theta^*, \epsilon)$:

$$\sum_{i=1}^{m} u_i(\theta^*) - u_i(\theta) \leq k \max_i |u_i(\theta^*) - u_i(\theta)| + (m-k)L\|\theta - \theta^*\| \leq kM + mL\epsilon. \quad (4)$$

With Eq. (4), we can lower-bound $P_{m+1}$:

$$
\begin{aligned}
P_{m+1} &= \int_\Theta e^{\lambda \sum_{i=1}^m u_i(\theta)} \rho_1(\theta) \mathrm{d}\theta \\
&\geq \int_{\mathcal{B}(\theta^*,\epsilon)} e^{\lambda \sum_{i=1}^m u_i(\theta)} \rho_1(\theta) \mathrm{d}\theta \\
&\geq \int_{\mathcal{B}(\theta^*,\epsilon)} e^{\lambda \left[ \sum_{i=1}^m u_i(\theta^*) - kM - mL\epsilon \right]} \rho_1(\theta) \mathrm{d}\theta \\
&= e^{\lambda \left[ \sum_{i=1}^m u_i(\theta^*) - kM - mL\epsilon \right]} \int_{\mathcal{B}(\theta^*,\epsilon)} \rho_1(\theta) \mathrm{d}\theta.
\end{aligned}
$$

Combining the upper and lower bounds for $P_{m+1}$, we have:

$$
\exp\left\{ \lambda\big[\sum_{i=1}^m u_i(\theta^*) - kM - mL\epsilon\big] \right\} \int_{\mathcal{B}(\theta^*,\epsilon)} \rho_1(\theta)\mathrm{d}\theta \leq P_{m+1} \leq \exp\left\{ \frac{e^{\lambda M} - 1}{M} \sum_{i=1}^m \mathbb{E}_{\theta \sim p_i} u_i(\theta) \right\} \int_\Theta \rho_1(\theta)\mathrm{d}\theta
$$

Rearranging the terms, we have

$$
\sum_{i=1}^m u_i(\theta^*) - \sum_{i=1}^m \mathbb{E}_{\theta \sim p_i} u_i(\theta) \leq kM + mL\epsilon + \frac{e^{\lambda M} - 1 - \lambda M}{\lambda M} \sum_{i=1}^m \mathbb{E}_{\theta \sim p_i} u_i(\theta) + \frac{V(\rho_1, \theta^*)^2}{\lambda}
$$

$$
\leq kM + mL\epsilon + \lambda M^2 m + \frac{V(\rho_1, \theta^*)^2}{\lambda},
$$

where in the last inequality we use the fact that $\forall x \in [0,1], e^x \leq 1 + x + x^2$ and $u_i(\theta) \leq M$. Taking expectation of both sides w.r.t. the randomness of $\{u_i\}_{i \in [m]}$ and noticing $\mathbb{E}k \leq \tilde{O}(\epsilon m)$ according to Definition 1 gives the result. □

**Corollary D.1** *If the non-convex loss functions $\{\ell_i\}_{i \in [m]}$ are uniformly $L$-Lipschitz over $\Theta$, then the number $k$ of discontinuities in Eq. (4) becomes $0$, and the expected regret $R_m \leq Lm^{1-\beta} + \lambda mM^2 + \frac{V(\rho_1, \theta^*)^2}{\lambda}$.*

***Proof of Proposition 2.*** Note that $f_t(\boldsymbol{\rho}) = -\log\langle \boldsymbol{\rho}_t^*, \boldsymbol{\rho}\rangle$ is a $\frac{1}{\gamma \text{vol}(\Theta_t)}$-Lipschitz function w.r.t. $\|\cdot\|_1$ over the constraint simplex domain $\{\boldsymbol{\rho} \in \mathbb{R}^{|\mathcal{D}_T|} : \|\boldsymbol{\rho}\|_1 = 1, \boldsymbol{\rho} \geq \gamma \hat{\boldsymbol{\nu}}\}$, since

$$
\begin{aligned}
\max_{\|\boldsymbol{\rho}\|_1=1, \boldsymbol{\rho}\geq\gamma\hat{\boldsymbol{\nu}}} \|\nabla f_t(\boldsymbol{\rho})\|_\infty &= \max_{\|\boldsymbol{\rho}\|_1=1, \boldsymbol{\rho}\geq\gamma\hat{\boldsymbol{\nu}}} \left\| \frac{\boldsymbol{\rho}_t^*}{\langle \boldsymbol{\rho}_t^*, \boldsymbol{\rho}\rangle} \right\|_\infty \\
&= \max_{A, \boldsymbol{\rho}\geq\gamma\hat{\boldsymbol{\nu}}} \frac{\boldsymbol{\rho}_{t[A]}^*}{\langle \boldsymbol{\rho}_t^*, \boldsymbol{\rho}\rangle} \leq \max_{\boldsymbol{\rho}\geq\gamma\hat{\boldsymbol{\nu}}} \frac{1}{\langle \boldsymbol{\rho}_t^*, \boldsymbol{\rho}\rangle} \leq \frac{1}{\langle \boldsymbol{\rho}_t^*, \gamma\hat{\boldsymbol{\nu}}\rangle} \\
&= \frac{1}{\gamma \sum_{A \in \mathcal{D}_T, A \subset \Theta_t} \hat{\boldsymbol{\nu}}_{[A]}} = \frac{\text{vol}(\Theta)}{\gamma \text{vol}(\Theta_t)}.
\end{aligned}
$$

Then applying the regret bound in (Shalev-Shwartz, 2012, Thm 2.11) for FTRL algorithm with convex and Lipschitz functions, and noticing that $\text{KL}(\boldsymbol{\rho}^* \| \boldsymbol{\nu})$ is 1-strongly convex w.r.t. its first argument, we can obtain

$$
\begin{aligned}
\sum_{t=1}^T f_t(\boldsymbol{\rho}_t) - f_t(\boldsymbol{\rho}^*) &\leq \frac{\text{KL}(\boldsymbol{\rho}^* \| \hat{\boldsymbol{\nu}})}{\eta} + \eta \sum_{t=1}^T \frac{\text{vol}(\Theta)^2}{\gamma^2 \text{vol}(\Theta_t)^2} \\
&\leq \frac{G^2}{\eta} + \eta T L^2 / \gamma^2.
\end{aligned}
$$

Setting $\eta = G\gamma / L\sqrt{T}$ gives the result. □

**Proposition D.1** *(Proposition 3 in the main paper) Assume that the FTL algorithm runs on the sequence of functions $\{h_t(v) = v + \frac{f_t(\rho_{t1})}{v}\}_{t \in [T]}$ with $f_t(\rho_{t1}) = -\log\langle \boldsymbol{\rho}_t^*, \boldsymbol{\rho}_{t1}\rangle$ over*

the domain $[0, D]$, where $D^2 = \max_{t \in [T]} \log \frac{\text{vol}(\Theta)}{\gamma \text{vol}(\Theta_t)} \geq \max_{t \in [T]} f_t(\rho_{t1})$. Denote $B^2 = \log \frac{\text{vol}(\Theta)}{\gamma \text{vol}(\Theta_t) + (1-\gamma)\text{vol}(\Theta)}$. Then we have

$$\sum_{t=1}^{T} h_t(v_t) - \min_{v \in [0, D]} \sum_{t=1}^{T} h_t(v) \leq \frac{D^7 (\log T + 1)}{4B^6}.$$

Furthermore, if we set $\gamma = m^{d\beta}/T^\alpha$ with $\alpha \in (0, 1/2)$, we have the regret bound $\sum_{t=1}^{T} h_t(v_t) - \min_{v \in [0, D]} \sum_{t=1}^{T} h_t(v) \leq \frac{(\alpha \log T + \log (\text{vol}(\Theta)/V_d))^{\frac{7}{2}} (\log T + 1)}{4B^6}$.

**Proof.** According to the proof of Proposition 2, the upper bound $D^2$ of $f_t(\rho_{t1})$ is: $f_t(\boldsymbol{\rho}_{t1}) = \log \frac{1}{\langle \boldsymbol{\rho}_t^*, \boldsymbol{\rho}_{t1} \rangle} \leq \log \frac{\text{vol}(\Theta)}{\gamma \text{vol}(\Theta_t)} = D^2$. Then we need to derive the lower bound $B^2$ of $f_t(\rho_{t1})$. Let us first consider the maximum of $\langle \boldsymbol{\rho}_t^*, \boldsymbol{\rho} \rangle$, under the conditions of $\|\boldsymbol{\rho}\|_1 = 1$ and $\boldsymbol{\rho} \geq \gamma \hat{\boldsymbol{\nu}}$. Using the Hölder inequality, we can simply obtain that $\langle \boldsymbol{\rho}_t^*, \boldsymbol{\rho} \rangle \leq \|\boldsymbol{\rho}_t^*\|_\infty \|\boldsymbol{\rho}\|_1 = 1$. But this means that $\log \frac{1}{\langle \boldsymbol{\rho}_t^*, \boldsymbol{\rho} \rangle} \geq \log 1 = 0$, so the lower bound of $\log \frac{1}{\langle \boldsymbol{\rho}_t^*, \boldsymbol{\rho} \rangle}$ is zero, and thus we cannot use Lemma C.1. Actually, considering the additional condition $\boldsymbol{\rho} \geq \gamma \hat{\boldsymbol{\nu}}$, we can use Lagrange multiplier method to calculate the maximum, or just simply achieve the maximum by observing the fact that: apart from the measure $\gamma \sum_{A \in \mathcal{D}_T} \hat{\boldsymbol{\nu}}_{[A]} = \gamma$, the rest measure $1 - \gamma$ of $\rho$ should be assigned uniformly to the indices $A \in \mathcal{D}_T$ where $\boldsymbol{\rho}_{t [A]}^* = 1$ (i.e. $A \subset \Theta_t$) to maximize $\langle \boldsymbol{\rho}_t^*, \boldsymbol{\rho} \rangle$. Assuming the cardinality of the set $\{A | A \in \mathcal{D}_T, A \subset \Theta_t\}$ is $n$, then $\max_{\boldsymbol{\rho} \geq \gamma \hat{\boldsymbol{\nu}}} \langle \boldsymbol{\rho}_t^*, \boldsymbol{\rho} \rangle = \frac{1-\gamma}{n} n + \sum_{A \in \mathcal{D}_T, A \subset \Theta_t} \gamma \hat{\boldsymbol{\nu}}_{[A]} = 1 - \gamma + \frac{\gamma \text{vol}(\Theta_t)}{\text{vol}(\Theta)}$, which is strictly less than 1. Thus we have $\log \frac{1}{\langle \boldsymbol{\rho}_t^*, \boldsymbol{\rho} \rangle} \geq \log \frac{1}{1 - \gamma + \frac{\gamma \text{vol}(\Theta_t)}{\text{vol}(\Theta)}} = \log \frac{\text{vol}(\Theta)}{\gamma \text{vol}(\Theta_t) + (1-\gamma)\text{vol}(\Theta)}$. Applying this lower bound and the upper bound $D^2$ of $f_t(\rho_{t1})$ into Lemma C.1, we obtain the first result. For the second result, noticing $D^2 = \log \frac{\text{vol}(\Theta)}{\gamma \text{vol}(\Theta_t)} = \log \frac{m^{d\beta} \text{vol}(\Theta)}{\gamma V_d} = \log \frac{T^\alpha \text{vol}(\Theta)}{V_d}$ completes the whole proof. $\square$

**Remark D.1** *Assume that $T$ is large enough. Then if $\alpha \in \left[ \frac{\ln m^{d\beta}}{\ln T}, \frac{\ln \left( \frac{em^{d\beta}}{e-1} - \frac{V_d}{(e-1)\text{vol}(\Theta)} \right)}{\ln T} \right] \subset (0, \frac{1}{2})$, we have $\frac{m^{d\beta}}{T^\alpha} \leq 1$ and $B^2 = \log \frac{\text{vol}(\Theta)}{\gamma \text{vol}(\Theta_t) + (1-\gamma)\text{vol}(\Theta)} \geq \log e = 1$. The regret bound for learning the step size of EWA algorithm in Proposition 3 is of order $O((\log T)^{\frac{9}{2}})$*

**Proof of Theorem 2.** According to Lemma D.1, we know that the constant $b$ in Theorem 1 is equal to $b = M^2$. Then setting the distribution $\rho_t^*$ in Theorem 1 as the Dirac measure on the optimal action $\theta_t^*$, and applying the regret bound in Theorem 1, we have

$$\frac{1}{T} \sum_{t=1}^{T} \sum_{i=1}^{m} \mathbb{E}\left[ \langle \rho_{ti}, \ell_{ti} \rangle - \ell_{ti}(\theta_t^*) \right] \leq \left( \frac{H_T(V)}{T} + \min\{ \frac{F_T(\rho^*)}{VT}, 2\sqrt{\frac{F_T(\rho^*)}{T}} \} + 2V \right) \sqrt{m} M + g(m),$$

where the expectation in the left-hand side is taken over the randomness of loss functions $\{\ell_{ti}\}_{t,i \geq 1}$. It remains to provide the regret bounds for learning the initialization and for learning the step size. For learning the initialization, note that $\gamma = \frac{m^{\beta d}}{T^\alpha}$, $L^2 = \frac{1}{T} \sum_{t=1}^{T} \frac{\text{vol}(\Theta)^2}{\text{vol}(\Theta_t)^2} = \frac{m^{2\beta d} \text{vol}(\Theta)^2}{V_d^2}$, then the regret of FTRL in Proposition 2 for learning the initialization satisfies:

$$F_T(\boldsymbol{\rho}^*) = \frac{\text{KL}(\boldsymbol{\rho}^* \| \hat{\boldsymbol{\nu}})}{\eta} + \frac{\eta T L^2}{\gamma^2}$$

$$= \frac{\text{KL}(\boldsymbol{\rho}^* \| \hat{\boldsymbol{\nu}})}{\eta} + \eta T^{1+2\alpha} \text{vol}(\Theta)^2 / V_d^2$$

$$\leq T^{\frac{1}{2}+\alpha} \text{vol}(\Theta)(\text{KL}(\boldsymbol{\rho}^* \| \hat{\boldsymbol{\nu}}) + 1) / V_d,$$

where the last inequality holds by setting $\eta = V_d T^{-\frac{1+2\alpha}{2}} \text{vol}(\Theta)^{-1}$. For learning the step size, it suffices to directly apply Proposition 3. Thus we can upper bound the task-averaged regret:

$$\bar{R}_{T,m} \leq g(m) + \left\{ \frac{(\alpha \log T + \log (\text{vol}(\Theta)/V_d))^{\frac{7}{2}} (\log T + 1)}{4TB^6} \right.$$

$$\left. + \min \left\{ \frac{\text{vol}(\Theta)(\text{KL}(\boldsymbol{\rho}^* \| \hat{\boldsymbol{\nu}}) + 1)}{V_d V T^{\frac{1}{2}-\alpha}}, 2\sqrt{\frac{\text{vol}(\Theta)(\text{KL}(\boldsymbol{\rho}^* \| \hat{\boldsymbol{\nu}}) + 1)}{V_d T^{\frac{1}{2}-\alpha}}} \right\} + 2V \right\} \sqrt{m} M. \qquad \square$$

# E    PROOFS OF THE REGRET BOUND FOR OWO META LEARNING WITH BOUNDED NON-LIPSCHITZ FUNCTIONS

**Lemma E.1 (Minimizer of the Expected Bregman Divergence)** *(Frigyik et al., 2008, Thm II.1)*
*Let $\mathcal{C}$ be a set of functions that lie on a finite-dimensional manifold $\Omega$, and have associated differential element $d\Omega$. Suppose there is a probability distribution $P_F$ defined over the set $\mathcal{C}$. Suppose the function $g^*$ minimizes the expected Bregman divergence $d_\phi$ between the random function $F$ and any function $g \in \mathcal{A}$ such that $g^* = \arg\inf_{g \in \mathcal{A}} \mathbb{E}_{P_F}[d_\phi(F, g)]$. Then, if $g^*$ exists, it is given by $g^* = \int_\Omega f P(f) d\Omega = \mathbb{E}_{P_F}[F]$.*

***Proof of Proposition 4.*** In the proof, we write $\rho_{t1}$ in the main text as $\rho_t$ for brevity. According to Lemma E.1, we have

$$\rho_t = \arg\min_{\rho \in \mathcal{P}(\Theta)} \sum_{s=1}^{t-1} \mathrm{KL}(\rho_t^* || \rho) = \arg\min_{\rho \in \mathcal{P}(\Theta)} \frac{1}{t-1} \sum_{s=1}^{t-1} \mathrm{KL}(\rho_s^* || \rho) = \frac{1}{t-1} \sum_{s=1}^{t-1} \rho_s^*.$$

The existence of $\rho_t$ can be guaranteed by the convexity of KL-divergence w.r.t. its second argument. We also have $\rho^* = \arg\min_{\rho \in \mathcal{P}(\Theta)} \sum_{t=1}^{T} \mathrm{KL}(\rho_t^* || \rho) = \frac{1}{T} \sum_{t=1}^{T} \rho_t^*$. Then we have

$$\sum_{t=1}^{T} \mathrm{KL}(\rho_t^* || \rho_t) - \sum_{t=1}^{T} \mathrm{KL}(\rho_t^* || \rho^*)$$

$$\leq \sum_{t=1}^{T} \mathrm{KL}(\rho_t^* || \rho_t) - \sum_{t=1}^{T} \mathrm{KL}(\rho_t^* || \rho_{t+1})$$

$$= \sum_{t=1}^{T} \int \log \frac{d\rho_t^*}{d\rho_t} \rho_t^*(d\theta) - \int \log \frac{d\rho_t^*}{d\rho_{t+1}} \rho_t^*(d\theta)$$

$$= \sum_{t=1}^{T} \int \log \left[ \frac{d\rho_t^*}{d\rho_t} \Big/ \frac{d\rho_t^*}{d\rho_{t+1}} \right] \rho_t^*(d\theta)$$

$$= \sum_{t=1}^{T} \int \log \left[ \frac{d\rho_{t+1}}{d\rho_t} \right] \rho_t^*(d\theta)$$

$$= \sum_{t=1}^{T} \int \log \left[ \frac{1}{t} \frac{d((t-1)\rho_t + \rho_t^*)}{d\rho_t} \right] \rho_t^*(d\theta)$$

$$= \sum_{t=1}^{T} \int \log \left[ \frac{t-1}{t} + \frac{1}{t} \frac{d\rho_t^*}{d\rho_t} \right] \rho_t^*(d\theta)$$

$$\leq \sum_{t=1}^{T} \log \int \left[ \frac{t-1}{t} + \frac{1}{t} \frac{d\rho_t^*}{d\rho_t} \right] \rho_t^*(d\theta)$$

$$= \sum_{t=1}^{T} \log \left[ \frac{t-1}{t} + \frac{1}{t} \int \frac{d\rho_t^*}{d\rho_t} \rho_t^*(d\theta) \right]$$

$$= \sum_{t=1}^{T} \log \left[ 1 + \frac{1}{t} \int \left[ (\frac{d\rho_t^*}{d\rho_t})^2 - 1 \right] \rho_t(d\theta) \right]$$

$$= \sum_{t=1}^{T} \log \left[ 1 + \frac{1}{t} \chi^2(\rho_t^* || \rho_t) \right]$$

$$\leq \sum_{t=1}^{T} \frac{\chi^2(\rho_t^* || \rho_t)}{t},$$

where the first inequality holds due to the the follow-the-leader lemma (i.e. (Shalev-Shwartz, 2012, Lemma 2.1)), the second inequality holds due to Jensen's inequality, and the last step due to the fact that $\log(1 + x) \leq x$. □

**Proof of Proposition 5.** Applying Lemma C.1 finishes the whole proof. It remains to verify that the lower bound of $f_t(\rho_{t1})$ is strictly greater that 0. Actually for each task $t \in [T]$, we use EWA algorithm to learn posterior $\rho_t^*$, then we have $\frac{\mathrm{d}\rho_t^*}{\mathrm{d}\rho_{t1}}(\theta) = \frac{\exp\{-\lambda_t \sum_{i=1}^m \ell_{ti}(\theta)\}}{\int \exp\{-\lambda_t \sum_{i=1}^m \ell_{ti}(\theta)\}\rho_{t1}(\theta)} \not\equiv$ constant. If not, then $\exp\{-\lambda_t \sum_{i=1}^m \ell_{ti}(\theta)\} \equiv$ constant and hence $\sum_{i=1}^m \ell_{ti}(\theta) =$ constant for all $\theta \in \Theta$, which is trivial. Therefore $\rho_t^* \neq \rho_{t1}$ and $\mathrm{KL}(\rho_t^*||\rho_{t1}) > 0$ for all $t \in [T]$. $\qquad\square$

**Proof of Theorem 3.** According to Lemma, set the positive constant $b$ in Theorem 1 as $M^2$, the additional term $g(m) = 0$. Then integrating the regret bounds from Proposition 4 and Proposition 5 into the regret bound in Theorem 1, we finish the proof. $\qquad\square$

## F  PROOFS OF THE TRANSFER RISK BOUND

**Proposition F.1** *Let a sequence of non-negative loss functions $\{\ell_i : \Theta \mapsto \mathbb{R}_{\geq 0}\}_{i\in[m]}$ drawn i.i.d. from some distribution $\mu$ be given to an online algorithm that generates a sequence of probability distributions $\{\rho_i \in \mathcal{P}(\Theta)\}_{i\in[m]}$ (i.e. $\rho_i = \rho_i(\{\ell_j\}_{j=1}^{i-1})$). Assume the regret upper bound for the online algorithm is $U_m$. Then let $\bar{\rho} = \frac{1}{m}\sum_{i=1}^m \rho_i$, for any $\rho^* \in \mathcal{P}(\Theta)$ that does not depend on the choice of the sequence of loss functions $\{\ell_i\}_{i\in[m]}$, we have the following bound:*

$$\mathbb{E}_{\{\ell_i\}_{i=1}^m \sim \mu^m} \mathbb{E}_{\ell\sim\mu} \mathbb{E}_{\theta\sim\bar{\rho}} \ell(\theta) \leq \frac{U_m}{m} + \mathbb{E}_{\ell\sim\mu}\mathbb{E}_{\theta\sim\rho^*}\ell(\theta).$$

**Proof.**

$$\mathbb{E}_{\{\ell_i\}_{i=1}^m} \mathbb{E}_{\ell\sim\mu}\mathbb{E}_{\theta\sim\bar{\rho}}\ell(\theta)$$

$$= \mathbb{E}_{\{\ell_i\}_{i=1}^m}\mathbb{E}_{\ell\sim\mu}\int \ell(\theta)\bar{\rho}(\mathrm{d}\theta)$$

$$= \mathbb{E}_{\{\ell_i\}_{i=1}^m}\mathbb{E}_{\ell\sim\mu}\frac{1}{m}\sum_{i=1}^m \int \ell(\theta)\rho_i(\mathrm{d}\theta)$$

$$= \mathbb{E}_{\{\ell_i\}_{i=1}^m}\frac{1}{m}\sum_{i=1}^m \int \mathbb{E}_{\ell_i'\sim\mu}\ell_i'(\theta)\rho_i(\mathrm{d}\theta)$$

$$= \mathbb{E}_{\{\ell_i\}_{i=1}^m}\frac{1}{m}\sum_{i=1}^m \left[\int \mathbb{E}_{\ell_i'\sim\mu}\ell_i'(\theta) - \ell_i(\theta)\rho_i(\mathrm{d}\theta) + \int \ell_i(\theta)\rho_i(\mathrm{d}\theta)\right]$$

$$= \frac{1}{m}\sum_{i=1}^m \mathbb{E}_{\{\ell_i\}_{i=1}^m}\left[\int \mathbb{E}_{\ell_i'\sim\mu}\ell_i'(\theta) - \ell_i(\theta)\rho_i(\mathrm{d}\theta)\right] + \mathbb{E}_{\{\ell_i\}_{i=1}^m}\frac{1}{m}\sum_{i=1}^m \int \ell_i(\theta)\rho_i(\mathrm{d}\theta)$$

$$= \frac{1}{m}\sum_{i=1}^m \mathbb{E}_{\{\ell_j\}_{j=1}^{i-1}\sim\mu^{i-1}}\left[\int \mathbb{E}_{\ell_i'\sim\mu}\ell_i'(\theta) - \mathbb{E}_{\ell_i\sim\mu}\ell_i(\theta)\rho_i(\mathrm{d}\theta)\right] + \mathbb{E}_{\{\ell_i\}_{i=1}^m}\frac{1}{m}\sum_{i=1}^m \int \ell_i(\theta)\rho_i(\mathrm{d}\theta)$$

$$= \mathbb{E}_{\{\ell_i\}_{i=1}^m}\frac{1}{m}\sum_{i=1}^m \int \ell_i(\theta)\rho_i(\mathrm{d}\theta)$$

$$= \mathbb{E}_{\{\ell_i\}_{i=1}^m}\frac{1}{m}\sum_{i=1}^m \left[\mathbb{E}_{\theta\sim\rho_i}\ell_i(\theta) - \mathbb{E}_{\theta\sim\rho^*}\ell_i(\theta)\right] + \mathbb{E}_{\{\ell_i\}_{i=1}^m}\frac{1}{m}\sum_{i=1}^m \mathbb{E}_{\theta\sim\rho^*}\ell_i(\theta)$$

$$\leq \frac{U_m}{m} + \mathbb{E}_{\{\ell_i\}_{i=1}^m}\frac{1}{m}\sum_{i=1}^m \mathbb{E}_{\theta\sim\rho^*}\ell_i(\theta)$$

$$= \frac{U_m}{m} + \mathbb{E}_{\ell\sim\mu}\mathbb{E}_{\theta\sim\rho^*}\ell(\theta),$$

where the third equality holds due to the independence between $\ell \sim \mu$ and $\rho_i$, the Fubini-Tonelli Theorem for changing the order of integrals of non-negative function, as well as the fact that $\ell_i'$ is the i.i.d. copy of $\ell_i$; in the six-th equality, $\{\ell_j\}_{j=1}^{i-1} \sim \mu^{i-1}$ is the abbreviation for the notation $\ell_j \sim \mu, \forall j \in [i-1]$; and $\mu^{i-1} = \mu \times \cdots \times \mu$ is the product measure of $i-1$ measures $\mu$. Both the six-th and the last equality hold due to the independence between $\rho^*$ and $\{\ell_i\}_{i=1}^m$ (i.e. Fubini's Theorem for exchanging the order of integrals). $\qquad\square$

**Proposition F.2**

$$\frac{1}{T}\sum_{t=1}^{T}\mathrm{KL}(\rho_t^*\|\frac{1}{T}\sum_{t=1}^{T}\rho_{t1}) - \frac{1}{T}\sum_{t=1}^{T}\mathrm{KL}(\rho_t^*\|\rho_{t1}) \le \frac{1}{T}\sum_{t=1}^{T}\mathrm{KL}(\rho_t^*\|\frac{1}{T}\sum_{t=1}^{T}\rho_t^*) = V^2. \quad (5)$$

*Furthermore, assume $G^2 \ge \max_{t\in[T]}\chi^2(\rho_t^*\|\rho_{t1})$, we have*

$$\frac{1}{T}\sum_{t=1}^{T}\mathrm{KL}(\rho_t^*\|\frac{1}{T}\sum_{t=1}^{T}\rho_{t1}) \le 2V^2 + \frac{G^2(\log T + 1)}{T}. \quad (6)$$

**Proof.** (1) For the first inequality. Using the joint convexity of KL-divergence w.r.t. its pair argument and Jensen's inequality, we have

$$\mathrm{KL}(\frac{1}{T}\sum_{t=1}^{T}\rho_t^*\|\frac{1}{T}\sum_{t=1}^{T}\rho_{t1}) \le \frac{1}{T}\sum_{t=1}^{T}\mathrm{KL}(\rho_t^*\|\rho_{t1}).$$

Plug the above result into the left-hand-sight of inequality in proposition, we have

$$\frac{1}{T}\sum_{t=1}^{T}\mathrm{KL}(\rho_t^*\|\frac{1}{T}\sum_{t=1}^{T}\rho_{t1}) - \frac{1}{T}\sum_{t=1}^{T}\mathrm{KL}(\rho_t^*\|\rho_{t1})$$

$$\le \frac{1}{T}\sum_{t=1}^{T}\Big[\mathrm{KL}(\rho_t^*\|\frac{1}{T}\sum_{t=1}^{T}\rho_{t1}) - \mathrm{KL}(\frac{1}{T}\sum_{t=1}^{T}\rho_t^*\|\frac{1}{T}\sum_{t=1}^{T}\rho_{t1})\Big]$$

$$= \frac{1}{T}\sum_{t=1}^{T}\Big[\mathbb{E}_{\theta\sim\rho_t^*}\ln\frac{\mathrm{d}\rho_t^*}{\mathrm{d}\frac{1}{T}\sum_{t=1}^{T}\rho_{t1}} - \mathbb{E}_{\theta\sim\frac{1}{T}\sum_{t=1}^{T}\rho_t^*}\ln\frac{\mathrm{d}\frac{1}{T}\sum_{t=1}^{T}\rho_t^*}{\mathrm{d}\frac{1}{T}\sum_{t=1}^{T}\rho_{t1}}\Big]$$

$$= \frac{1}{T}\sum_{t=1}^{T}\Big[\mathbb{E}_{\theta\sim\rho_t^*}\ln\big(\frac{\mathrm{d}\rho_t^*}{\mathrm{d}\frac{1}{T}\sum_{t=1}^{T}\rho_t^*}\cdot\frac{\mathrm{d}\frac{1}{T}\sum_{t=1}^{T}\rho_t^*}{\mathrm{d}\frac{1}{T}\sum_{t=1}^{T}\rho_{t1}}\big) - \mathbb{E}_{\theta\sim\frac{1}{T}\sum_{t=1}^{T}\rho_t^*}\ln\frac{\mathrm{d}\frac{1}{T}\sum_{t=1}^{T}\rho_t^*}{\mathrm{d}\frac{1}{T}\sum_{t=1}^{T}\rho_{t1}}\Big]$$

$$= \frac{1}{T}\sum_{t=1}^{T}\Big[\mathbb{E}_{\theta\sim\rho_t^*}\ln\big(\frac{\mathrm{d}\rho_t^*}{\mathrm{d}\frac{1}{T}\sum_{t=1}^{T}\rho_t^*}\big) + \mathbb{E}_{\theta\sim\rho_t^*}\ln\big(\frac{\mathrm{d}\frac{1}{T}\sum_{t=1}^{T}\rho_t^*}{\mathrm{d}\frac{1}{T}\sum_{t=1}^{T}\rho_{t1}}\big) - \mathbb{E}_{\theta\sim\frac{1}{T}\sum_{t=1}^{T}\rho_t^*}\ln\frac{\mathrm{d}\frac{1}{T}\sum_{t=1}^{T}\rho_t^*}{\mathrm{d}\frac{1}{T}\sum_{t=1}^{T}\rho_{t1}}\Big]$$

$$= \frac{1}{T}\sum_{t=1}^{T}\mathbb{E}_{\theta\sim\rho_t^*}\ln\frac{\mathrm{d}\rho_t^*}{\mathrm{d}\frac{1}{T}\sum_{t=1}^{T}\rho_t^*}$$

$$= \frac{1}{T}\sum_{t=1}^{T}\mathrm{KL}(\rho_t^*\|\frac{1}{T}\sum_{t=1}^{T}\rho_t^*).$$

(2) For the second inequality, notice the following decomposition

$$\frac{1}{T}\sum_{t=1}^{T}\mathrm{KL}(\rho_t^*\|\frac{1}{T}\sum_{t=1}^{T}\rho_{t1}) - \frac{1}{T}\sum_{t=1}^{T}\mathrm{KL}(\rho_t^*\|\frac{1}{T}\sum_{t=1}^{T}\rho_t^*)$$

$$= \big(\frac{1}{T}\sum_{t=1}^{T}\mathrm{KL}(\rho_t^*\|\frac{1}{T}\sum_{t=1}^{T}\rho_{t1}) - \frac{1}{T}\sum_{t=1}^{T}\mathrm{KL}(\rho_t^*\|\rho_{t1})\big) + \big(\frac{1}{T}\sum_{t=1}^{T}\mathrm{KL}(\rho_t^*\|\rho_{t1}) - \frac{1}{T}\sum_{t=1}^{T}\mathrm{KL}(\rho_t^*\|\frac{1}{T}\sum_{t=1}^{T}\rho_t^*)\big).$$

Therefore, we can upper bound the first part of the right-hand-side of above equality by applying the first inequality in this proposition (i.e. Eq. (5)), and upper bound the second part of the right-hand-side of above equality by applying the regret bound in Proposition 4. Combining both upper bounds yields the second inequality in this proposition (i.e. Eq. (6)). □

**Proof of Theorem 4.** Denote $\rho_{T+1,1} = \frac{1}{T}\sum_{t=1}^{T}\rho_{t1}, \lambda_{T+1} = \sqrt{\frac{\sum_{t=1}^{T}\mathrm{KL}(\rho_t^*\|\rho_{t1})}{TmM^2}}$ for brevity, then applying Proposition F.1, we have

$$\mathbb{E}_{\mu\sim\tau}\mathbb{E}_{\{\ell_i\}_{i=1}^m\sim\mu^m}\mathbb{E}_{\ell\sim\mu}\mathbb{E}_{\theta\sim\bar\rho}\ell(\theta) \le \mathbb{E}_{\mu\sim\tau}\Big(\mathbb{E}_{\ell\sim\mu}\mathbb{E}_{\theta\sim\rho^*}\ell(\theta) + \frac{U_m(\rho_{T+1,1},\lambda_{T+1})}{m}\Big).$$

We next give an upper bound on $\mathbb{E}_{\mu\sim\tau}\frac{U_m(\rho_{T+1,1},\lambda_{T+1})}{m}$. Actually,

$$
\begin{aligned}
&\mathbb{E}_{\mu\sim\tau}\frac{U_m(\rho_{T+1,1},\lambda_{T+1})}{m}\\
=&\mathbb{E}_{\mu\sim\tau}\frac{\lambda_{T+1}M^2m+\mathrm{KL}(\rho^*||\rho_{T+1,1})/\lambda_{T+1}}{m}\\
=&M\sqrt{\frac{\sum_{t=1}^T\mathrm{KL}(\rho_t^*||\rho_{t1})}{Tm}}+\mathbb{E}_{\mu\sim\tau}\frac{M\sqrt{T}\,\mathrm{KL}(\rho^*||\rho_{T+1,1})}{\sqrt{m}\sqrt{\sum_{t=1}^T\mathrm{KL}(\rho_t^*||\rho_{t1})}}\\
\leq&M\sqrt{\frac{TV^2+G^2(\log T+1)}{Tm}}+\mathbb{E}_{\mu\sim\tau}\frac{M\,\mathrm{KL}(\rho^*||\rho_{T+1,1})}{\sqrt{m}B}\\
=&M\sqrt{\frac{V^2}{m}+\frac{G^2(\log T+1)}{Tm}}+\frac{MH}{\sqrt{m}B}\mathbb{E}_{\mu\sim\tau}\frac{\mathrm{KL}(\rho^*||\rho_{T+1,1})}{H}\\
\leq&M\sqrt{\frac{V^2}{m}+\frac{G^2(\log T+1)}{Tm}}+\frac{MH}{\sqrt{m}B}\Big(\frac{1}{T}\sum_{t=1}^T\frac{\mathrm{KL}(\rho_t^*||\rho_{T+1,1})}{H}+\sqrt{\frac{\log 1/\delta}{2T}}\Big)\\
\leq&M\sqrt{\frac{V^2}{m}+\frac{G^2(\log T+1)}{Tm}}+\frac{MH}{\sqrt{m}B}\Big(\frac{2V^2}{H}+\frac{G^2(\log T+1)}{TH}+\sqrt{\frac{\log 1/\delta}{2T}}\Big)\\
\leq&M\Big(\frac{V}{\sqrt{m}}+\sqrt{\frac{G^2(\log T+1)}{Tm}}+\frac{2V^2}{\sqrt{m}B}+\frac{G^2(\log T+1)}{T\sqrt{m}B}+\frac{H}{B}\sqrt{\frac{\log 1/\delta}{2Tm}}\Big),
\end{aligned}
$$

where the first inequality holds due to the regret bound in Proposition 4 , the second inequality due to Hoeffding's inequality holds with probability $1-\delta$, the third inequality holds due to Eq. (6) in Proposition F.2. Combining these results completes the whole proof. $\qquad\square$

# G  PROOFS OF THE PAC-BAYESIAN GENERALIZATION BOUNDS FOR STATISTICAL MULTI-TASK LEARNING

**Lemma G.1** *(Lugosi & Neu, 2023, Lemma 26) Let $\{X_t\}_{t=1}^n$ be a sequence of non-negative random variables and for $t\geq 0$, let $\mathcal{F}_t$ denote the $\sigma$-algebra generated by $X_1,\ldots,X_t$. Assume that $X_t$ has finite conditional mean $\mu_t=\mathbb{E}[X_t|\mathcal{F}_{t-1}]$ and second moment $\sigma_t^2=\mathbb{E}[X_t^2|\mathcal{F}_{t-1}]$. Then, for any $\lambda>0$, the following bound holds with probability at least $1-\delta$:*

$$
\sum_{t=1}^n\big(\mu_t-X_t\big)\leq\frac{\lambda}{2}\sum_{t=1}^n\sigma_t^2+\frac{\log\frac{1}{\delta}}{\lambda}\leq\sqrt{2(\sum_{t=1}^n\sigma_t^2)\log\frac{1}{\delta}}.
$$

***Proof of Proposition 6.*** For any $t\in[T]$, define the regret of online algorithm $\Pi_{tm}$ w.r.t. the probability distribution $\rho_t^*$ as $R_{\Pi_{tm}}(\rho_t^*)=\sum_{i=1}^m\langle\rho_{ti}-\rho_t^*,c_{ti}\rangle$. According to (Lugosi & Neu, 2023, Thm 1), for any task $t\in[T]$, we have $\overline{\mathrm{gen}}(\mathcal{A}_t,S_t)=\frac{R_{\Pi_{tm}}(\mathcal{A}_t(S_t))}{m}-M_{\Pi_{tm}}$. Then we have

$$
\frac{1}{T}\sum_{t=1}^T\overline{\mathrm{gen}}(\mathcal{A}_t,S_t)=\frac{1}{T}\sum_{t=1}^T\Big(\frac{R_{\Pi_{tm}}(\mathcal{A}_t(S_t))}{m}-M_{\Pi_{tm}}\Big)=\frac{\bar{R}_{T,m}}{m}-\frac{1}{T}\sum_{t=1}^TM_{\Pi_{tm}}.\qquad\square
$$

***Proof of Theorem 5.*** Notice that $\mathbb{E}[\langle\rho_{ti},c_{ti}\rangle^2|\mathcal{F}_{t,i-1}]\leq M^2$, then applying Lemma G.1 to bound the sum of normalized martingale differences $-\frac{1}{Tm}\sum_{t=1}^T\sum_{i=1}^m\langle\rho_{ti},c_{ti}\rangle$ in Proposition 6, we have

$$
\frac{\bar{R}_{T,m}}{m}-\frac{1}{T}\sum_{t=1}^TM_{\Pi_{tm}}\leq\frac{\bar{U}_{T,m}}{m}-\frac{1}{Tm}\sum_{t=1}^T\sum_{i=1}^m\langle\rho_{ti},c_{ti}\rangle\leq\frac{\bar{U}_{T,m}}{m}+\sqrt{\frac{2M^2\log\frac{1}{\delta}}{Tm}},
$$

where the last inequality holds with probability at least $1-\delta$. $\qquad\square$

**Remark G.1** *(**Two Technical Novelties of deriving our Generalization Bounds for Statistical Meta Learning**). The technical novelties of our generalization bounds for statistical meta learning lie in the following two aspects: **(1) For the transfer risk bound in Theorem 4:** the novelties of*

*our online-to-batch technique lie in 2 aspects: **(i)** The first novelty lies in bounding $\mathrm{KL}(\rho^*||\rho_{T+1,1})$ with Proposition 4 and Proposition F.2, both of which require the technical analysis of the properties of the Radon-Nikodym derivative. To the best of our knowledge, such analysis does not exist in the previous online-to-batch literature. **(ii)** We further use concentration inequality to bound $\mathbb{E}_{\mu \sim \tau} \frac{\mathrm{KL}(\rho^*||\rho_{T+1,1})}{H}$, leading to the transfer risk bound in the non-convex setting. Such transfer risk bound, as shown in our Remark 3, has almost the same convergence rate when compared with the latest transfer risk bound in the convex setting (i.e. (Balcan et al., 2021, Thm E.1), which is obtained by Jensen inequality of convex loss), demonstrating the novelty of the online-to-batch analysis developed in our Theorem 4. **(2)** **For the PAC-Bayes generalization bound in Theorem 5:** first we need to admit that our Proposition 6 is a direct corollary of recent online-to-PAC result in (Lugosi & Neu, 2023, Thm 1) (i.e. we extent the result from single-task learning (Lugosi & Neu, 2023) to our multi-task learning setting), but the novelty of our Theorem 5 lies in the combination of the online-to-PAC analysis and the task-averaged regret analysis developed in our Theorem 3. Such combination pioneers a new research direction to demonstrating PAC-Bayes generalization error bounds for statistical multi-task learning. If we do not combine the aforementioned two analysis tools, but just use the online-to-PAC analysis in (Lugosi & Neu, 2023, Thm 1) and the traditional regret analysis for single-task learning in (Alquier, 2021, Thm 2.1), we can only obtain the following trivial (to some extent) PAC-Bayes generalization error bound for statistical multi-task learning:*

$$\frac{1}{T}\sum_{t=1}^{T} \mathbb{E}_{\theta \sim \mathcal{A}_t(S_t)}\Big[\mathbb{E}_{z \sim \mu_t}\ell(\theta, z) - \frac{1}{m}\sum_{i=1}^{T}\ell(\theta, z_{ti})\Big] \le \frac{\sqrt{m}M(1 + \frac{1}{T}\sum_{t=1}^{T}\mathrm{KL}(\rho_t^*||\rho_{t1}))}{m} + M\sqrt{\frac{2\log\frac{1}{\delta}}{Tm}},$$

*which is less informative than our PAC-Bayes bound for statistical multi-task learning in Theorem 5.*

## H  EXPERIMENTS

We conduct $k$-center clustering experiment to verify the convergence performance of task-averaged regret $\bar{R}_{T,m}/m$ of our Algorithm 1 for non-convex OWO meta learning. We follow the existing work (Balcan et al., 2021) and meta learn the hyper-parameter $\alpha$ in the $\alpha$-Lloyd's clustering algorithm.

**Introduction of $\alpha$-Lloyd's Algorithm**. $\alpha$-Lloyd's algorithm consists of two phases: seeding phase and local search phase. The goal of the seeding phase is to output $k$ initial centers with $\mathrm{d}^\alpha$-sampling. Concretely, each point $v$ is sampled with probability proportional to $\min_{c \in C} \mathrm{d}(v, c)^\alpha$, where $\mathrm{d}(\cdot, \cdot)$ is the distance metric and $C$ is the set of centers updated so far. The goal of local search phase is to run an iterative two-step procedure to output final centers. Specifically, the first step is to create a Voronoi tiling of all points induced by the initial set of centers from seeding phase; then, the new set of centers is updated by computing the centroid (e.g. median or mean) of each Voronoi tile. The $\alpha$-Lloyd's algorithm family include popular clustering methods like randomly initialized $k$-means ($\alpha = 0$) and farthest-first traversal ($\alpha = \infty$). The indicator used to measure the performance of $\alpha$-Lloyd's algorithm is the Hamming loss between the outputted clustering and the optimal target clustering (which is given in advance). The Hamming loss is a piecewise constant function of $\alpha$ and hence is a piecewise Lipschitz function. More explanations for this algorithm family and proof for the piecewise constant property of Hamming loss can be found in (Balcan et al., 2021, Section 4.1).

**Experimental Setting**. We conduct $k$-center clustering experiment on both synthetic and real-world datasets, under the same settings as that in Balcan et al. (2021). On the one hand, we create a Gaussian mixture binary classification dataset, where each class is a 2-dimensional diagonal Guassian distribution with variance $\sigma$ and $2\sigma$, as well as the expectation $(0,0)$ and $(b\sigma, 0)$. We set $b \in [2, 3]$ to generate different tasks. On the other hand, we utilize the split of the real-world Omniglot dataset to create clustering tasks, by drawing random samples each composed of five characters among which four are constant throughout. We set the number $T \in [1, 10]$ of training tasks and the number $m \in [5, 50]$ of samples per task for online optimization. Analogous to Balcan et al. (2021), we set the parameters $\gamma = \eta = 0.01$ (not hyper-parameter searched), and set the step size $\lambda$ in EWA algorithm to minimize the regret in Eq. (2) (not meta-learned). For any fixed $T$ and $m$, we run our non-convex OWO meta learning algorithm to learn the hyper-parameter $\alpha$ in the $\alpha$-Lloyd's algorithm, and calculate the normalized task-averaged regret $\bar{R}_{T,m}/m$ over the $T$ training tasks. In Figure 1, we show the convergence performance of $\bar{R}_{T,m}/m$ with respect to different number $T \in [1, 10]$ of training tasks, and validate the advantage of OWO meta learning over single-task learning. Each training task has $m$ samples and is called $m$-shot learning. In Figure 2, we exhibit the asymptotic performance of averaged regret $\bar{R}_{T,m}/m$ with respect to different number $m$ of samples on one task.

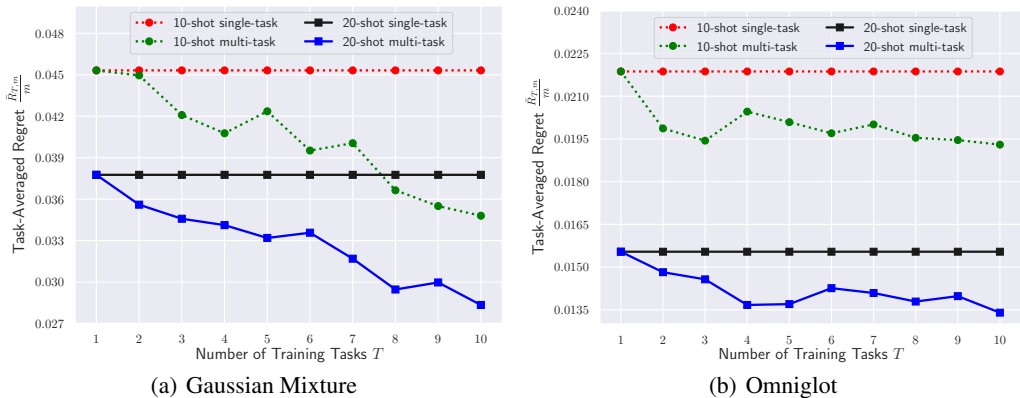

Figure 1: Task-averaged regret $\frac{\bar{R}_{T,m}}{m}$ with respect to the number $T$ of training tasks.

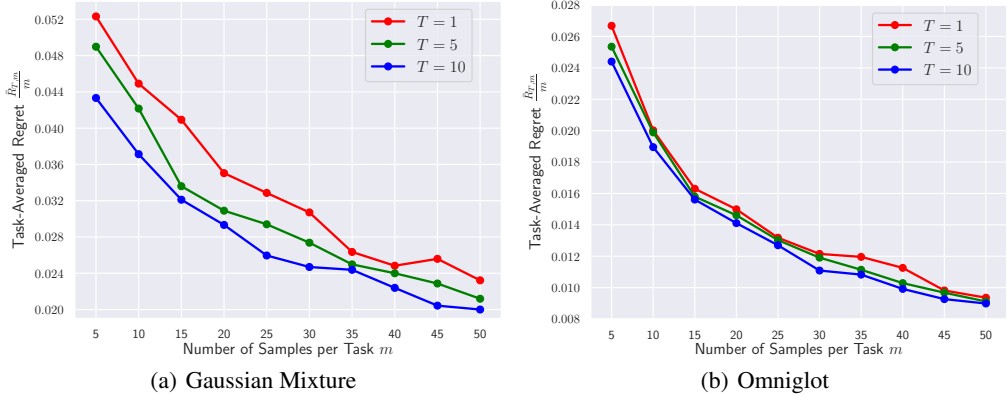

Figure 2: Task-averaged regret $\frac{\bar{R}_{T,m}}{m}$ with respect to the sample size $m$ per training task.

**Experimental Results in Figure 1**. We can observe that: **(1)** On both synthetic and real-world clustering dataset, running OWO meta learning algorithm can achieve sharper regret than that of online single-task algorithm, indicating the advantage of online meta learning framework. **(2)** When the training task number $T \geq 8$, the task-averaged regret $\bar{R}_{T,m}/m$ of '10-shot multi-task' algorithm in Figure 1(a) is smaller than the regret of '20-shot single-task' algorithm. This shows that, leveraging meta learning paradigm, even with less samples per task, can achieve the same or better performance compared with the single-task learning paradigm where the task has more training samples, hence alleviating the cost of collecting labeled data. **(3)** On both synthetic and real-world clustering dataset, the normalized task-averaged regret $\bar{R}_{T,m}/m$ always decreases with the increase of the number $T$ of training tasks, empirically validating the convergence performance of our regret for OWO meta learning. However, we need to point out that, in some cases of Figures 1(a)-(b),regret $\bar{R}_{T,m}/m$ does not decrease and even slightly increase when $T$ becomes larger. We attribute the counter-intuitive phenomenon to the encountering of a tough training task that is dissimilar to previous tasks and can lead to the increase of the value $V$ defined in Section 4. Therefore, the task-averaged regret bound that involves the task similarity $V$ in our Theorem 2 possibly becomes larger, and so does the regret. **(4)** On the real-world dataset Omniglot, the phenomenon that $\bar{R}_{T,m}/m$ slightly increases with the larger $T$ appears more frequently than on the synthetic Gaussian mixture dataset. This implies the higher degree of similarity of Gaussian mixture clustering task than that of Omniglot splitting tasks, and attaches great importance of task similarity to the success of online meta learning algorithms.

**Experimental Results in Figure 2**. We can observe that: **(1)** On both Gaussian mixture and Omniglot datasets, for any fixed $T$, the normalized task-averaged regret $\bar{R}_{T,m}/m$ decreases with the increase of the number $m$ of samples per task, verifying the vanishing regret property (i.e. $\bar{R}_{T,m} = o(m)$) of our results. **(2)** Leveraging OWO meta learning paradigm (i.e. when $T = 5$ or $T = 10$) achieves sharper regret than online single-task learning paradigm (i.e. $T = 1$). **(3)** When using more training tasks, the regret improvements on Gaussian mixture dataset are larger

than that on Omniglot dataset. This is consistent with the observation in Figure 1 that Gaussian mixture clustering tasks share a higher degree of task similarity than that of Omniglot splitting tasks, revealing the great effect of task similarity to the performance of online meta learning algorithms.

**Remark H.1** *(More Discussions of our Algorithm 2) (1) The Limitation Aspect. We need to admit that at the current stage we are unable to run our Algorithm 2 in practice, because it is hard to compute the analytic form of RN derivative of $\rho_{t+1,m}$ w.r.t. the Lebesgue measure $\nu$ (if we want to implement our algorithm in real-life applications, we need to compute $\frac{d\rho_{ti}}{d\nu}$ to let $\rho_{ti}$ be tractable in Euclidean space). The computation problem actually lies in line 6 (i.e. the meta update step of $\rho_{t+1,1} = \frac{1}{t}\sum_{s=1}^{t}\rho_{sm}$) in Algorithm 2, even though $\rho_{sm}$ is the Lebesgue measure (i.e. corresponding to uniform distribution) or Gaussian measure (i.e. corresponding to Gaussian distribution). Concretely, consider the simplest case where $\rho_{11}$ is the Lebesgue measure $\nu$, and hence using EWA obtains the Gaussian measure $\rho_{1m}$ (with density $\mathcal{N}(\mu_1,\sigma_1)$). Assume that $\rho_{2m}$ is also a Gaussian measure (with density $\mathcal{N}(\mu_2,\sigma_2)$), then $\rho_{31} = \frac{\rho_{1m}+\rho_{2m}}{2}$ and the density of $\rho_{3m}$ w.r.t. to $\nu$ is $\frac{d\rho_{3m}}{d\nu} = \frac{d\rho_{3m}}{d\rho_{31}} \cdot \frac{d(\rho_{1m}+\rho_{2m})}{2d\nu} \propto \exp\{-\sum_{i\in[m]}\ell_{3i}\}\cdot(\mathcal{N}(\mu_1,\sigma_1)+\mathcal{N}(\mu_2,\sigma_2))$, indicating that $\rho_{3m}$ is not a Gaussian measure and it is hard to compute the precise form of the density function $\frac{d\rho_{3m}}{d\nu}$. Nevertheless, it will also be difficult to compute the precise form of the density function $\frac{d\rho_{t+1,1}}{d\nu}$, as well as the precise value of the normalized constant $\int\exp\{-\lambda_{t+1}\sum_{j=1}^{m}\ell_{t+1,j}(\theta)\}\rho_{t+1,1}(d\theta)$. Therefore, it is also not easy to compute the precise form of $\frac{d\rho_{t+1,m}}{d\rho_{t+1,1}}$ over the continuous domain.*
*(2) The Potential Application Aspect. However, we point out that our Algorithm 2 is potentially applicable to the discrete domain where the updating rule $\rho_{t+1,1} = \sum_{s=1}^{t}\rho_{sm}/t$ of our Algorithm 2 corresponds to the averaging of previous $t$ discrete probability density vectors, which is more feasible. Thus our algorithm may be utilized to the Expert Advice problem for online model selection (see (Cesa-Bianchi & Lugosi, 2006, Sect 2)), and this serves as one of our ongoing research directions.*

**Remark H.2** *(Detailed Comparisons between our Algorithm 1 and other Algorithms for Non-Convex OWO Meta Learning) We discuss more differences between our Algorithm 1 and the state-of-the-art algorithm from (Balcan et al., 2021, Alg 3) as well as other baseline for OWO meta learning. The detailed explanations are three-fold: (1) The comparisons with the single-task algorithm. Actually, we choose single-task EWA algorithm as our baseline in all experiments. In both Figure 1 ($m$-shot multi-task method v.s. single-task baseline) and Figure 2 ($T = 5$ multi-task method v.s. $T = 1$ single-task baseline), we show advantages of the meta-learning based EWA algorithm over the single-task EWA algorithm baseline, and verify the convergence performance of task-averaged regret of our meta learning algorithm. (2) The comparisons with the state-of-the-art (Balcan et al., 2021, Alg 3) for non-convex piecewise-Lipschitz OWO meta learning. Our Algorithm 1 is actually based on the modification of (Balcan et al., 2021, Alg 3), and the two main differences are as follows: (i) For learning the initialization $\rho$, we use FTRL algorithm to achieve the regret of $O(T^{1/2+\alpha})$, $\alpha \in (0,\frac{1}{2})$, and the mixture parameter $\gamma = m^{d\beta}/T^{\alpha}$ is irrelevant to the optimal $\rho^*$ (hence $\gamma$ can be set in advance). Existing work (Balcan et al., 2021) also uses FTRL algorithm, but (Balcan et al., 2021) leverages a more complicated analysis, attaining a larger regret bound $O(m^{d/2}T^{3/4})$. Nevertheless, their choice of $\gamma = \frac{\mathrm{KL}(\rho^*||\hat{\nu})^{1/4}m^{d\beta/2}}{T^{1/4}V_d^{1/2}}$ in (Balcan et al., 2021, Thm 3.2) depends on the knowledge of optimal distribution $\rho^*$ that contains information of $T$ training tasks, which is unfeasible in the sequential online meta learning setting. (ii) For learning the step size $v$, we choose FTL algorithm to learn the step size $v_t = argmin_{v\in[B^2,D^2]}\sum_{s=1}^{t-1}h_s(v) = \left(\sum_{s=1}^{t-1}f_s(\rho_{s1})/(t-1)\right)^{1/2}$ to derive a logarithmic regret bound of $O(\log T)$. In contrast, (Balcan et al., 2021) chooses $\epsilon$-FTL algorithm to optimize the functions $\{h_t(v) = v + (f_t(\rho_{t1}) + \epsilon^2)/v\}_{t\in[T]}$ on the domain $[0,D^2]$ (where $D^2 \geq \max_t f_t(\rho_{t1})$) to learn the step size $v_t = \arg\min_{v\in[0,D^2]}\sum_{s=1}^{t-1}h_s(v) = \left(\sum_{s=1}^{t-1}f_s(\rho_{s1})/(t-1) + \epsilon^2\right)^{1/2}$, and lead to the regret $O(T\epsilon^2 + (\log T)/\epsilon^2)$, which is of $O(\sqrt{T}\log T)$ if we set $\epsilon = 1/T^{1/4}$ and is slower than our regret bound. (3) Practical implementation of meta learning algorithm. However, (Balcan et al., 2021) uses a heuristic method to learn (instead of meta learn) the step size $\lambda_t$, so their implementation is not rigorously the same as their pseudo code in (Balcan et al., 2021, Alg 3). We follow almost the same implementation details as (Balcan et al., 2021) (which is also explained in detail in the Experimental Setting part), with the main difference being that we verify the regret bound over training tasks but (Balcan et al., 2021) verify the regret bound over test tasks. Thus, we do not compare the convergence performance of regret of (Balcan et al., 2021, Alg 3) in our experiments.*

