# OpenReview forum: "Improved Regret Bounds for Non-Convex Online-Within-Online Meta Learning"
_ICLR.cc/2024/Conference — ICLR 2024 poster_

### Official Review · Reviewer_NU9X · 2023-10-20

**Soundness:** 3 good
**Presentation:** 2 fair
**Contribution:** 3 good
**Rating:** 6
**Confidence:** 2

**Summary:**

In this work, the authors consider the problem of non-convex online within online (owo) meta learning, which is a framework where a learner has to adapt to several online learning problems sequentially, and aims to use knowledge of the learned learning rate and step size of previous tasks to speed up learning.
The author first consider the same restrictions for the loss function (namely bounded and piece-wise Lipschitz) as in a previous work of Balcan et al. (2021) and widely improve the results, deriving regret bounds with improved dependencies on the number of tasks $T$ and the number of episodes within each task $m$. This algorithm is easy to implement and experiments can be found in the appendix that highlight that transferring information helps.

They then propose a different more general approach, which only requires the assumption of bounded loss functions. They derive regret bounds that are tighter than for the first algorithm, and are even sharper than the bounds derived by Khodak et al. (2019). While the theoretical results hold, it is also indicated in the appendix that they cannot manage to implement the algorithm, which limits the practical interest of this algorithm to specific and easy to implement problem instances.

Finally, they also provide generalization bounds for statisitical meta-learning and PAC-Bayesian bounds for statistical multi-task learning.

**Strengths:**

The paper proposes significant improvements in the analysis of the non-convex OWO problem, which is a complex problem that has not been widely studied up to now.
The proposed algorithms rely on well studied frameworks of online learning, notably the EWA algorithms as well as the FTRL and FTL framework.
Using a similar framework as the only exisiting previous result for non convex OWO (to the best of my knowledge), they propose an algorithm with significantly tighter bounds in terms of $T$ and $m$, which can be implemented and appears to perform fine in experiments.

The second algorithm appears simple, but achieves a significant improvement.

The generalization bounds are novel and a good addition to the work.

From what I got to see, the proofs seem correct and are well detailed.

**Weaknesses:**

It would be good to have a more detailed explanation of why the second algorithm does not work well in practice.
In particular, the EWA step is normally expressed in close form solution and thus computation should not be an issue.
Detailing the relation between the EWA and the FTRL formulations of Algorithm 2 l.4.  and how that prevents computations would be a good step towards understanding the limitations of this algorithm.
In the current format of the paper, it is necessay to reach the last remark of the appendix to understand why the result of Theorem 2 is not
directly eclipsed by the more general and tighter bound of Theorem 3, which affects the clarity of the paper.

A discussion of the lower bounds and of the optimality of the results is lacking and would help getting a better understanding of how much the bounds can be further improved.

**Questions:**

Could you clarify the relation between the EWA and the FTRL formulations of Algorithm 2 l.4.  and how that prevents computations would be a good step towards understanding the limitations of this algorithm?

Could you discuss existence of the lower bounds for this problem and of the optimality of the results?

---

> ### Author Response · Authors · 2023-11-17
> **Response to the Review by Reviewer NU9X (Part I)**
>
> **Q1. Could you clarify the relation between the EWA and the FTRL formulations of Algorithm 2 l.4. and how that prevents computations would be a good step towards understanding the limitations of this algorithm?**\
> A1: Sorry for the confusion. We have clarified our Remark H.1 in the revised version, and our opinions for the difficulties of implementing Algorithm 2 are two-fold: \
> $\mathbf{(1)}$ It is true that the EWA step $ \rho _{ti}=argmin _{\rho\in\mathcal{P}(\Theta)}\operatorname{KL}(\rho||\rho _{t1})+\lambda _{t}\sum _{j=1}^{i-1}\langle \ell _{tj}, \rho\rangle$ in Algorithm 2 (i.e. line 4 in Algorithm 2) is also a kind of FTRL formulations, and the explicit form of $\rho _{ti}$ is $$\frac{\mathrm{d}\rho _{ti}}{\mathrm{d}\rho _{t1}}(\theta)={\exp\\{-\lambda _{t}\sum _{j=1}^{i-1}\ell _{j}(\theta)\\} }\big/{\int\exp\\{-\lambda _{t}\sum _{j=1}^{i-1}\ell _{tj}(\theta)\\}\rho _{t1}(\mathrm{d}\theta)}.$$ Such $\rho _{ti}$ will be tractable in some cases where the RN derivative of  $\rho _{t1}$ w.r.t. Lebesgue measure $\nu$ is easy to compute (e.g. when $\rho _{t1}$ is the Lebesgue measure and hence $\rho _{ti}$ will be a Gaussian measure, see more details in [1, Sect 4]), and then the RN derivative of $\rho _{ti}$ w.r.t. Lebesgue measure $\nu$ (if we want to implement our algorithm in real life, we need to compute $\frac{\mathrm{d} \rho _{ti}}{\mathrm{d}\nu}$ to let $\rho _{ti}$ be tractable in Euclidean space) will also be easy to compute by the chain rule of RN derivative $\frac{\mathrm{d} \rho _{ti}}{\mathrm{d}\nu} = \frac{\mathrm{d} \rho _{ti}}{\mathrm{d}\rho _{t1}}\cdot\frac{\rho _{t1}}{\mathrm{d}\nu} $.  \
> $\mathbf{(2)}$ The computation problem actually lies in line 6 (i.e. the meta update step of $\rho _{t+1,1} = \frac{1}{t} \sum _{s=1}^{t} \rho _{sm}$) in Algorithm 2, which leads to the difficulty of computing the analytic form of RN derivative of $\rho _{t+1,m}$ w.r.t. the Lebesgue measure $\nu$, even though $\rho _{sm}$ is the Lebesgue measure (i.e. corresponding to uniform distribution) or Gaussian measure (i.e. corresponding to Gaussian distribution). Concretely, consider the simplest case where $\rho _{11}$ is the Lebesgue measure $\nu$, and hence using EWA obtains the Gaussian measure $\rho _{1m}$ (with density $\mathcal{N}(\mu  _{1}, \sigma  _{1})$). Assume that $\rho _{2m}$ is also a Gaussian measure (with density $\mathcal{N}(\mu _{2}, \sigma _{2})$), then $\rho _{31}=\frac{\rho _{1m}+\rho _{2m}}{2}$ and the density of $\rho _{3m}$ w.r.t. to $\nu$ is $\frac{\mathrm{d}\rho _{3m}}{\mathrm{d}\nu}=\frac{\mathrm{d}\rho _{3m}}{\mathrm{d}\rho _{31}}\cdot\frac{\mathrm{d}(\rho _{1m}+\rho _{2m})}{2\mathrm{d}\nu} \propto \exp\\{-\sum _{i\in[m]}\ell _{3i}\\}\cdot(\mathcal{N}(\mu _{1}, \sigma _{1})+ \mathcal{N}(\mu _{2},\sigma _{2}))$, indicating that $\rho _{3m}$ is not a Gaussian measure and it is hard to compute the precise form of the density function $\frac{\mathrm{d}\rho _{3m}}{\mathrm{d}\nu}$. Nevertheless, it will also be difficult to compute the precise form of the density function $\frac{\mathrm{d}\rho _{t+1,1}}{\mathrm{d}\nu}$, as well as the precise value of the normalized constant $\int\exp\\{-\lambda _{t+1}\sum _{j=1}^{m}\ell _{t+1,j}(\theta)\\}\rho _{t+1,1}(\mathrm{d}\theta)$. Therefore, it is not easy to compute the precise form of $\frac{\mathrm{d}\rho _{t+1,m}}{\mathrm{d}\rho _{t+1,1}}$ over the continuous domain.

---

> ### Author Response · Authors · 2023-11-17
> **Response to the Review by Reviewer NU9X (Part II)**
>
> **Q2. It is necessary to reach the last remark of the appendix to understand why the result of Theorem 2 is not directly eclipsed by the more general and tighter bound of Theorem 3, which affects the clarity.**\
> A2: Sorry for the confusion. We have made more explanations for the differences between our Theorem 2 and Theorem 3 in Remark B.5 in the revised version. The explanations are three-fold: \
> $\mathbf{(1)}$ The main difference between Theorem 2 and Theorem 3 is that they use different task similarity notions. Theorem 2 uses $V^{2}=min _{\rho:\Theta \mapsto \mathbb{R} _{\geq 0}, \int _{\Theta}\rho(\theta)\mathrm{d}\theta=1}-\frac{1}{T}\sum _{t=1}^{T}\log{\int _{\mathcal{B}(\theta _{t}^{*},m^{-\beta})}\rho(\theta)\mathrm{d}\theta} $ as the similarity notion between different tasks, but Theorem 3 uses the task similarity $V^{2}=min _{\rho \in \mathcal{P}(\Theta)}\frac{1}{T}\sum _{t=1}^{T}\operatorname{KL}(\rho _{t}^{\*}||\rho)$. Besides, the task similarity in Theorem 2 is defined according to the specific property (i.e. $\epsilon$-radius) of the piecewise-Lipschitz function, and is particularly applicable to the piecewise-Lipschitz setting.\
> $\mathbf{(2)}$ These two theorems use different techniques to learn initialization $\rho$. In Theorem 2, we use the action-space-discretization technique (described in Section 4.1, such technique is of independent interest) to translate the minimization problem $\min _{\rho \in \Theta}f _{t}(\rho)$ over the set of distributions into a tractable online convex optimization problem, and then use FTRL algorithm to get a regret bound of $O(T^{1/2 +\alpha})$ ($\alpha \in (0, 1/2)$). In Theorem 3, we use FTL algorithm to run over the functionals $\\{f _{t}(\rho)=\operatorname{KL}(\rho _{t}^{\*}||\rho)\\} _{t\in [T]}$, and develop a novel analysis (such as using the technical properties of Radon-Nikodym derivative) to derive the logarithmic regret $O(\log{T})$.\
> $\mathbf{(3)}$ At present, the first regret bound has higher application value than the second regret bound. The Algorithm 1 (with the first regret bound in Theorem 2) can be applied in the continuum domain, but Algorithm 2 (with the second regret bound in Theorem 3) at the current stage is still difficult to be applied in the continuum domain (see more explanations in our Remark H.1). \
> $\mathbf{(4)}$ The first improved regret is obtained under the same assumptions (e.g. piecewise-Lipschitz and bounded loss functions) as that in [2], via a more technical analysis. We list this improved regret in our work to make a fair comparison and show rigorous improvements over the existing work [2].
>
> **Q3. Could you discuss existence of the lower bounds for this problem and of the optimality of the results?**\
> A3: Thanks for your comments. We have added the following discussion as our Remark B.6 in the revised version: According to our Theorem 1, we decompose the task-averaged regret bound problem into two subproblems: **(1)** minimizing $\\{f _{t}(\rho)=V(\rho, \rho _{t}^{*})^{2}\\} _{t \in [T]}$ to learn initialization $\rho$, and **(2)** minimizing $\\{h _{t}(v) = v + \frac{f _{t}(\rho _{t1})}{v}\\} _{t \in [T]}$ to learn step size $v$. Combining the above two results leads to task-averaged regret bounds in our Theorems 2~3, which are actually optimal w.r.t. $m$ (i.e. of order $O(\sqrt{m})$). Therefore, what we can improve is the convergence rate w.r.t. $T$, and our explanations are three-fold: \
> $\mathbf{(1)}$ For learning the initialization $\rho$, our Proposition 4 achieves a logarithmic regret $O(\log{T})$. According to the Addition Related Work of OCO in Appendix A, the optimal regret for strongly-convex online optimization is $O(\log{T})$. Therefore, we believe that our Proposition 4 achieves the optimal regret for learning the initialization. \
> $\mathbf{(2)}$ For learning the step size $v$, our Propositions 3 and 5 actually achieve the (polynomial) logarithmic regret $O(\log{T})$. Therefore, we also obtain optimal or near optimal regret for learning the step size $v$.\
> $\mathbf{(3)}$ Consider $\mathbf{(1)}$ and $\mathbf{(2)}$, if we still adopt the regret upper bound decomposition framework, we should refine the proof of our Theorem 1. For example, there seems to be some improvement space in the 4-th inequality in the proof of our Theorem 1 (since other inequalities in this proof hold due to the definition of regret upper bound); If not, we should find other task-averaged regret analysis to see whether we can obtain better convergence rate w.r.t. $T$ or a smaller multiplier constant.
>
> **Reference**\
> [1] Pierre Alquier. Non-exponentially weighted aggregation: Regret bounds for unbounded loss functions. In International Conference on Machine Learning (ICML), pp. 207–218, 2021.
>
> [2] Maria-Florina Balcan, Mikhail Khodak, Dravyansh Sharma, and Ameet Talwalkar. Learning-to-learn non-convex piecewise-lipschitz functions. In Advances in Neural Information Processing Systems (NeurIPS), pp. 15056–15069, 2021.

---

> ### Comment · Reviewer_NU9X · 2023-11-22
>
> Thank you for the detailed feedback. I donøt have any further questions at this point.

---

> > ### Author Response · Authors · 2023-11-22
> > **Thank you for your positive feedback!**
> >
> > Dear Reviewer NU9X, \
> > Thanks for your positive feedback and  appreciation! And thank you again for your constructive comments on our paper.

---

### Official Review · Reviewer_8VuH · 2023-10-28

**Soundness:** 2 fair
**Presentation:** 2 fair
**Contribution:** 3 good
**Rating:** 6
**Confidence:** 2

**Summary:**

This paper first improves the regret bound of an existing non-convex online-within-online OWO) meta-learning algorithm for bounded and piecewise Lipschitz functions, and then design a new efficient OWO meta learning algorithm for bounded functions (maybe non-Lipschitz). This paper also derives a transfer risk bound and a PAC-Bayes bound for statistical meta learning via the regret analysis.

**Strengths:**

1) The problem of non-convex online-within-online OWO) meta-learning problem studied in this paper is interesting, which does have many real applications.
2) This paper has proposed two improved regret bounds for the non-convex OWO meta-learning problem.
3) The authors also extend the improved regret bound to the transfer risk bound and PAC-Bayes generalization bound for multi-task learning.

**Weaknesses:**

1) Although improved regret bounds are presented, the authors do not explain why their analysis and algorithm led to these improvements.
2) Moreover, it seems that the second improved regret bound is tighter than the first one, and holds in a more general case (non-Lipschitz). So, the value of the first one is not clear.
3) The transfer risk and PAC-Bayes generalization bounds are derived by using the online-to-batch and online-to-PAC techniques, which are incremental to some extent.
4) Although some experimental results are provided, no existing algorithms are compared and discussed in the experiments.

**Questions:**

1) The authors should explain why their analysis and algorithm led to the improvements in the regret bounds.
2) The authors should explain whether there exist some advantages of the first improved regret bound when it is compared with the second one.
3) The authors should conduct some experiments to compare their algorithms against existing algorithms.

---

> ### Author Response · Authors · 2023-11-17
> **Response to the Review by Reviewer 8VuH (Part I)**
>
> **Q1. The authors should explain why their analysis and algorithm led to the improvements in the regret bounds.**\
> A1: Thanks for your comments. We have explained our three technical novelties in deriving improved regret bounds for OWO meta learning in Remark B.2 of Appendix B in the original version, and the details are as follows: \
> $\mathbf{(1)}$ **The first novelty** lies in deriving improved regret bound for the online algorithm that runs over the functions $\\{h _{t}(v)=v+f _{t}(\rho _{t1})/v\\} _{t \in [T]}$ on the domain $[B^{2}, D^{2}]$ to learn the step size $v _{t}$ of EWA algorithm. Throughout the whole paper, we choose the efficient Follow-The-Leader (FTL) algorithm to learn the step size $v _{t}=argmin _{v \in [B^{2}, D^{2}]}\sum _{s=1}^{t-1}h _{s}(v)=\big({\sum _{s=1}^{t-1}f _{s}(\rho _{s1})/(t-1)}\big)^{{1}/{2}}$ and use the primal-dual analysis from [1, Cor 1] to derive a logarithmic regret bound of $O(\log{T})$. The key step in obtaining the logarithmic regret $O(\log{T})$ is to show that $\min _{t \in [T]}f _{t}(\rho _{t1})$ is strictly positive (i.e. $B>0$) to guarantee the strong-convexity of $h _{t}(v)$ and the boundedness of $\partial h _{t}(v _{t})$ (i.e. the Lipschitz property of $h _{t}(v)$ at the point $v _{t}$). The positiveness of $\min _{t \in [T]}f _{t}(\rho _{t1})$ is guaranteed for piecewise Lipschitz functions in Proposition 3 and for non-Lipschitz functions in Proposition 5 respectively, via a fine-grained estimation of the lower bound $B^{2}$ of $f _{t}(\rho _{t1})$. In contrast, existing works [2, Prop B.2] and [3, Cor 3.2] both choose $\epsilon$-FTL algorithm to optimize the functions $\\{h _{t}(v)=v+(f _{t}(\rho _{t1})+\epsilon^{2})/v\\} _{t \in [T]}$ on the domain $[0, D^{2}]$ (where $D^{2}\geq\max _{t}f _{t}(\rho _{t1})$) to learn the step size $v _{t}$, and lead to the regret $O(T\epsilon^{2}+(\log{T})/\epsilon^{2})$, which is of $O(\sqrt{T}\log{T})$ if we set $\epsilon={1/T^{1/4}}$ and is slower than our bound. \
> $\mathbf{(2)}$ **The second novelty** lies in deriving improved regret bound for the online algorithm that runs over the functions $\{f _{t}(\rho)=V(\rho, \rho _{t}^{\*})\} _{t\in [T]}$ in the piecewise Lipschitz case. We use Follow-The-Regularized-Leader (FTRL) algorithm to achieve the regret of $O(T^{1/2+\alpha})$, $\alpha \in (0, \frac{1}{2})$, and the mixture parameter $\gamma = m^{d\beta}/T^{\alpha}$ is irrelevant to the optimal $\boldsymbol{\rho}^{\*}$ (hence $\gamma$ can be set in advance). Existing work [3, Thm 3.2] also uses FTRL algorithm, but [3] leverages a more complicated analysis, and attains a larger regret bound $O(m^{d/2}T^{3/4})$. Nevertheless, their choice of $\gamma=\frac{\operatorname{KL}(\boldsymbol{\rho}^{\*}||\hat{\boldsymbol{\nu}})^{1/4}m^{d\beta/2}}{T^{1/4}V _{d}^{1/2}}$ in [3, Thm 3.2]  depends on the knowledge of optimal distribution $\boldsymbol{\rho}^{\*}$ that contains information of $T$ training tasks, which is unfeasible in the sequential online meta learning setting. \
> $\mathbf{(3)}$ **The third novelty** lies in deriving improved regret bound for the online algorithm that runs over the functions $\\{f _{t}(\rho)=\operatorname{KL}(\rho _{t}^{\*}||\rho)\\} _{t\in [T]}$ in the non-Lipschitz case. Obtaining regret bounds for FTL algorithm run over the functions $\\{\operatorname{KL}(\rho _{t}^{*}||\rho)\\} _{t\in [T]}$ is hard, because $\operatorname{KL}(\rho _{t}^{\*}||\rho)$ is the functional of the probability distribution $\rho$, and we are unable to use traditional regret analysis for the functions over Euclidean space (e.g. the gradient boundedness and strong convexity analysis of the functions). Instead, we leverage the insightful Lemma E.2 to obtain the analytic form $\rho _{t1}=argmin _{\rho \in \mathcal{P}(\theta)}\sum _{s=1}^{t-1}\operatorname{KL}(\rho _{s}^{\*}||\rho)=\frac{1}{t-1}\sum _{s=1}^{t-1}\rho _{s}^{\*}$ of the solution of FTL algorithm. Then, we use this analytic form, as well as technical properties of RN derivative to estimate the upper bound of the regret and ultimately obtain a non-trivial logarithmic regret $O(\log{T})$, achieving so far the tightest regret bound for learning the initialization of EWA algorithm.

---

> ### Author Response · Authors · 2023-11-17
> **Response to the Review by Reviewer 8VuH (Part II)**
>
> **Q2. It seems that the second improved regret bound is tighter than the first one, and holds in a more general case (non-Lipschitz). So, the value of the first one is not clear.**\
> A2: Sorry for the confusion. The value/advantages of the first improved regret bound lie in the following 4 aspects, and we have made its value clearer in Remark B.5 in Appendix B of the revised version: \
> $\mathbf{(1)}$ The first improved regret bound in our Theorem 2 is not a special case of our Theorem 3. The main reason is that Theorem 2 uses $V^{2}=min _{\rho:\Theta \mapsto \mathbb{R} _{\geq 0}, \int _{\Theta}\rho(\theta)\mathrm{d}\theta=1}-\frac{1}{T}\sum _{t=1}^{T}\log{\int _{\mathcal{B}(\theta _{t}^{*},m^{-\beta})}\rho(\theta)\mathrm{d}\theta} $ as the similarity notion between different tasks, but Theorem 3 uses the task similarity notion $V^{2}=min _{\rho \in \mathcal{P}(\Theta)}\frac{1}{T}\sum _{t=1}^{T}\operatorname{KL}(\rho _{t}^{\*}||\rho)$. Besides, the task similarity in Theorem 2 is defined according to the specific property (i.e. $\epsilon$-radius) of the piecewise-Lipschitz function, and hence is particularly applicable to the piecewise-Lipschitz setting.\
> $\mathbf{(2)}$ The action-space-discretization technique (described in Section 4.1 to obtain the first regret bound in Theorem 2) is of independent interest. The defined task similarity in Theorem 2 also requires a novel action-space-discretization method to translate the minimization problem $\min _{\rho \in \Theta}f _{t}(\rho)$ over the set of distributions into a tractable online convex optimization problem. \
> $\mathbf{(3)}$ At present, the first regret bound has higher application value than the second regret bound. The Algorithm 1 (corresponding to the first regret bound in our Theorem 2) can be applied in the continuum domain, but Algorithm 2 (corresponding to the second regret bound in our Theorem 3) at the current stage is still not easy to be applied in the continuum domain (see more explanations in our Remark H.1). \
> $\mathbf{(4)}$ The first improved regret is obtained under the same assumptions (i.e. piecewise-Lipschitz and bounded loss functions) as that in [3], via a more technical analysis. We list this improved regret in our work to make a fair comparison and show rigorous improvements over the existing work [3].
>
> **Q3. The transfer risk and PAC-Bayes generalization bounds are derived by using the online-to-batch and online-to-PAC techniques, which are incremental to some extent.**\
> A3: Thanks for your comments. But we believe our generalization bounds for statistical meta learning are not incremental, due to the following 2 main reasons (which are also added as Remark G.1 in Appendix G of the revised version): \
> $\mathbf{(1)}$ **For the transfer risk bound in Theorem 4:** the novelties of our online-to-batch technique lie in 2 aspects: $\mathbf{(i)}$ The first novelty lies in bounding $\operatorname{KL}(\rho^{\*}||\rho _{T+1,1})$ with Proposition 4 and Proposition F.2, both of which require the technical analysis of the properties of the Radon-Nikodym derivative. To the best of our knowledge, such analysis does not exist in the previous online-to-batch literature. $\mathbf{(ii)}$ We further use concentration inequality to bound $ \mathbb{E} _{\mu \sim \tau}\frac{\operatorname{KL}(\rho^{\*}||\rho _{T+1,1})}{H}$, leading to the transfer risk bound in the non-convex setting. Such transfer risk bound, as shown in our Remark 3, has almost the same convergence rate when compared with the latest transfer risk bound in the convex setting (i.e. [2, Thm E.1], which is obtained by Jensen’s inequality of convex loss), demonstrating the novelty of the online-to-batch analysis developed in our Theorem 4. \
> $\mathbf{(2)}$ **For the PAC-Bayes generalization bound in Theorem 5:** first we need to admit that our Proposition 6 is a direct corollary of recent online-to-PAC result in [4, Thm 1] (i.e. extent the result [4] from single-task learning to our multi-task learning setting), but the novelty of our Theorem 5 lies in the combination of the online-to-PAC analysis and the task-averaged regret analysis developed in our Theorem 3. Such combination pioneers a new research direction to demonstrating PAC-Bayes generalization error bounds for statistical multi-task learning. If we do not combine the aforementioned two analysis tools, but just use the online-to-PAC analysis (i.e. [4, Thm 1]) and traditional regret analysis for single-task learning (i.e. [5, Thm 2.1]), we can only obtain the following PAC-Bayes bound for statistical multi-task learning:
> $$\frac{1}{T}\sum _{t=1}^{T}\mathbb{E} _{\theta \sim \mathcal{A} _{t}(S _{t})}\Big[\mathbb{E} _{z\sim \mu _{t}}\ell(\theta, z)-\frac{1}{m}\sum _{i=1}^{T}\ell(\theta,z _{ti})\Big]\leq \frac{\sqrt{m}M(1+\frac{1}{T}\sum _{t=1}^{T}\operatorname{KL}(\rho _{t}^{\*}||\rho _{t1}))}{m} + M\sqrt{\frac{2\log{\frac{1}{\delta}}}{Tm}},$$which is less informative than the generalization bound in our Theorem 5.

---

> ### Author Response · Authors · 2023-11-17
> **Response to the Review by Reviewer 8VuH (Part III)**
>
> **Q4. Although some experimental results are provided, no existing algorithms are compared and discussed in the experiments.**\
> A4: Thanks for your suggestion. We have discussed more differences between our Algorithm 1 and the state-of-the-art algorithm from [3, Alg 3] in Remark H.2 of Appendix H of the revised version. The detailed explanations are three-fold: \
> $\mathbf{(1)}$ The comparisons with the single-task algorithm. Actually, we choose single-task EWA algorithm as our baseline in all experiments. In both Figure 1 ($m$-shot multi-task method v.s. single-task baseline) and Figure 2 ($T =5$ multi-task method v.s. $T=1$ single-task baseline), we show advantages of the meta-learning based EWA algorithm over the single-task EWA algorithm baseline, and verify the convergence performance of task-averaged regret of our meta learning algorithm. \
> $\mathbf{(2)}$ The comparisons with the state-of-the-art algorithm for non-convex piecewise-Lipschitz OWO meta learning [3, Algorithm 3]. Our Algorithm 1 is actually based on the modification of [3, Algorithm 3], and the two main differences are as follows: \
> $\mathbf{(i)}$ For learning the initialization $\rho$, we use FTRL algorithm to achieve the regret of $O(T^{1/2+\alpha})$, $\alpha \in (0, \frac{1}{2})$, and the mixture parameter $\gamma = m^{d\beta}/T^{\alpha}$ is irrelevant to the optimal $\boldsymbol{\rho}^{\*}$ (hence $\gamma$ can be set in advance). Existing work [3] also uses FTRL algorithm, but [3] leverages a more complicated analysis, attaining a larger regret bound $O(m^{d/2}T^{3/4})$. Nevertheless, their choice of $\gamma=\frac{\operatorname{KL}(\boldsymbol{\rho}^{\*}||\hat{\boldsymbol{\nu}})^{1/4}m^{d\beta/2}}{T^{1/4}V _{d}^{1/2}}$ in [3, Thm 3.2] depends on the knowledge of optimal distribution $\boldsymbol{\rho}^{\*}$ that contains information of $T$ training tasks, which is unfeasible in the sequential online meta learning setting. \
> $\mathbf{(ii)}$ For learning the step size $v$, we choose FTL algorithm to learn the step size $v _{t}=argmin _{v \in [B^{2}, D^{2}]}\sum _{s=1}^{t-1}h _{s}(v)=\big({\sum _{s=1}^{t-1}f _{s}(\rho _{s1})/(t-1)}\big)^{{1}/{2}}$ to derive a logarithmic regret bound of $O(\log{T})$. In contrast, [3] chooses $\epsilon$-FTL algorithm to optimize the functions $\\{h _{t}(v)=v+(f _{t}(\rho _{t1})+\epsilon^{2})/v\\} _{t \in [T]}$ on the domain $[0, D^{2}]$ (where $D^{2}\geq\max _{t}f _{t}(\rho _{t1})$) to learn the step size $v _{t}=argmin _{v \in [0, D^{2}]}\sum _{s=1}^{t-1}h _{s}(v)=\big({\sum _{s=1}^{t-1}f _{s}(\rho _{s1})/(t-1)}+\epsilon^{2}\big)^{{1}/{2}}$, and lead to the regret $O(T\epsilon^{2}+(\log{T})/\epsilon^{2})$, which is of $O(\sqrt{T}\log{T})$ if we set $\epsilon={1/T^{1/4}}$ and is slower than our regret bound.\
> $\mathbf{(3)}$ Practical implementation of meta learning algorithm. However, [3] used a heuristic method to learn (instead of meta learn) the step size $\lambda _{t}$, so their implementation is not rigorously the same as the pseudo code in their paper (i.e. [3, Algorithm 3]). We follow almost the same implementation details as in [3] (which is also explained in detail in the Experimental Setting part in our Section H), with the main difference being that we verify the regret bound over the training tasks but [3] verify the regret bound over the test tasks. Therefore, we did not compare the convergence rate of regret of [3] in our experiments.
>
> **Reference**\
> [1] Shai Shalev-Shwartz and Sham M. Kakade. Mind the duality gap: Logarithmic regret algorithms for online optimization. In Advances in Neural Information Processing Systems (NeurIPS), pp.1457–1464, 2008.
>
> [2] Mikhail Khodak, Maria-Florina Balcan, and Ameet Talwalkar. Adaptive gradient-based metalearning
> methods. In Advances in Neural Information Processing Systems (NeurIPS), pp. 5915–5926, 2019.
>
> [3] Maria-Florina Balcan, Mikhail Khodak, Dravyansh Sharma, and Ameet Talwalkar. Learning-to-learn non-convex piecewise-lipschitz functions. In Advances in Neural Information Processing Systems (NeurIPS), pp. 15056–15069, 2021.
>
> [4] Gabor Lugosi and Gergely Neu. Online-to-pac conversions: Generalization bounds via regret analysis. arXiv preprint arXiv.2305.19674, 2023.
>
> [5] Pierre Alquier. Non-exponentially weighted aggregation: Regret bounds for unbounded loss functions.
> In International Conference on Machine Learning (ICML), pp. 207–218, 2021.

---

> > ### Comment · Reviewer_8VuH · 2023-11-22
> >
> > Thank the authors for the detailed feedback. My concerns are addressed, and I will increase my rating.

---

> > > ### Author Response · Authors · 2023-11-22
> > > **Thank you for your positive feedback!**
> > >
> > > Dear Reviewer 8VuH, \
> > > Thanks for your positive feedback and appreciation! And thank you again for your constructive comments on our paper.

---

### Official Review · Reviewer_ESpG · 2023-11-02

**Soundness:** 3 good
**Presentation:** 2 fair
**Contribution:** 3 good
**Rating:** 6
**Confidence:** 4

**Summary:**

In this paper, the authors study an Online Within Online (OWO) meta learning problem in which both the tasks and data within each task become available in a sequential order. The correlation among tasks is assumed to be helpful for learning the latter tasks. This paper provides a sub-linear regret w.r.t. the number of iterations $m$, and the regret exhibit a better convergence rate w.r.t. the number of tasks $T$. The authors investigate both non-convex piecewise Lipschitz continuous functions and non-Lipschitz functions in section 4.1 and 4.2, which makes the contribution fruitful. The proposed algorithm uses FT(R)L to update distribution $\rho_i$ in two different settings, which is motivated by previous work. The improvement comes from the learning initialization step and the choice of step size.

The authors are quite honest to discuss the limitation in Remark~B.1 of Appendix~B and Remark~H.1. Thus, there is no need to further discuss these issues in my comments.

I strongly recommend the authors to complement more details and explanations in the proof to make it more readable. For instance, I was quite confused on the third equation in the proof of Proposition~F.1., where the explanation that $\ell_i'$ ``is the i.i.d. copy...'' is not enough to show the equivalence, the boundedness of $\ell_i$ is also required. I cannot fully understand $\{\ell\}_{j=1}^{i-1}\sim \mu^{i-1}$, $j=0$ and $\mu^{i-1}$ are not defined. The issues like that are common in proofs. I have to admit that I was convincing myself some (in)-equalities are correct since the authors are honest to admit their issues. Due to the time limit, I can only proof check some part of the proofs. I hope other reviewers and the authors could help checking the theoretical part in the future.

**Strengths:**

Please refer to the summary.

**Weaknesses:**

Please refer to the summary.

**Questions:**

Please refer to the summary.

---

> ### Author Response · Authors · 2023-11-17
> **Response to the Review by Reviewer ESpG**
>
> **Q1. I was quite confused on the third equation in the proof of Proposition~F.1., where the explanation that $\ell _{i}$``is the i.i.d. copy...'' is not enough to show the equivalence, the boundedness of $\ell _{i}$ is also required.**\
> A1: Sorry for the confusion. We have detailed the explanations on the third equation in the proof of Proposition F.1 in the revised version, and the explanations are as follows: \
> $\mathbf{(1)}$ We start from the second equation in this proof:
> $$ \mathbb{E} _{\\{\ell _{i}\\} _{i=1}^{m}}	\mathbb{E} _{\ell \sim \mu}\frac{1}{m}\sum _{i=1}^{m}\int \ell(\theta) \rho _{i}(\mathrm{d}\theta)= \mathbb{E} _{\\{\ell _{i}\\} _{i=1}^{m}}	\frac{1}{m}\sum _{i=1}^{m}\int \mathbb{E} _{\ell \sim \mu}\ell(\theta) \rho _{i}(\mathrm{d}\theta).$$ The above equation holds due to the independence between $\ell \sim \mu$ and $\rho _{i} = \rho _{i}(\\{\ell _{j}\\} _{j=1}^{i-1})$, as well as Fubini-Tonelli’s Theorem for changing the order of integrals of non-negative function $\ell$.\
> $\mathbf{(2)}$ To achieve the third equation, we use the RHS of the above equation to obtain
> $$ \mathbb{E} _{\\{\ell _{i}\\} _{i=1}^{m}}	\frac{1}{m}\sum _{i=1}^{m}\int \mathbb{E} _{\ell \sim \mu}\ell(\theta) \rho _{i}(\mathrm{d}\theta) = \mathbb{E} _{\\{\ell _{i}\\} _{i=1}^{m}}	\frac{1}{m}\sum _{i=1}^{m}\int \mathbb{E} _{\ell’ _{i}	 \sim \mu}\ell’ _{i}(\theta) \rho _{i}(\mathrm{d}\theta),$$ where the equation holds due to the fact that $\ell’ _{i}$ is the independently and identically distributed copy of $\ell _{i}$ (hence $\\{\ell’ _{i}\\} _{i=1}^{m}$ and $\ell$ are drawn independently according to $\mu$) and the fact that $\rho _{i}$ only depends on $\\{\ell _{j}\\} _{j=1}^{i-1}$ (hence $\rho _{i}$ is independent of $\ell’ _{i}$). Such technique was also used in many existing works for deriving excess/transfer risk bounds, for example in the first inequality of the proof of Proposition A.1 in [1].
>
> **Q2. I cannot fully understand $\\{\ell_{j}\\}_{j=1}^{i-1} \sim \mu^{i-1}$ and $\mu^{i-1}$ are not defined.**\
> A2: Sorry for the confusion. $\\{\ell _{j}\\} _{j=1}^{i-1} \sim \mu^{i-1}$ is the abbreviation for the notation $\ell _{j} \sim \mu, \forall j \in [i-1]$; and $\mu^{i-1}=\mu \times \cdots \times \mu $ is the product measure of $i-1$ measures $\mu$. We have made the explanations in the proof part clearer in the revised version.
>
> **Reference**\
> [1] Mikhail Khodak, Maria-Florina Balcan, and Ameet Talwalkar. Adaptive gradient-based metalearning
> methods. In Advances in Neural Information Processing Systems (NeurIPS), pp. 5915–5926, 2019.

---

### Official Review · Reviewer_wQQZ · 2023-11-06

**Soundness:** 3 good
**Presentation:** 3 good
**Contribution:** 3 good
**Rating:** 6
**Confidence:** 3

**Summary:**

The authors consider the problem of meta-learning, in particular the online-within-online setting with non-convex loss functions.  They consider the initialization and step size of the Exponentially Weighted Aggregation (EWA) algorithm.  For functions that are bounded, non-convex, and piecewise Lipschitz they propose modifications to the state of the art algorithm and obtain better task-averaged regret bounds.  For bounded, non-convex, non-Lipschitz functions they propose a new method and obtain the first task-averaged regret bounds for that problem.  They also apply those results to obtain novel PAC-Bayes generalization bounds for meta-learning.

**Strengths:**

### Results
- For adversarial bounded, possibly non-convex, piecewise Lipschitz loss functions, the authors propose modifications to a state-of-the-art method in Balcan et al. (2021) and obtain significantly improved task-averaged regret bounds.  In contrast to prior bounds, the bound (Theorem 1) is sub-linear w.r.t. the number of per-task iterations $m$).  The dependence on the number of tasks $T$ is also improved.
- The authors consider a more general class of problems (removing assumption of piecewise Lipschitz) and propose a new algorithm and the first regret bounds for this setting.  Though there are some concerns about this set of results (discussed below)


### Writing & Soundness
- Overall I found the writing and organization to be good in terms of organization and clarity.  I did not carefully check the analysis, but as far I could tell the results appear sound.

**Weaknesses:**

### Significance of Algorithm 2 and its regret bound
The non-convex non-Lipschitz setting is challenging.  The authors derive the first task-averaged regret bounds  for non-convex non-Lipschitz loss functions.  However some issues (which are acknowledged and discussed in the appendix) lead to concerns about the significance of the results for this problem.
- It is noted in the appendix (Remark B.1, which is referenced at the end of Section 4) that due to the initialization update rule of the FTL algorithm, the regret bound in Theorem 3 may be vacuous under the regret definition in Eq (1) when the task optimal distribution $\rho_t^*$ is a Dirac measure.    It is mentioned in Remark B.1 that alternative update rules of $\rho_{t1}$ may address the issue but (i) are there candidates in mind and (ii) how badly might they impact the regret bound?  i.e. is it plausible that an analog to Proposition 4 could be found that would still yield the same or almost the same good task-averaged upper-regret bound as Theorem 3?
- It is also noted in the appendix (Remark H.1, which I don’t think was referenced in Section 4) that the analytic form of the RN derivatives can’t be computed in practice even for uniform or Gaussian distributions. This also leads to concerns over the significance of Theorem 3

### Experiments
- It was good to include some experiments, but only Algorithm 1’s performance is shown, not (Balcan et al. (2021)) or any other baseline, even though the experiments used are set up the same as those in (Balcan et al. (2021)).

**Questions:**

### Main comments/questions

1. Can you include some discussion on how the problem set up, methods, and regret bounds for OWO relate to the problem of online learning with dynamic comparators, (eg “Online Optimization : Competing with Dynamic Comparators” https://proceedings.mlr.press/v38/jadbabaie15.pdf and more recent works).  It seems for OWO meta-learning the comparator sequence would be fixed for each task and the changes would be known a priori (every $m$ rounds).  Perhaps there would be a significant gap between regret bounds from online optimization with dynamic comparators specialized to OWO (eg accounting for task similarities etc) versus regret bounds for methods designed specially for OWO.

### Minor comments/questions
2. Can you add some discussion for problem parameter sizes --- eg for some potential applications would the number of iterations $m$ be large while the number of tasks $T$ be much smaller or vice versa?
3. (Related works) “These regret bounds are irrelevant to the sample size m per task…” does that mean ‘the regret bounds do not depend on the sample size m per task’ or something else?

### Very minor notation/wording
- Abstract and intro “mete learning”
- Section 4.2 “at i-the round”
- Experiments “Lloyd” not “Llyod”
- Section 4 there is text in blue

---

> ### Author Response · Authors · 2023-11-17
> **Response to the Review by Reviewer wQQZ (Part I)**
>
> **Q1. It is mentioned in Remark B.1 that alternative update rules of $\rho_{t1}$ may address the issue but are there candidates in mind?**\
> A1: Thanks for pointing this out, we have made a more rigorous statement of Remark B.1 in the revised version. Our answer to this question is **No**, and we need to admit that at present it is still hard for us to find alternative update rules of $\rho_{t1}$ (i.e. the online algorithms to minimize sequence of $\operatorname{KL}(\rho_{t1}||\rho)$), due to the following two reasons: \
> $\mathbf{(1)}$ It is hard to use gradient descent based online algorithms to update $\rho_{t1}$. Since $\operatorname{KL}(\rho_{t}^{*}||\rho)$ is the functional of distribution $\rho$, it is hard to find the gradient of functional $\operatorname{KL}(\rho_{t}^{\*}||\rho)$ w.r.t. $\rho$. Thus, we are unable to use gradient-descent based online algorithms (e.g. online gradient descent or mirror descent) to minimize the sequence of $\operatorname{KL}(\rho_{t}^{\*}||\rho)$ to update $\rho_{t1}$.\
> $\mathbf{(2)}$ It is hard to use FTRL algorithm to update $\rho_{t1}$, due to two reasons: $\textbf{(i)}$ Note that the minimization objective function of FTRL is $\sum_{s=1}^{t-1}\operatorname{KL}(\rho_{s}^{\*}||\rho) + \lambda \operatorname{Norm}(\rho)$. As mentioned in the discussion under Proposition 4 of the main text, it is hard to find a norm (e.g. $L_{p}$-norm, $p\geq 1$) in the space of distributions $\rho$ to verify the strong-convexity or Lipschitzness of functional $\operatorname{KL}(\rho_{t}^{\*}||\rho)$. Thus, we are unable to utilize traditional analysis for the functions over Euclidean space from [1] to give the regret of the FTRL. Besides, it is also difficult to find the close form of $\rho_{t1} = argmin_{\rho \in \mathcal{P}(\theta)}\sum_{s=1}^{t-1}\operatorname{KL}(\rho_{s}^{\*}||\rho) + \lambda \operatorname{Norm}(\rho)$, and cannot use the analysis developed in the proof of our Proposition 4 to give the regret bound for FTRL algorithm (actually, even though we may derive the regret upper bound for this FTRL algorithm via the analysis tool from [2, Sect A.3], we are unable to derive the close form of $\rho$ to achieve knowledge transfer across different tasks). $\textbf{(ii)}$ If we set the regularization function in FTRL as $\operatorname{Norm}(\rho)=\operatorname{KL}(\rho||\pi)$ w.r.t. some reference distribution $\pi$, then the objective function is actually the same as that of FTL in our Proposition 4, because the update rule $\rho_{t1}=\frac{\pi + \sum_{s=1}^{t-1}\rho_{s}^{\*}}{t}$ is almost the same as that in our Algorithm 2.
>
> **Q2. Is it plausible that an analog to Proposition 4 could be found that would still yield the same or almost the same good task-averaged upper-regret bound as Theorem 3?**\
> A2: Thanks. Our explanations for the possible analog to Proposition 4 are three-fold: \
> $\mathbf{(1)}$ Its existence: we believe that the analog likely exists, because our regret bound for FTL in Proposition 4 only holds for the sequence of $\operatorname{KL}$-divergence $\operatorname{KL}(\rho_{t}^{\*}||\rho)$. However, we can obtain for generalized EWA algorithm (see [3, Thm 2.1]) regret bounds that have a similar form as that in our Eq.(3) but replace the $\operatorname{KL}$-divergence with other kinds of $f$-divergence (e.g. the $\chi^{2}$-divergence in [3, Cor 2.4]). Using online algorithms to minimize the sequence of $f$-divergence may lead to a new regret bound. \
> $\mathbf{(2)}$ Its convergence rate w.r.t. $T$: the convergence rate of regret bound in the analog will not be better than the logarithmic regret in our Proposition 4. Because without additional information of the loss function, logarithmic regret is optimal (see more explanations for online convex optimization in Appendix A). \
> $\mathbf{(3)}$ Its tightness when setting optimal distribution $\rho _{t}^{\*}$ as a Dirac measure: In online non-convex setting where the regret is defined as the gap between the cumulative loss (w.r.t. a sequence of distributions $\\{\rho _{ti}\\} _{i \in [m]}$)  and the minimal loss (w.r.t. the optimal distribution $\rho _{t}^{\*}$), the regret bounds of online algorithm always involve a divergence between the optimal distribution $\rho _{t}^{\*}$ and the initial distribution $\rho _{t1}$ (see examples in [3, Thm 2.1]). All of these $f$-divergences may be vacuous if we set $\rho _{t}^{\*}$ as a Dirac measure, indicating to some extent the limitation of the regret  (i.e. the gap between the cumulative loss and the minimal loss w.r.t. the optimal distribution $\rho _{t}^{\*}$) defined for EWA-type algorithm.

---

> ### Author Response · Authors · 2023-11-17
> **Response to the Review by Reviewer wQQZ (Part II)**
>
> **Q3. It is noted in the appendix (Remark H.1, which was not referenced in Section 4) that the analytic form of the RN derivatives can’t be computed in practice. This also leads to concerns over the significance of Theorem 3.**\
> A3: Thanks for this suggestion, we have referenced Remark H.1 at the end of Remark 2 in Section 4 in the revised version. Our explanations for the significance of our Theorem 3 lie in 3 aspects: \
> $\mathbf{(1)}$ It is truly that at the current stage we are unable to compute the analytic form of RN derivative of probability measure over continuum domain, because of the initialization update rule $\rho_{t1} = \frac{1}{t-1}\sum_{s=1}^{t-1}\rho_{s}^{\*}$. We treat this issue as one of the ongoing research directions of this work. \
> $\mathbf{(2)}$ We also need to point out that our Algorithm 2 is applicable to the discrete domain (e.g. to the Expert Advice problem for online model selection) where the updating rule $\rho_{t1} = \frac{1}{t-1}\sum_{s=1}^{t-1}\rho_{s}^{\*}$ corresponds to the averaging of previous $t-1$ discrete probability density vectors, which is more feasible and applicable than over continuum domain. \
> $\mathbf{(3)}$ Theorem 3 is not only useful to derive task-averaged regret bound for OWO meta learning algorithms, but also necessary to be applied to give PAC-Bayes generalization bounds for statistical meta learning model in our Theorem 5. This is also one of the significances of our Theorem 3.
>
> **Q4. It was good to include some experiments, but only Algorithm 1’s performance is shown, not (Balcan et al. (2021)) or any other baseline.**\
> A4: Thanks for your comments. More differences between our Algorithm 1 and the state-of-the-art algorithm from (Balcan et al. (2021)) have been discussed in Remark H.2 of Appendix H of the revised version, and our explanations are two-fold: \
> $\mathbf{(1)}$ Actually, we choose single-task EWA algorithm as our baseline in all experiments. In both Figure 1 ($m$-shot multi-task method v.s. single-task baseline) and Figure 2 ($T =5$ multi-task method v.s. $T=1$ single-task baseline), we show advantages of the meta-learning based EWA algorithm over the single-task EWA algorithm baseline, and verify the convergence performance of regret of our meta learning algorithm. \
> $\mathbf{(2)}$ It is true that our experimental setting is the same as that in (Balcan et al. (2021)). However, (Balcan et al. (2021)) used a heuristic method to learn (instead of meta learn) the step size $\lambda _{t}$, so their implementation is not rigorously the same as the pseudo code in their paper (i.e. their Algorithm 3). We follow almost the same implementation details as in (Balcan et al. (2021)) (which is also explained in detail in the Experimental Setting part of our Section H), with the main difference being that we verify the task-averaged regret bound over the training tasks but (Balcan et al. (2021)) verify the regret bound over the test tasks.

---

> ### Author Response · Authors · 2023-11-17
> **Response to the Review by Reviewer wQQZ (Part III)**
>
> **Q5. Can you include some discussion on how the problem set up, methods, and regret bounds for OWO relate to the problem of online learning with dynamic comparators?**\
> A5: Yes, we can a rough comparison between our task-averaged regret for OWO meta learning and the task-averaged regret obtained via dynamic regret analysis. The comparison is also added as Remark B.4 in Appendix B of the revised version.\
> $\mathbf{(1)}$ First, we need to derive a task-averaged regret bound for OWO meta learning through the lens of dynamic regret analysis. Denote $\phi _{t}(\theta)= \sum _{i=1}^{m}\ell _{ti}(\theta)$, then the regret for OWO meta learning in our Eq.(1) can be rewritten roughly as $\frac{1}{T}\sum _{t=1}^{T}\mathbb{E} _{\theta \sim \bar{\rho} _t}\phi _t(\theta)-\phi _t(\theta _t^{\*})$  (actually, we think we cannot rigorously rewrite the task-averaged regret as the equivalent form of dynamic regret, because we cannot write $\sum _{i=1}^{m}\mathbb{E} _{\theta \sim \rho _{ti}} \ell _{ti}(\theta)$ as the expectation of $\phi _{t}(\theta)$ over a common distribution $\rho \in \mathcal{P}(\Theta)$), where $\bar{\rho} _t  = \frac{1}{m}\sum _{i=1}^{m}\rho _{ti}$. Assume that $\sum _{t=1}^{T-1}\Vert\phi _t - \phi _{t+1}\Vert\leq V _{T}$, then according to the latest dynamic regret bound in [4, Thm 1] for non-convex online optimization (to the best of our knowledge this is the latest dynamic regret for non-convex online learning), the task-averaged regret is bounded by $\frac{1}{T}\sum _{t=1}^{T}\mathbb{E} _{\theta \sim \bar{\rho} _t}\phi _t(\theta)-\phi _t(\theta _t^{\*}) \leq O(\frac{\sqrt{T + V _{T}T}}{T})= O(\frac{\sqrt{1+ V _{T}}}{\sqrt{T}})$. \
> $\mathbf{(2)}$ We next compare the regret $O((\frac{\log{T}}{T}+V)\sqrt{m})$ in our work and the regret $ O(\frac{\sqrt{1+ V _{T}}}{\sqrt{T}})$ obtained via dynamic regret analysis, from 3 aspects: \
> $\mathbf{(i)}$ Our regret analysis does not adopt the bounded total variation assumption (i.e. $V _T$ is a bounded constant), when compared with dynamic regret analysis. \
> $\mathbf{(ii)}$ Our regret bound $O((\frac{\log{T}}{T}+V)\sqrt{m})$ is more informative, revealing the importance of task similarity $V$ to the generalization of OWO meta leaning algorithm. \
> $\mathbf{(iii)}$ Our regret bound $O((\frac{\log{T}}{T}+V)\sqrt{m})$ seems to have a faster convergence rate w.r.t. $T$ when compared with $ O(\frac{\sqrt{1+ V _{T}}}{\sqrt{T}})$.
>
> **Q6. Can you add some discussion for problem parameter sizes --- eg for some potential applications would the number of iterations be large while the number of tasks be much smaller or vice versa?**\
> A6: Yes. Consider the autonomous vehicles application, a self-driving car may drive in different environments (for example in a superhighway, in the miry road of the rural area, at the crossroads of a city, etc) and these environments can be considered as sequential different tasks. In each task, the self-driving car takes photos of the environment, uses the photos for training, and finally adjusts its route. The photos taken in different environments can be considered as the training samples available in a sequential order. In the above application, the number of environments is the number $T$ of training tasks in OWO meta learning, and the number of photos taken in each environment is the number $m$ of training samples per task. If the self-driving car encounters many environments, then maybe $T >> m$; if the car drives in the same environment for a long time and takes many photos, then maybe $m >> T$.
>
> **Q7. (Related works) “These regret bounds are irrelevant to the sample size m per task…” does that mean ‘the regret bounds do not depend on the sample size m per task’ or something else?**\
> A7: Yes. The regret bounds (e.g. $O(\log{T})$ or $O(\sqrt{T})$) for OWB meta learning only depend on the number $T$ of training tasks, and the properties of loss functions (e.g. $L$-Lipschitzness, smoothness, etc), but do not depend on the sample size $m$ per task.
>
> **Q8. Very minor notation/wording.**\
> A8: Many thanks for your detailed reading and kind suggestions. We have fixed the minor notation issues in the revised version. The blue text in Section 4 is used to help readers find the general definition of the task similarity $V$ in the dense main text.
>
> **Reference**\
> [1] Shai Shalev-Shwartz. Online learning and online convex optimization. Found. Trends Mach. Learn., 4(2):107–194, 2012. \
> \
> [2] Gabor Lugosi and Gergely Neu. Online-to-pac conversions: Generalization bounds via regret analysis. arXiv preprint arXiv.2305.19674, 2023. \
> \
> [3] Pierre Alquier. Non-exponentially weighted aggregation: Regret bounds for unbounded loss functions.
> In International Conference on Machine Learning (ICML), pp. 207–218, 2021. \
> \
> [4] Xiang Gao , Xiaobo Li , Shuzhong Zhang . Online Learning with Non-Convex Losses and Non-Stationary Regret. International Conference on Artificial Intelligence and Statistics (AISTATS), 2018.

---

> > ### Comment · Reviewer_wQQZ · 2023-12-05
> >
> > I thank the reviewers for their detailed responses.

---

### Comment · Area_Chair_j8gV · 2023-11-10
**Authors-Reviewers discussion starts today, ends on Nov 22**

Dear authors and reviewers,

@Authors: please make sure you make the most of this phase, as you have the opportunity to clarify any misunderstanding from reviewers on your work. Please write rebuttals to reviews where appropriate, and the earlier the better as the current phase ends on Nov 22, so you might want to leave a few days to reviewers to acknowledge your rebuttal. After this date, you will no longer be able to engage with reviewers. I will lead a discussion with reviewers to reach a consensus decision and make a recommendation for your submission.

@Reviewers: please make sure you read other reviews, and the authors' rebuttals when they write one. Please update your reviews where appropriate, and explain so to authors if you decide to change your score (positively or negatively). Please do your best to engage with authors during this critical phase of the reviewing process.

This phase ends on November 22nd.

Your AC

---

### Author Response · Authors · 2023-11-17
**Summary for Revision**

We sincerely thank all reviewers for their detailed reading and constructive comments. In the revision, we made the following major changes (rendered in purple in .pdf) regarding the reviewers' concerns:

1.	A more rigorous statement (in our Remark B.1) of the limitations of the regret bound in our Theorem 3 (In response to the Weakness by **Reviewer wQQZ**)

2.	A new remark (Remark B.4 in Appendix B) explaining the differences between our task-averaged bound for OWO meta learning and the task-averaged regret bound obtained via dynamic regret analysis (In response to the Question 1 by **Reviewer wQQZ**);

3.	More detailed explanations for the proof of Proposition~F.1 in Appendix F (As suggested by **Reviewer ESpG**);

4.	A new remark (Remark B.5 in Appendix B) explaining the advantages of the regret bound in our Theorem 2 when compared with the regret bound in our Theorem 3. (In response to Weakness 2 by **Reviewer 8VuH**);

5.	A new remark (Remark G.1 in Appendix G) explaining the two main technical novelties of deriving our generalization bounds for statistical meta learning. (In response to Weakness 3 by **Reviewer 8VuH**);


6.	A new remark (Remark H.2 in Appendix H) discussing detailed comparisons between our Algorithm 1 and other algorithms for OWO meta learning (In response to Weakness 4 by **Reviewer 8VuH**);

7.	Clarifying the implementation difficulty of our Algorithm 2 in Remark H.1 in Appendix H (In response to Question 1 by **Reviewer NU9X**);

8.	A new remark (Remark B.6 in Appendix B) discussing the optimality and the potential improvement space of our theoretical results (In response to Question 2 by **Reviewer NU9X**);


Besides the above major changes, we also fixed other minor typos.

---

### Meta-Review · Area_Chair_j8gV · 2023-12-05

**Metareview:**

This meta-review is a reflection of the reviews, rebuttals, discussions with reviewers and/or authors, and calibration with my senior area chair. This paper explores meta-learning, in particular the online-within-online setting with non-convex loss functions, and contributes an important analysis of a refined notion of regret. There is a consensus among reviewers that the rebuttals are convincing and the final reviews highlight the several merits of the paper.

**Justification For Why Not Higher Score:**

Some limitations noted by reviewers.

**Justification For Why Not Lower Score:**

Important new result on regret with sound justifications.

---

### Decision · Program_Chairs · 2024-01-16

Accept (poster)